# Statistically and Computationally Efficient Linear Meta-representation Learning

**Kiran Koshy Thekumparampil**[†]**, Prateek Jain**[‡]**, Praneeth Netrapalli**[¶]**, Sewoong Oh**[±] [∗]
[†]University of Illinois at Urbana-Champaign, [‡]Google Research India,
[¶]Microsoft Research India, [±]University of Washington, Seattle

## Abstract

In typical few-shot learning, each task is not equipped with enough data to be learned in isolation. To cope with such data scarcity, meta-representation learning methods train across many related tasks to find a shared (lower-dimensional) representation of the data where all tasks can be solved accurately. It is hypothesized that any new arriving tasks can be rapidly trained on this low-dimensional representation using only a few samples. Despite the practical successes of this approach, its statistical and computational properties are less understood. Recent theoretical studies either provide a highly suboptimal statistical error, or require many samples for every task, which is infeasible in the few-shot learning setting. Moreover, the prescribed algorithms in these studies have little resemblance to those used in practice or they are computationally intractable. To understand and explain the success of popular meta-representation learning approaches such as ANIL [43], MetaOptNet [36], R2D2 [9], and OML [33], we study a alternating gradient-descent minimization (AltMinGD) method (and its variant alternating minimization (AltMin) in the Appendix) which underlies the aforementioned methods. For a simple but canonical setting of shared linear representations, we show that AltMinGD achieves nearly-optimal estimation error, requiring only $\Omega(\mathrm{polylog}\,d)$ samples per task. This agrees with the observed efficacy of this algorithm in the practical few-shot learning scenarios.

## 1 Introduction

Common real world tasks follow a long tailed distribution where most of the tasks only have a small number of labeled examples [51]. Collecting more clean labels is often costly (e.g., medical imaging). As each task does not have enough examples to be learned in isolation under this few-shot learning scenario, meta-learning attempts to jointly learn across a large number of tasks to exploit some structural similarities among those tasks.

One popular approach is to learn a *shared representation*, where new arriving tasks can be solved accurately [45]. The premise is that $(i)$ there is a shared low-dimensional representation $f_U(x) \in \mathbb{R}^r$ represented by a task-independent meta-parameter $U$ and $(ii)$ a simple linear model of $\langle v_i, f_U(x) \rangle$ can make accurate prediction on the $i$-th task with a task-specific parameter $v_i$. Once the representation $f_U$ has been learnt, we can rapidly adapt to new arriving tasks as the representation dimension $r$ is much smaller than the dimension $d$ of the input data. This approach is becoming increasingly popular with a growing list of recent applications [33, 36, 9, 42, 43, 27, 47, 14, 13, 18] and has been empirically shown to achieve the state-of-the-art performances on benchmark few-shot learning datasets [47, 14, 43].

---

[∗][†]thekump2@illinois.edu, [‡]prajain@google.com, [¶]pnetrapalli@google.com (Part of the work was done while at Google Research India), and [±]sewoong@cs.washington.edu.

These successes rely on a simple but effective training algorithm which alternately updates $U$ and $\{v_i\}$ which we call AltMinGD (Alternating Minimization and Gradient Descent). Suppose we are given $t$ tasks, and the $i$-th task is associated with a dataset $\{(x_j^{(i)} \in \mathbb{R}^d, y_j^{(i)})\}_{j=1}^m$ of size $m$. In this paper, we closely follow the formulation of [47], which solves for a function $f_U : \mathbb{R}^d \to \mathbb{R}^r$ (typically a deep neural network) and a task-specific linear model $v_i \in \mathbb{R}^r$ on a choice of a loss $\ell(\cdot, \cdot)$:

$$\min_U \left\{ \sum_{i \in [t]} \min_{v_i \in \mathbb{R}^r} \sum_{j \in [m]} \ell(\langle v_i, f_U(x_j^{(i)}) \rangle, y_j^{(i)}) \right\}, \tag{1}$$

by alternately applying a (stochastic) gradient descent step of $U$ in the outer loop (for given $v_i$'s) and numerically finding the optimal solution $v_i$ in the inner loop (for a given $U$). Several closely related algorithms have been proposed, including separating training-set used for the inner loop and the validation-set used for the outer-loop [43, 36, 9, 5], early stopping the inner-loop [33], applying to datasets with imbalanced data sizes [42, 14], and proposing new architectures and regularizers [27]. There is an increasing list [47, 14, 43] of numerical evidences showing that these meta representation learning improves upon competing approaches including MAML [20] and its variants [23, 32, 39]. Further, [43] provides experimental evidences that shared representation is the dominant component in the efficacy of MAML [20], even though MAML does not explicitly seek a shared representation.

In this paper, we analyze the computational and statistical properties of AltMinGD and its variant AltMin under the simple but canonical setting of learning a shared linear representation for linear regression tasks [48]. The fundamental question of interest is: as the number of tasks grow, does AltMinGD learn the underlying $r$-dimensional shared representation (subspace) more accurately, and consequently make more accurate predictions on new tasks? This question is critical in explaining the empirical success in few-shot learning where the number of tasks in the training set is large while each of those tasks is data starved. Further, in settings like crowdsourcing or bioinformatics, collecting more data on new tasks is easier than collecting more data on existing tasks.

**Contributions.** We analyze the widely adopted AltMinGD and prove a nearly optimal error rate. We show that AltMinGD requires only $m = \Omega(\log t + \log \log(1/\varepsilon))$ samples per task to achieve an error of $\varepsilon$ in estimating the representation $U$ when we have a large enough number $t$ of tasks in the training data and assuming a constant dimensionality $r = O(1)$ of the representation. Under this condition, AltMinGD achieves an error decaying as $\widetilde{O}(\sigma\sqrt{d/mt})$, which nearly matches the fundamental lower bound. Together, these analyses imply that AltMinGD is able to compensate for having only a few samples per task (small $m$) by having many few-shot tasks (large $t$), significantly improving the state-of-the-art (see Table 1). Note that the $\log \log(1/\varepsilon)$ dependence of $m$ is hidden in the $\widetilde{\Omega}$ notation and is not explicitly visible from our main theorems or the table. A fine grained analysis showing this dependence is provided in Theorem 9 in Appendix C.

We follow the proof strategy of alternating minimization algorithms for matrix sensing [30, 38], but there are important differences making the analysis challenging. First, the meta-learning dataset does not satisfy Restricted Isometry Property (RIP) central in the existing matrix sensing analysis, and hence none of the technical lemmas can be directly applied. We leverage on the *task diversity property* in Assumption 2, to prove all necessary concentration bounds. Next, there is an inherent asymmetry in the problem; we require accurate estimation of $U$ for generalization to new arriving tasks (which is the primary goal of meta-learning), but we do not necessarily require accurate estimation of $v_i$'s. We exploit this to ensure accurate estimation of $U$ with a small $m$.

Our analysis of AltMinGD leads to a fundamental theoretical question: is the condition $m = \Omega(\log t)$ necessary? We introduce a variation AltMinGD-S, which at each iteration selects a subset of tasks that are well-behaved (covering the $r$-dimensional subspace of current estimated $U$) and uses (the empirical risk of) only those tasks in the update. While $\log t$ dependence is unavoidable if we require all $t$ tasks to be well-behaved, ensuring a large fraction to be well-behaved requires smaller $m$. When the noise is sufficiently small with variance $O(1/\log t)$, we show that AltMinGD-S requires only $m = \Omega(\log \log(1/\varepsilon))$ (with no dependence on $t$) to estimate the shared representation accurately.

Inspired by a long line of successes in matrix completion and matrix sensing [30], we also analyze a variation of AltMinGD that alternately applies minimization for $U$ and $\{v_i\}$ updates, which we call AltMin, and prove a slightly improved guarantees at an extra computational cost of a factor of $dr^2$.

**Notations:** $[n] = \{1, 2, \ldots, n\}$. $\|A\|$ and $\|A\|_F$ denote the spectral and Frobenius norms of a matrix A. $\langle A, B \rangle$ denotes the inner product. $A^\dagger$ is the Moore-Penrose pseudoinverse. $x \sim \mathcal{N}(0, \mathbf{I}_{d \times d})$ means

that $x$ is a $d$ dimensional standard isotropic Gaussian random vector. $\widetilde{O}$, $\widetilde{\Omega}$ and $\widetilde{\Theta}$ hide logarithmic terms in dimension $d$, rank $r$, tolerance $\varepsilon$ and other problems parameters.

## 1.1 Related work

There is a large body of work in meta-learning since the seminal work in learning to learn [46], inductive bias learning [7], and multitask learning [12]. One popular approach starting from [28, 6] is to learn a shared low-dimensional representation for a set of related tasks. This is becoming increasingly popular with empirical successes in the few-shot learning scenarios [33, 36, 9, 42, 43, 27, 47, 14].

**Linear representation learning.** In this paper, we show that the popular AltMinGD algorithm for solving meta representation learning, achieves near-optimal error rate and sample complexity when applied to recovering linear representations, i.e. $f_U(x) = U^T x$. This problem has been studied in [3, 44, 40] and Nuclear-norm minimization approaches are proposed in [4, 24, 2, 41] but they do not provide subspace/generalization error guarantees and suffer from large training time. Closest to our work are [48, 35, 34, 19] which propose new algorithms with statistical guarantees. We also point out a concurrent and independent work [16], which also analyzes AltMinGD but for a special case of the noiseless setting. Authors empirically showed that AltMinGD performs better than other baseline federated learning algorithms for neural meta-representation learning on some datasets. We compare these results with our guarantees in Section 4.1.

Competing against meta-representation learning approaches listed above are the bi-level optimization based methods. A pioneer in this direction is MAML [20], which is analyzed under linear regression tasks in [15, 21, 8, 22]. [15] and [21] identify that MAML outperforms a simple Empirical Risk Minimization (ERM) when tasks are heterogeneous in their respective level of difficulty. [8] shows that, perhaps surprisingly, negative learning rate is optimal for MAML applied to linear regression tasks, where zero learning rate corresponds to the standard ERM.

**Matrix sensing.** Starting from matrix sensing and completion problems [11, 37, 30], recovering a low-rank matrix from linear measurements has been a popular topic of research. Linear meta-learning is a special case of matrix sensing, but with a non-standard sensing operator of the form $\mathcal{A}(UV^T) = [A_1(UV^T), \ldots, A_{mt}(UV^T)]$ where $A_{ij}(UV^T) = \langle x_{ij}e_i^\top, UV^\top \rangle$. This operator does not satisfy restricted isometry property in general, and existing matrix sensing results do not apply. Similar sensing operators have been studied in [29, 52] which gives $m = \Omega(d)$. We provide a significantly tightened analysis to require only $m = \Omega(\log t + \log\log(1/\varepsilon))$.

## 2 Problem Formulation: Meta-learning of Shared Representation

We focus on the meta-learning problem with a shared linear representation for linear regression tasks. Let $t$ denote the number of tasks. The $i$-th task is associated with $m$ samples $\{(x_j^{(i)} \in \mathbb{R}^d, y_j^{(i)} \in \mathbb{R})\}_{j=1}^m$. We assume there is a common *low-dimensional representation* $(U^*)^T x$ of each data point $x$, parameterized by $U^* \in \mathbb{R}^{d \times r}$ where $r \ll d$. The corresponding observation $y$ is sampled by regressing over the low-dimensional representation $(U^*)^T x$. Now, in general, learning $U^*$ is NP-hard [25]. Instead, similar to [48], we study the problem in the following tractable random design setting.

**Assumptions 1.** *Let $U^* \in \mathbb{R}^{d \times r}$ be an orthonormal matrix. For a task $i \in [t]$, with task specific parameter vector $v^{*(i)} \in \mathbb{R}^r$ and $j$-th example $x_j^{(i)} \sim \mathcal{N}(0, \mathbf{I}_{d \times d})$, its observation is:*

$$y_j^{(i)} = \langle x_j^{(i)}, U^* v^{*(i)} \rangle + \varepsilon_j^{(i)}, \tag{2}$$

*where $\varepsilon_j^{(i)} \sim \mathcal{N}(0, \sigma^2)$ is the measurement noise which is independent of $x_j^{(i)}$. We denote by $\widetilde{v}^{*(i)} = U^* v^{*(i)}$ the model parameter vector for each regression task in $d$-dimensions. We denote the matrix of these parameters as: $\widetilde{V}^* = U^*(V^*)^T$ where $(V^*)^T = [v^{*(1)}, \ldots, v^{*(t)}]$.*

The difficulty of estimating $U^*$ still depends on the *diversity* or incoherence of the tasks.

**Assumptions 2.** *Let $\lambda_1^*$ and $\lambda_r^*$ denote the largest and smallest eigenvalues of the task diversity matrix $(r/t)(V^*)^T V^* \in \mathbb{R}^{r \times r}$ respectively. Let $\kappa = \lambda_1^*/\lambda_r^*$. We say that $V^*$ is $\mu$-incoherent if*

$$\max_{i \in [t]} \|v^{*(i)}\|^2 \leq \mu \lambda_r^*. \tag{3}$$

To estimate the subspace $U$, we minimize the empirical risk of the $t$ tasks in the training data, over the meta-parameter $U \in \mathbb{R}^{d \times r}$ and the task-specific model parameters $V = [v^{(1)}, \ldots, v^{(t)}]^T \in \mathbb{R}^{t \times r}$:

$$\mathcal{L}(U, V) = \sum_{i=1}^{t} \sum_{j=1}^{m} \frac{1}{2} \left( y_j^{(i)} - \langle Uv^{(i)}, x_j^{(i)} \rangle \right)^2. \tag{4}$$

The problem is non-convex due to the bi-linearity of $U$ and $V$. We are interested in the few-shot learning setting where the goal is to learn the representation accurately despite a small number of samples per task in the training data. Now, even if the representation $U^*$ is known a priori, we would require $O(r)$ samples per task to learn the parameter $v$. Furthermore, information theoretically the total number of samples $m \cdot t$ should scale at most linearly with the data dimension $d$.

## 3 Alternating minimization

We focus on AltMinGD from [47], which learns a shared parameterized representation $f_U(\cdot)$ as in (1). Several variations of this algorithm are widely used, for example [43, 36, 9]. However, we note that these previous works neither referred to this algorithm as AltMinGD, nor explicitly related it to the Alternating Minimization (AltMin) framework [30]. To highlight this connection we follow the notations from the latter [30]. AltMinGD alternately updates the matrix of regression parameters $V$ using exact minimization with fixed $U$, and updates the representation parameter $U$ using the standard gradient descent step. Concretely,

$$v^{(i)} \in \arg\min_{v \in \mathbb{R}^r} \sum_{j \in [m]} \ell(\langle v, f_U(x_j^{(i)}) \rangle, y_j^{(i)}) \quad , \forall i \in [t],$$

$$U \leftarrow U - \eta \sum_{i \in [t]} \sum_{j \in [m]} \nabla_U \ell(\langle v^{(i)}, f_U(x_j^{(i)}) \rangle, y_j^{(i)}).$$

As $\ell(\cdot, \cdot)$ is typically a convex function, we can estimate $v^{(i)}$ efficiently for a fixed $U$. Note that many methods used in practice (ANIL [43] and MetaOptNet [36]) back-propagate through their respective inner (potentially inexact) optimization step. However, since we do exact inner minimization with respect to $V$, by a generalization of the Danskin's theorem [10], back-propagating through the inner minimization is equivalent to computing the gradient $\nabla_U \mathcal{L}(U, V)$ with respect to $U$ and then setting $V = V^*(U)$, where $V^*(U) \in \arg\min_V L(U, V)$ is the minimizer for the current $U$. That is, $\nabla_U \min_V L(U, V) = \nabla_U L(U, V^*(U))$. For the linear representation learning problem specified in Section 2, the above updates reduce to the following:

$$v^{(i)} \in \arg\min_v \sum_j (y_j^{(i)} - \langle x_j^{(i)}, Uv \rangle)^2, \text{ for all } i \in [t],$$

$$U \leftarrow U - \eta \nabla_U \mathcal{L}(U, V) = U + \eta \sum_{i,j} (y_j^{(i)} - \langle x_j^{(i)}, Uv^{(i)} \rangle) x_j^{(i)} (v^{(i)})^\top.$$

Given $U$, we can efficiently estimate each of the low $r$-dimensional regression parameters $v^{(i)}$'s *separately and in parallel* using standard least squares regression. Our analysis requires that when we update $V$ for current $U$, $U$ should be independent from the training points. Similarly, during the update for $U$, $V$ should be independent of the data points. We ensure the independence using two strategies: $(a)$ similar to standard online meta-learning settings [20], we randomly select (previously unseen) tasks to update $U$ and $V$, $(b)$ within each task, we divide the datapoints into two sets to update $V$ and $U$ separately. But in our experiments, we re-used all the samples at each iteration. Algorithm 1 presents a pseudo-code of AltMinGD applied to Problem (4). Note that in Algorithm 1, we apply QR-decomposition on $U$ after every $U$ update to ensure that magnitude of $U$ and $V$ does not stray far away from that of true $U^*$ and $V^*$, respectively. Otherwise, the sample complexity requirements of the algorithm increase in the condition number factors.

**Run-time and memory usage**: Exact update for $v^{(i)}$ has a time complexity of $O(mr^2 + r^3)$, which can be brought down to $O(m \cdot r)$ by using gradient descent for solving the least squares. Our analysis shows that under the sample complexity assumptions of Theorem 1, each of the least squares problem has a constant condition number. So, the total number of iterations for this update scale as $\log \frac{1}{\epsilon}$ to achieve $\epsilon$ error. If we set $\epsilon = 1/poly(t, \sigma)$, then using standard error analysis, we should be able to

---

**Algorithm 1** AltMinGD : Meta-learning linear regression parameters via alternating minimization gradient descent

---

**Required**: Data: $\{(x_j^{(i)} \in \mathbb{R}^d, y_j^{(i)} \in \mathbb{R})\}_{j=1}^m$ for all $1 \le i \le t$, $K$: number of steps, $\eta$: stepsize.

1   Initialize $U \leftarrow U_{\text{init}}$

2   Randomly shuffle the tasks $\{1,\ldots,t\}$

     **for** $1 \le k \le K$ **do**

3       $\mathcal{T}_k \leftarrow [1 + \frac{t(k-1)}{K}, \frac{tk}{K}]$

       **for** $i \in \mathcal{T}_k$ **do**

4         $v^{(i)} \leftarrow \arg\min_{\widehat{v} \in \mathbb{R}^r} \sum_{j \in [m/2]} \left(y_j^{(i)} - \langle U\widehat{v}, x_j^{(i)}\rangle\right)^2$

       **end**

5       $U \leftarrow U + \eta \sum_{i = \mathcal{T}_k} \sum_{j = 1 + \frac{m}{2}}^m \left(y_j^{(i)} - \langle Uv^{(i)}, x_j^{(i)}\rangle\right) x_j^{(i)} (v^{(i)})^\top$

6       $U \leftarrow \text{QR}(U)$

     **end**

7   **return** U

---

---

**Algorithm 2** AltMinGD-S : Meta-Learning regression parameters via AltMinGD over task subsets

---

**Required**: Data: $\{(x_j^{(i)} \in \mathbb{R}^d, y_j^{(i)} \in \mathbb{R})\}_{j=1}^m$ for all $1 \le i \le t$, $K$: number of steps, $\eta$: stepsize. Use the same steps as AltMinGD (Algorithm 1), but replace Line 3 with:

3   $\mathcal{T}_k \leftarrow \left\{i \in [1 + \frac{t(k-1)}{K}, \frac{tk}{K}] \;\middle|\; \sigma_{\max}(U^\top S^{(i)} U) \le 2; \; \sigma_{\min}(U^\top S^{(i)} U) \ge \frac{1}{2}; \right.$

         $\left. \text{where} \quad S^{(i)} = \frac{2}{m} \sum_{j \in [m/2]} x_j^{(i)} (x_j^{(i)})^\top \right\}$

---

obtain the optimal error rate in Theorem 9. The gradient descent update for $U$ requires $O(mt \cdot dr)$ time assuming large enough $mt$. Furthermore space complexity of AltMinGD is $O(dr + t \cdot r^2)$.

We provide an estimate of the statistical efficiency of the AltMinGD in Theorem 1. We also provide an analysis of the traditional Alternating Minimization algorithm (AltMin) which uses exact minimization for updating $U$ in the Appendix A. We obtain a slightly improved statistical guarantee for AltMin in terms of the condition number but its run-time is slower than that of AltMinGD.

### 3.1   Subset Selection

Algorithm 1 operates over all the tasks in a batch, each of which are generated using a random process. Now, if the number of tasks $t$ is large, then there is a non-trivial probability that some of the tasks are *outliers*, i.e., they have a large amount of error. This might lead to an arbitrary poor solution due to the outlier tasks. This is reflected in our analysis of AltMinGD (see Theorem 1), where the number of samples per task grows logarithmically with $t$ which is non-intuitive as typically larger number of tasks should not hurt the sample complexity.

In more general representation learning problems, when the number of tasks $t$ is large, there is more chance that some of them are outlier tasks. Ideally we want to design an estimator for shared representation $U$ that is robust to a few outlier tasks. For the linear representation learning problem, we observe that to ensure small error for a task, we require the Hessian to be well-conditioned. So, we compute the eigenvalues of the Hessian $U^\top S^{(i)} U$ for each task with the current $U$, and select only the tasks whose eigenvalues lie close to the expected repeated eigenvalue 1. Algorithm 2 applies this criteria to select tasks in each iteration, and then use the standard AltMinGD updates on those selected tasks. This leads to an improved dependence on $t$ as we show in Theorem 3.

**Run-time**: On top of the run-time complexity of AltMinGD, the subset selection scheme adds an additive $O(mt \cdot dr + t \cdot r^3)$ term. This arises due to $O(r^3)$ eigen-decompositions of $\widetilde{U}^\top S^{(i)} \widetilde{U} \in \mathbb{R}^{r \times r}$ for each task.

# 4 Statistical guarantees for Alternating Minimization algorithms

We reiterate that $\widetilde{O}$ and $\widetilde{\Omega}$ hide logarithmic terms in $d$ and $r$ and other problem parameters. We analyze the AltMinGD and AltMinGD-S algorithms using the rescaled Frobenius norm error: $\|(\mathbf{I} - U^*(U^*)^\top)U\|_F/\sqrt{r} \in [0, 1]$ between the rank-$r$ subspaces corresponding to the true $U^*$ and the output of the algorithm $U$. We first provide our main results analyzing these algorithms, and present detailed comparisons to previous results in Section 4.1

***AltMinGD:*** We first present our main result for the AltMinGD method (Algorithm 1), applied to the linear representation learning problem described in Section 2.

**Theorem 1** (Simplified version of Theorem 9 in Appendix C). *Let there be $t$ linear regression tasks, each with $m$ samples satisfying Assumptions 1, 2. Let $\kappa := \lambda_1^*/\lambda_r^*$ and let,*

$$m \geq \widetilde{\Omega}(r^2 + r \log t + \kappa \cdot (\sigma/\sqrt{\lambda_r^*})^2 r^2 \log t), \ t \geq \widetilde{\Omega}(\kappa \cdot \mu^2 r^3), \ \text{and}$$

$$mt \geq \widetilde{\Omega}(\kappa \cdot \mu dr^2 + \kappa^3 \cdot \mu dr^2 (\sigma/\sqrt{\lambda_r^*})^2).$$

*Then AltMinGD (Algorithm 1), initialized at $U_{\text{init}}$ s.t. $\|(\mathbf{I} - U^*(U^*)^\top)U_{\text{init}}\|_F \leq \min(21/121, \widetilde{O}(1/\kappa))$ and run for $K = \Omega(\lceil \kappa \log(mt/(\kappa \cdot \mu dr \cdot (\sigma/\sqrt{\lambda_r^*}))) \rceil)$ iterations with the stepsize $\eta = \frac{r/t}{2\lambda_1^*}$, outputs $U$ so that the following holds (w.p. $\geq 1 - K/(dr)^{10}$):*

$$\frac{\|(\mathbf{I} - U^*(U^*)^\top)U\|_F}{\sqrt{r}} \ \leq \ \widetilde{O}\left(\sqrt{\kappa}\left(\frac{\sigma}{\sqrt{\lambda_r^*}}\right)\sqrt{\frac{\mu dr}{mt}}\right). \tag{5}$$

**Remark 1 (Initialization)**: Our result holds if the initial point $U_{\text{init}}$ is reasonably accurate. One choice of initialization is to use the Method-of-Moments (MoM) [48]. Due to sub-optimality of MoM approach ([48, Theorem 3], also provided in Theorem 12 in Appendix), we get an additional sample complexity requirement of $mt \geq \widetilde{\Omega}(\kappa^2 dr^2 (\mu\kappa + r(\sigma/\sqrt{\lambda_r^*})^4)$. Note that this does not degrade the asymptotic error rate, $\widetilde{O}(\sqrt{dr/mt})$ when $\varepsilon = \widetilde{O}(\sqrt{dr/mt}) \to 0$. In our experiments, we observed that random initialization works just as well. Such a requirement of a good initialization is common in theoretical analyses of alternating update methods [30, 38], where it has been widely observed that random initialization works well in practice.

**Remark 2 (Generalization in few-shot learning)**: Learning a shared representation helps in generalizing to new arriving tasks in few-shot learning. Suppose we run Algorithm 1, under the conditions of Theorem 1 to get an estimated subspace $U$. Let a new task, whose task specific regression parameter $v^{*+}$ lie in $U^*$, be introduced with $m^+$ samples. Now, we can apply the step 4 of Algorithm 1, with $U$ and the new samples, to meta-learn an estimate $v^+$ of $v^{*+}$. Then the mean-squared-error (MSE) of the estimated parameter is $\widetilde{O}((\sigma/\sqrt{\lambda_r^*})(\mu dr^2/mt + r/m^+))$. Therefore, as long as $mt$ is large enough, we only need $m^+ = \Omega(r)$ additional samples to get an arbitrarily small MSE, as opposed to $m^+ = \Omega(d)$ of the trivial baseline of solving the new task by itself. We also improve upon other baselines from [48] in terms of dependence on $\sigma$ and $t$; see Section 4.1 and Table 1 for more details.

**Remark 3 (Near-optimality of the error rate):** We note that our error rate matches – up to $poly(\kappa, \mu)$ factors – the information theoretic lower bound given in Corollary 2.

**Corollary 2.** *[48, Theorem 5] Let $r \leq d/2$ and $mt \geq r(d - r)$, then for all $V^*$, w.p. $\geq 1/2$*

$$\inf_{\widehat{U}} \sup_{U \in \text{Gr}_{r,d}} \frac{\|(\mathbf{I} - U^*(U^*)^\top)\widehat{U}\|_F}{\sqrt{r}} \ \geq \ \Omega\left(\frac{1}{\kappa}\frac{\sigma}{\sqrt{\lambda_r^*}}\sqrt{\frac{dr}{mt}}\right), \tag{6}$$

*where $G_{r,d}$ is the Grassmannian manifold of $r$-dimensional subspaces in $\mathbb{R}^d$, the infimum for $\widehat{U}$ is taken over the set of all measurable functions that takes $mt$ samples in total from the model in Section 2 satisfying Assumption 1 and 2.*

However, the sufficient conditions on $mt$ in Theorem 1 has a factor $r$ gap from the necessary condition above, which we discuss with a concrete example in the next remark.

**Remark 4 (Gaussian example):** Let us interpret our result using a concrete example. Consider independent Gaussian parameters $v^{*(i)} \sim \mathcal{N}(0, (1/r)\mathbf{I}_{r \times r})$ such that the signal-to-noise ratio (i.e., $x^T U^* v^{*(i)}/\sigma^2$) is independent of $r$. Then with high probability $\|v^{*(i)}\| = \widetilde{\Theta}(1)$ and $\lambda_1^* = \lambda_r^* =$

$\widetilde{\Theta}(1)$. It follows that as per Assumption 2 the condition number $\kappa = \widetilde{\Theta}(1)$ and $\mu = \widetilde{\Theta}(1)$. To estimate $U^*$ up to an $\varepsilon$ error, AltMinGD needs a total of $mt = \widetilde{O}(dr^2 + \sigma^2 dr/\varepsilon^2)$ samples. The second term is dominant for small $\varepsilon$ and is optimal, which follows from the near-optimality in Remark 2. However, it is an open question if the first term is necessary, as the best known lower bound in the noiseless case will require $mt = \Omega(dr)$. In this well-behaved Gaussian case, AltMinGD requires $m \geq \widetilde{\Omega}(r^2 + (1 + \sigma^2)r \log t)$ per task samples.

**Remark 5 (Dependence on the minimum eigenvalue):** Notice that in the limit of $\lambda_r^* \to 0$, $V^*$ is rank deficient, thus making it impossible to recover the entire subspace of $U^*$. This is reflected in our Theorem 3 where the error-rate approaches the maximum possible value of one as $\lambda_r^*$ approaches zero (the LHS of Eq. (5) is at most one). However, for prediction error, smaller rank of $V^*$ implies smaller dimensional representation to be learned, thus the *prediction error* bound should *improve* with lower $\lambda_r^*$ (and also smaller rank of $V^*$). Proving a tight guarantee in the prediction error is more challenging and most of the existing results in matrix sensing literature [29] only provide guarantees in parameter estimation error. On the contrary, the lower-bound in (6) becomes zero as $\lambda_r^*$ decreases, implying that the lower-bound is significantly weaker in $\lambda_r^*$. This is expected since the lower-bound is derived through a lower-bound for the corresponding subspace regression loss. Intuitively when $\lambda_r^* = 0$ the tasks become less diverse (more homogeneous), and therefore the regression becomes easier. Such condition number mismatch in upper and lower-bounds are common in low-rank literature [30].

*Task subset selection (AltMinGD-S):* One downside of Algorithm 1 is that $m$ needs to increase with $t$ (i.e., $m = \Omega(\log t)$). We introduce AltMinGD-S in Algorithm 2 to study a fundamental question of whether this $\log t$ dependence is necessary. We show that when the noise is sufficiently small, AltMinGD-S achieves a per task sample complexity that does not increase with $t$.

**Theorem 3** (Simplified version of Theorem 11 in Appendix D). *Consider the setting of Theorem 1. Let $\kappa := \lambda_1^*/\lambda_r^*$.*

$$m \geq \widetilde{\Omega}(r^2 + \kappa \cdot (\sigma/\sqrt{\lambda_r^*})^2 r^2 \log t), \quad t \geq \widetilde{\Omega}(\kappa \cdot \mu^2 r^3), \quad and$$
$$mt \geq \widetilde{\Omega}(\kappa \cdot \mu dr^2 + \kappa^3 \cdot \mu dr^2 (\sigma/\sqrt{\lambda_r^*})^2).$$

*Then AltMinGD (Algorithm 2), initialized at $U_{\text{init}}$ s.t. $\|(\mathbf{I}-U^*(U^*)^\top)U_{\text{init}}\|_F \leq \min(21/121, \widetilde{O}(\frac{1}{\kappa}))$ and run for $K = \Omega(\lceil \kappa \log(mt/(\kappa \cdot \mu dr \cdot (\sigma/\sqrt{\lambda_r^*})))\rceil)$ iterations using the stepsize $\eta = \frac{r/t}{2\lambda_1^*}$, outputs $U$ so that the following holds (w.p. $\geq 1 - K/(dr)^{10}$):*

$$\frac{\|(\mathbf{I} - U^*(U^*)^\top)U\|_F}{\sqrt{r}} \leq \widetilde{O}\left(\sqrt{\kappa}\left(\frac{\sigma}{\sqrt{\lambda_r^*}}\right)\sqrt{\frac{\mu dr}{mt}}\right). \tag{7}$$

**Remark 7 (Bias of Subset Selection):** One may observe that this scheme may introduce a bias in the training data at each iteration. However, we control this bias by adding a new requirement that the number of tasks should be at least $t \geq \widetilde{\Omega}(\kappa \cdot \mu^2 r^3)$ (Theorem 6). This ensures that the only a small $O(1/\mu r)$ fraction of the tasks are discarded at each step (Lemma D.1 in the Appendix D.1), and this leads to a low bias. This requirement may be insignificant in our regime of interest where the number of tasks may be exponentially large, so that AltMinGD-S can provide a gain over AltMinGD.

**Remark 6 (When noise is small enough):** Note that when the noise variance $\sigma$ is small enough or when there are large number of tasks, i.e. $\sigma^2 \ll O(1/\log t)$, AltMinGD-S only needs $m \geq \widetilde{\Omega}(r^2)$ samples per task, assuming suitable initialization (see Remark 1). Furthermore, since AltMin-S selects a fraction of tasks to perform updates and the selection process requires only $O(mt \cdot dr + t \cdot r^3)$, the time-complexity of the method remains same as that of AltMinGD, up to constant factors. Next we see that AltMinGD-S removes the dependence of $m$ on $t$ completely, in the noiseless setting.

**Corollary 4.** *Let there be $t$ linear regression tasks, each with $m$ samples satisfying Assumptions 1, 2,*

$$m \geq \widetilde{\Omega}(r^2), \quad t \geq \widetilde{\Omega}(\mu^2 r^3 K), \quad and \quad mt \geq \widetilde{\Omega}(\mu dr^2 K).$$

*Additionally assume that the observations are noiseless, i.e. $\sigma = 0$. Then AltMinGD-S (Algorithm 2), initialized at $U_{\text{init}}$ s.t. $\|(\mathbf{I} - U^*(U^*)^\top)U_{\text{init}}\|_F \leq \min(21/121, \widetilde{O}(1/\kappa))$ and run for $K$ iterations using the stepsize $\eta = \frac{1}{2\lambda_1^*}$, outputs $U$ so that the following holds (w.p. $\geq 1 - K/(dr)^{10}$):*

$$\frac{\|(\mathbf{I} - U^*(U^*)^\top)\widetilde{U}\|_F}{\sqrt{r}} \leq (1 - \frac{1}{6\kappa})^K \widetilde{O}(\kappa). \tag{8}$$

The above corollary shows that in the noiseless setting, the per-task sample complexity for AltMinGD-S does not grow with $t$, and is nearly optimal. Also note that that even for noiseless setting, techniques like Method-of-Moments (MoM) still incur error of $\sqrt{dr/mt}$, ignoring $\kappa$ terms. In contrast, AltMinGD-S when initialized using MoM method (see Remark 1)), incurs just $\widetilde{O}(\exp(-t/\kappa))$ error. Proofs of Theorems 1 & 3 are in Appendix C.1 & Appendix D.1.

## 4.1 Sample complexity comparison

To the best of our knowledge, Theorems 1 and 3 presents the first analysis of an efficient method for achieving optimal error rate in $\sigma$, $d$ and $r$. [48] is most relevant that analyzes the landscape of the Empirical Risk Minimization (ERM) with Burer-Monteiro factorization. It shows that ERM can achieve a rescaled Frobenius norm error of $\varepsilon$ with $t$ tasks (assuming $t \geq d$), when $m \geq \widetilde{\Omega}(r^4 \log(t) + r^2 \log(t)\sigma^2/\varepsilon^2)$. We stress that this is highly sub-optimal as for small estimation $\varepsilon$, more tasks do not help improve the per-task sample complexity. This also does not reconcile with practice where more tasks tend to help accuracy and helps overcome small number of samples per-task. In contrast, AltMinGD requires $m \geq \widetilde{\Omega}(r^2(1 + \sigma^2)\log(t) + (r^2\sigma^2/\varepsilon^2)(d/t))$ where small error $\varepsilon$ can be achieved by collecting more tasks and increasing $t$. [19] studies the global minimizer of the non-convex ERM optimization in Eq. (4), without providing an efficient algorithm to solve it. The authors show that non-convex ERM achieves a small generalization error if $m = \widetilde{\Omega}(d)$, which is impractical in the few-shot learning setting.

Another approach is Method-of-Moments (MoM), which estimates $U$ by finding the principal directions of a particular 4th moment of the data [48, 35]. MoM can indeed trade-off smaller error $\varepsilon$ by increasing the number of tasks $t$. But the algorithm is inexact, i.e., even for $\sigma = 0$, we need $m \to \infty$ to achieve exact recovery of $U^*$; see Appendix G. This is in a stark contrast with AltMinGD and AltMinGD-S where for noiseless case, we can find $U^*$ *exactly*, as long as $m = O(r \log t + r^2)$ and $t = O(dr)$; see Figure 1a for an illustration. We consolidate these comparisons in Table 1.

Finally, a *concurrent and independent* work by [16] also analyzes AltMinGD but only for the special case when there is no noise, i.e., $\sigma = 0$. We show tighter results that are more generally applicable: $(i)$ our analysis applies to general noise $\sigma$ that is not necessarily zero, $(ii)$ even in the noiseless case, our analysis of AltMinGD is tighter and shows a smaller sample complexity, and $(iii)$ we present novel AltMinGD-S that further improves the sample complexity. Precisely, in the noiseless case, [16] proves that $m = \widetilde{\Omega}(\kappa^2 \cdot r^3 \log t)$ is sufficient for finding $U^*$ with a large enough $t$. Our tighter analysis shows that $m = \widetilde{\Omega}(r \log t + r^2)$ (Theorem 1) is sufficient with no dependence in $\kappa$. Note that the condition number $\kappa > 1$ and can be arbitrarily large depending on the problem instance. Further, we present a novel algorithm, AltMinGD-S , that only requires $m \geq \widetilde{\Omega}(r^2)$ (Corollary 4).

Table 1: Comparison of per-task sample complexity results $m(t, \varepsilon)$ to reach $\varepsilon$ error when solving linear meta-representation learning with $t$ tasks, $d$ dimensions, subspace rank $r = O(1)$ and noise variance $\sigma^2$ (Sections 4, 2); let $t > d$. We also report if the prescribed algorithm is computationally tractable and extendable to practical neural-net setting. AltMinGD-S relies on the eigen values of the data when projected onto $U$ and cannot be directly applied to neural networks.

| Analysis | Per-task sample complexity $m(t, \varepsilon)$ | Tractable? | Practical? |
|---|---|---|---|
| Non-convex ERM [19] | $\widetilde{\Omega}(d + \log(t) + \frac{\sigma^2}{\varepsilon^2})$ | No | – |
| Burer-Monteiro ERM [48] | $\widetilde{\Omega}(\log(t) + \frac{\sigma^2}{\varepsilon^2})$ | Yes | Yes |
| Method-of-Moments [48, 35] | $\Omega(1 + \frac{d}{t\varepsilon^2} + \frac{\sigma^2 d}{t\varepsilon^2})$ | Yes | No |
| AltMinGD (Theorem 1) | $\widetilde{\Omega}((1 + \sigma^2)\log t + \frac{\sigma^2 d}{t\varepsilon^2})$ | Yes | Yes |
| AltMinGD-S (Theorem 3) | $\widetilde{\Omega}(1 + \sigma^2 \log(t) + \frac{\sigma^2 d}{t\varepsilon^2})$ | Yes | No |
| Lower-bound [48] | $\widetilde{\Omega}(1 + \frac{\sigma^2 d}{t\varepsilon^2})$ | – | – |

## 5 Experimental results

In this section we empirically compare the performance of AltMinGD (Algorithm 1) and its exact minimization variant AltMin (Algorithm 3 in Appendix), two different versions of Method-of-

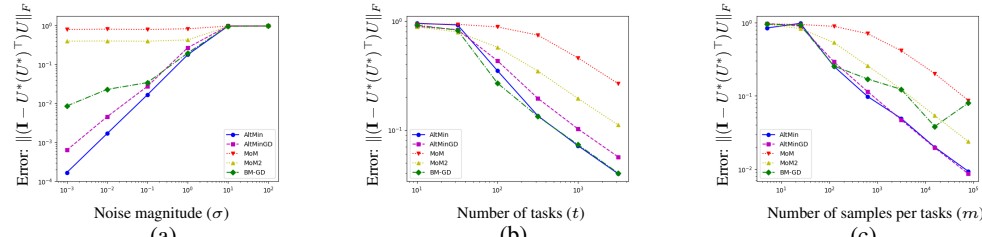

Figure 1: (a): AltMin and AltMinGD achieves vanishing error as noise $\sigma$ decreases, whereas the error of the two Method-of-Moments (MoM, MoM2) stay bounded away from zero. BM-GD, which seems unstable and hard to tune, achieves an intermediate level of error. (b), (c): AltMin, AltMinGD and BM-GD incurs significantly smaller error in estimation of true subspace $U^*$ than MoM and MoM2, both for growing number of tasks ($t$) and for growing number of samples per task ($m$).

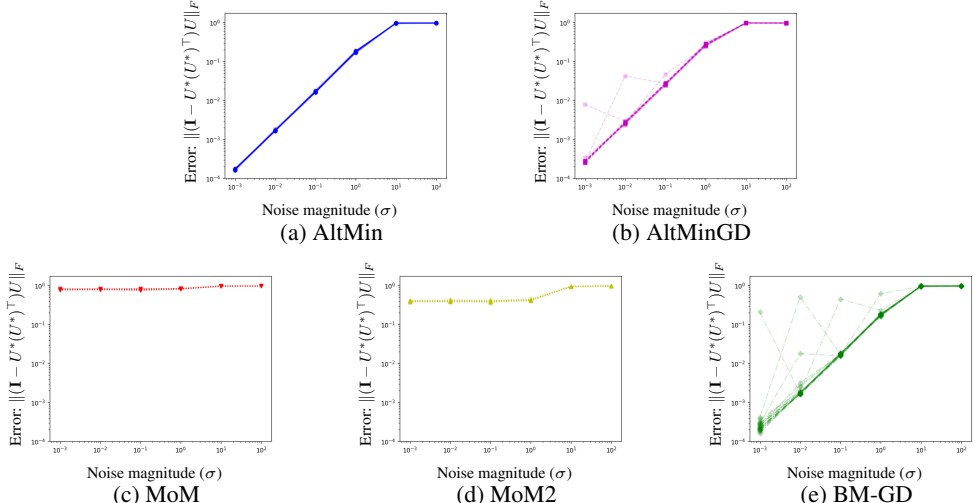

Figure 2: Compared to others, BM-GD is unstable and challenging to tune in the low-noise regime.

Moments (MoM [48], MoM2 [35]), and simultaneous gradient descent on $(U, V)$ using the Burer-Monteiro factorized loss (4) (BM-GD [48]). We omit AltMinGD-S here because the logarithmic gain ($1/\log(t)$) of AltMinGD-S will only be observed when we have an exponentially large number of tasks ($t$). This is challenging to simulate using our modest computing hardware. However, for big-data scenarios similar subset select schemes may be useful. In all the figures, the magenta dashed line with square marker represents AltMinGD, the blue straight line with circular marker denotes the AltMin , the red dotted line with downwards pointing triangular marker denotes the MoM, the yellow dotted line with upwards pointing triangular marker represents the MoM2, and the green dashed and dotted line with diamond marker represents the BM-GD. In all the figures we plot the subspace estimation error of the output $U$ of the algorithms. The error is calculated using the rescaled Frobenius norm $\|(\mathbf{I} - U^*(U^*)^\top)U\|_F/\sqrt{r}$, which takes a value in the interval $[0, 1]$. All results are averaged over multiple runs, and more experiments details and plots are provided in Appendix H.

Figure 1a plots subspace distance against the standard deviation $\sigma$ of the regression noise, $\varepsilon_j^{(i)} \sim \mathcal{N}(0, \sigma^2)$; see (2). Clearly, as predicted by Theorems 1 and 5 (in Appendix), the methods we consider—AltMinGD and AltMin —achieve smaller error than MoM methods for small noise regime. Here the error of AltMinGD and AltMin are linearly proportional to $\sigma$. However as predicted the error of MoM and MoM is a constant multiple of $\sqrt{dr^3/mt} = \sqrt{r}$ for all values of $\sigma$, and it does not improve when $\sigma$ decreases (see Table 1). While BM-GD does not have any known algorithmic guarantees, it still performs better than MoM methods. However, BM-GM becomes unstable and challenging to tune at low noise regime, even at a lower or comparable step-size than AltMinGD. To highlight this, the individual trials for each algorithm in this plot are plotted in Figure 2. Figure 1b plots the subspace error against the number of tasks $t$. In Figure 1c, we plot the the error against

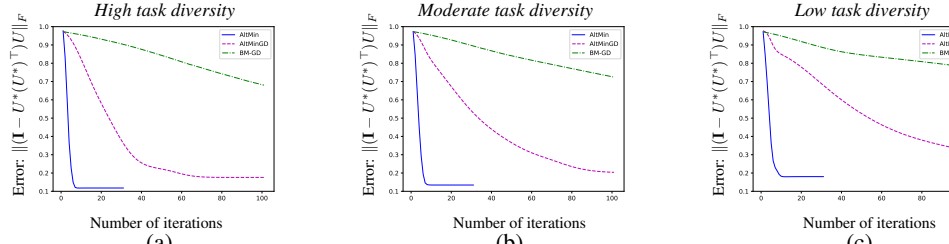

Figure 3: AltMin converges using fewer iterations than AltMinGD and BM-GD, but AltMinGD can be faster in practice due to its computationally cheaper iteration. While the performance of all the methods degrade as task diversity decreases, AltMin appears to be most robust to changes in diversity.

the number samples per tasks $m$. In both of these figures, we observe that, AltMinGD, AltMin and BM-GD achieve much smaller subspace error than the MoM and MoM2. Furthermore, as predicted, the squared error of all these methods decrease linearly in $m$ and $t$. We again note that BM-GD is unstable and hard to tune, especially for large $t$. The individual trials for each algorithm on these two plots are plotted in the Appendix H. ERM-based BM-GD performs poorer than Alternate Minimization-based AltMin and AltMinGD. This might be due to the presence of many bad local minima in the optimization landscape of the ERM problem jointly over $(U, V)$ [48, Theorem 2].

In Figure 3, we plot the subspace estimation error against the number of iterations of AltMinGD, Alt-Min, and BM-GD for varying levels of task diversity $\mu$ (Assumption 2). We observe that AltMin takes significantly fewer iterations to converge than AltMinGD and BM-GD, and AltMinGD converges earlier than BM-GD. However, each iteration of AltMin is very slow as it needs $O(d^3)$ operations, where as AltMinGD and BM-GD need only $O(d)$ operations per iteration. Therefore, AltMinGD could be the fastest in practical high-dimensional setting. BM-GD seems to be slower than AltMinGD because BM-GD seems to need a smaller stepsize than AltMinGD to stabilize its convergence. While all the methods perform worse when the task diversity decreases $((a) \rightarrow (b) \rightarrow (c))$, we see that AltMin is more robust than others. This may be attributable to AltMin's tighter dependence on the condition number $\kappa$ (Theorem 5, in Appendix) when compared to AltMinGD (Theorem 1).

## 6  Conclusion

When learning a shared representation for multiple tasks, a common approach is to alternate between finding the best linear model for each task on the current representation, and taking one gradient descent step to update the shared representation. This algorithm, AltMinGD, has been widely used in meta-representation learning with little theoretical understanding. We provide insights into the empirical success of AltMinGD by studying it in the canonical problem of linear meta-learning. We showed that, AltMinGD provides a nearly optimal error rate, along with nearly optimal per-task and overall sample complexities in their dependence in the dimensionality $d$ of the data. To the best of our knowledge, this is the first such optimal error rate that scales appropriately with the noise in observations, while still ensuring per-task sample complexity to be nearly independent of the dimensionality $d$. Latter is a key requirement in meta-learning as individual tasks are data-starved. The limitations of our results are: $(i)$ the analysis does not extend to non-linear representations, $(ii)$ the dependence on the rank $r$ of the shared subspace, the incoherence $\mu$, and the condition number $\kappa$ may not be tight; and $(iii)$ our analysis is "local" and requires a good initialization. We also proposed and analyzed a task subset selection-based method (AltMinGD-S) that further improves the per-task sample complexity and ensures that it is *independent* of the number of tasks in small noise or large number of tasks regime. However, the subset selection scheme heavily relies on the linearity of the shared representation. Therefore, this scheme cannot be directly applied to more practical neural network training. It also remains an open question if it is possible to achieve a per-task sample complexity that does not depend on the number of tasks $t$, even in the large noise setting.

Our work leads to several interesting future directions and questions. For the non-linear version of the problem, ensuring optimal error rate with optimal per-task sample complexity is an interesting open question. Finally, analyzing alternating minimization methods with stochastic gradients and streaming tasks is another promising direction. Our proof techniques could be combined with that of recent results in efficient one-pass SGD [31] to design a nearly optimal stochastic algorithm.

## Acknowledgements

Oh acknowledges funding from Google faculty research award, NSF grants IIS-1929955, CCF-1705007, CNS-2002664, CCF 2019844 as a part of Institute for Foundation of Machine Learning, and CNS-2112471 as a part of Institute for Future Edge Networks and Distributed Intelligence. We also thank anonymous reviewers for their reviews and suggestions for improving our manuscript.

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
