# Appendix

## Table of Contents

This appendix contains additional results, proofs for all the claims, and details of the experiments. Section A provides the alternating (exact) minimization algorithm (AltMin) and its task subset selection-based variant, and their statistical and computational guarantees. Sections C and D contain the analyses of Algorithm 1 and 2, respectively. Sections B and D contain the analyses of Algorithm 3 and 4, respectively. Section E contains corollaries of some known results. Section F contains some general technical lemmas used in this paper. Section H provides the details of the experiments.

## A  Alternating Exact Minimization algorithms

### A.1  Alternating minimization (AltMin) algorithm

In this section we discuss the alternating (exact) minimization algorithm (Algorithm 3), which has been widely studied in different but related problems including matrix completion, tensor completion, phase retrieval, and matrix sensing. The algorithm follows the standard alternating minimization procedure [30, 17] where we update the representation matrix $U$ and regression parameters $V$

---

**Algorithm 3** AltMin : Meta-Learning linear regression parameters via Alternating Minimization

---
**Required**: Data: $\{(x_j^{(i)} \in \mathbb{R}^d, y_j^{(i)} \in \mathbb{R})\}_{j=1}^m$ for all $1 \leq i \leq t$, $K$: number of steps.

1 Initialize $U \leftarrow U_{\text{init}}$

2 Randomly shuffle the tasks $\{1,\dots,t\}$

  **for** $1 \leq k \leq K$ **do**

3     $\mathcal{T}_k \leftarrow [1 + \frac{t(k-1)}{K}, \frac{tk}{K}]$

    **for** $i \in \mathcal{T}_k$ **do**

4         $v^{(i)} \leftarrow \arg\min_{\widehat{v} \in \mathbb{R}^r} \sum_{j \in [m/2]} \left( y_j^{(i)} - \langle U\widehat{v}, x_j^{(i)} \rangle \right)^2$

    **end**

5     $\widehat{U} \leftarrow \arg\min_{\widehat{U} \in \mathbb{R}^{d \times r}} \sum_{i=\mathcal{T}_k} \sum_{j=1+\frac{m}{2}}^m \left( y_j^{(i)} - \langle \widehat{U}v^{(i)}, x_j^{(i)} \rangle \right)^2$

6     $U \leftarrow \text{QR}(\widehat{U})$

  **end**

  **return** U

---

alternately. Note that, given $U$, we can estimate each of the parameters vector $v^{(i)}$ *separately* using standard least squares regression, i.e.,

$$v^{(i)} = \arg\min_v \sum_j (y_j^{(i)} - \langle x_j^{(i)}, Uv \rangle)^2 .$$

Similarly, given the updated regression parameter vectors $v^{(i)}$'s, we can now update $U$ as:

$$U = \arg\min_U \sum_{i,j} (y_j^{(i)} - \langle x_j^{(i)}, Uv^{(i)} \rangle)^2 .$$

To ensure ease of analysis, we analyze a modification of the algorithm where the columns of the next iterate form an orthonormal basis for the subspace containing the columns of $U$. Such a $U$ we can obtain by applying the QR-decomposition on the above $U$. However, this is not a necessity, as all the analyses of AltMin still follows through even without the QR decomposition due to a simple equivalence argument between the subspaces obtained with or without the QR decomposition. However, this equivalence argument critically uses the fact that we exactly minimize $U$ for the current $V$. Thus the same equivalence cannot be proved for AltMinGD which uses gradient descent to update $U$, because the gradient scaling may change across iterations due to changes in the magnitude of $U$.

Similar to AltMinGD, our analysis requires that when we update $V$ using current $U$, we require $U$ to be independent from the training data points. Similarly, during the update for $U$, we require $V$ to be independent of the data points. Again this is ensured using the same two strategies: a) similar to standard online meta-learning settings [20], we select random (previously unseen) tasks and update $U$ and $V$, b) within each task, we divide the data points into two sets to update $V$ and $U$ separately.

**Run-time and memory usage**: Although AltMin is a conceptually simpler algorithm than AltMinGD (Algorithm 1), per iteration cost of AltMin is larger than AltMinGD due to the exact minimization on $U$. Our update for $v^{(i)}$ require $O(mr^2 + r^3)$ time complexity, which can be brought down to $O(m \cdot r)$ by using gradient descent for solving the least squares. Our analysis shows that under the sample complexity assumptions of Theorem 5, each of the least squares problem has a constant condition number. So, the total number of iterations scale as $\log(1/\epsilon)$ to achieve $\epsilon$ error. If we set $\epsilon = 1/poly(t, \sigma)$, then using standard error analysis, we should be able to obtain the optimal error rate in Theorem 8. Similarly, *exact* update for $U$ requires $O((dr)^3 + mt \cdot (dr)^2)$ time, that decreases to $O(mt \cdot d \cdot r)$ when using gradient descent updates.

### A.1.1 Subset Selection (AltMin-S)

Similar to AltMinGD-S (Algorithm 2), to reduce the per-task sample complexity, we also provide an algorithm AltMin-S (Algorithm 4) based on selecting a subset of tasks. This uses the same subset selection scheme as AltMinGD-S. Since, AltMin-S selects a fraction of tasks to perform updates, it has the same run-time and memory complexities as AltMin.

---

**Algorithm 4** AltMin-S : Meta-Learning regression parameters via AltMin over task subsets

---

**Required**: Data: $\{(x_j^{(i)} \in \mathbb{R}^d, y_j^{(i)} \in \mathbb{R})\}_{j=1}^m$ for all $1 \leq i \leq t$, $K$: number of steps, $\eta$: stepsize. Use the same steps as AltMinGD (Algorithm 3), but replace Line 3 with:

3 $\mathcal{T}_k \leftarrow \left\{ i \in [1 + \frac{t(k-1)}{K}, \frac{tk}{K}] \mid \sigma_{\max}(\widetilde{U}^\top S^{(i)} \widetilde{U}) \leq 2; \ \sigma_{\min}(\widetilde{U}^\top S^{(i)} \widetilde{U}) \geq \frac{1}{2}; \right.$

$\quad$ where $\quad S^{(i)} = \frac{2}{m} \sum_{j \in [m/2]} x_j^{(i)} (x_j^{(i)})^\top \hfill \left. \right\}$

---

## A.2 Statistical guarantees for Alternating Exact Minimization algorithms

***Alternating minimization (AltMin)***: We first present our main result for a standard alternating minimization method (Algorithm 3) when applied to the meta-learning linear regression problem in the problem setting described in Section 2.

**Theorem 5** (Simplified version of Theorem 8 in Appendix B). *Let there be $t$ linear regression tasks, each with $m$ samples satisfying Assumptions 1, 2. Let $\kappa := \lambda_1^*/\lambda_r^*$ and let,*

$$m \geq \widetilde{\Omega}(r^2 + r\log t + (\sigma/\sqrt{\lambda_r^*})^2 r^2 \log t), \ t \geq \widetilde{\Omega}(\mu^2 r^3), \ \text{and}$$
$$mt \geq \widetilde{\Omega}(\kappa \cdot \mu dr^2 + \kappa \cdot \mu dr^2 (\sigma/\sqrt{\lambda_r^*})^2).$$

*Then AltMin (Algorithm 3), initialized at $U_{\text{init}}$ s.t. $\|(\mathbf{I} - U^*(U^*)^\top)U_{\text{init}}\|_F \leq \min(21/121, \widetilde{O}(1/\sqrt{\kappa}))$ and run for $K = \Omega(\lceil \log(mt/(\kappa \cdot \mu dr \cdot (\sigma/\sqrt{\lambda_r^*}))) \rceil)$ iterations, outputs $U$ so that the following holds (w.p. $\geq 1 - K/(dr)^{10}$):*

$$\frac{\|(\mathbf{I} - U^*(U^*)^\top)U\|_F}{\sqrt{r}} \leq \widetilde{O}\left( \left( \frac{\sigma}{\sqrt{\lambda_r^*}} \right) \sqrt{\frac{\mu r d}{m t}} \right). \tag{9}$$

**Remark 7 (Initialization)**: Our result holds if the initial point $U_{\text{init}}$ is reasonably accurate. One choice of initialization is to use the Method-of-Moments (MoM) [48]. Due to sub-optimality of MoM approach ([48, Theorem 3], also provided in Theorem 12 in Appendix), we get an additional sample complexity requirement of $mt \geq \widetilde{\Omega}(\kappa dr^2 (\mu\kappa + r(\sigma/\sqrt{\lambda_r^*})^4)$. Note that this does not degrade the asymptotic error rate, $\widetilde{O}(\sqrt{dr/mt})$ when $\varepsilon = \widetilde{O}(\sqrt{dr/mt}) \to 0$. Similar to case of AltMinGD, in our experiments, we observed that random initialization works just as well for AltMin too. This is analogous to alternating minimizaition for other problems where it has been widely observed that random initialization works well in practice [30, 38].

**Remark 8 (Optimality and comparison to AltMinGD)**: Similar to Theorem 1, this error rate is nearly optimal in terms of $d, r, m, t$ and $\sigma/\sqrt{\lambda_r^*}$, as it matches best possible rate when $V^*$ is specified a priori (Corollary 2). However, we see that the error rate, the additional sample complexities and required initial error are all tighter than Theorem 1 in terms of condition number $\kappa$ factors. However, in high-dimensions, one iteration of AltMinGD may be much faster than that of AltMin.

***Task subset selection (AltMin-S)***: Just as we did in AltMinGD-S (Algorithm 2), we reduce the dependence of the per-task sample complexity $m = \Omega(\log(t))$ of AltMin (Algorithm 3) on the number of tasks $t$. This achieved through a subset selection-based AltMin-S (Algorithm 4) algorithm, which has the following guarantees for noisy and noiseless ($\sigma = 0$) observations.

**Theorem 6** (Simplified version of Theorem 10 in Appendix D). *Let there be $t$ linear regression tasks, each with $m$ samples satisfying Assumptions 1, 2. Let $\kappa := \lambda_1^*/\lambda_r^*$ and let,*

$$m \geq \widetilde{\Omega}(r^2 + (\sigma/\sqrt{\lambda_r^*})^2 r^2 \log t), \ t \geq \widetilde{\Omega}(\mu^2 r^3), \ \text{and}$$
$$mt \geq \widetilde{\Omega}(\kappa \cdot \mu dr^2 + \kappa \cdot \mu dr^2 (\sigma/\sqrt{\lambda_r^*})^2).$$

*Then AltMin-S (Algorithm 4), initialized at $U_{\text{init}}$ s.t. $\|(\mathbf{I} - U^*(U^*)^\top)U_{\text{init}}\|_F \leq \min(21/121, \widetilde{O}(1/\sqrt{\kappa}))$ and run for $K = \Omega(\lceil \log(mt/(\kappa \cdot \mu dr \cdot (\sigma/\sqrt{\lambda_r^*}))) \rceil)$ iterations, outputs $U$ so that, w.p. $\geq 1 - K/(dr)^{10}$*

$$\frac{\|(\mathbf{I} - U^*(U^*)^\top)U\|_F}{\sqrt{r}} \leq \widetilde{O}\left( \left( \frac{\sigma}{\sqrt{\lambda_r^*}} \right) \sqrt{\frac{\mu r d}{m t}} \right). \tag{10}$$

**Corollary 7.** *Consider $t$ linear regression tasks, each with $m$ samples satisfying Assumptions 1 and 2 with $\sigma = 0$, and*

$$m \geq \widetilde{\Omega}(r^2), \ t \geq \widetilde{\Omega}(\mu^2 r^3), \ and \ mt \geq \widetilde{\Omega}(\kappa \cdot \mu d r^2).$$

*Then AltMin-S (Algorithm 4), initialized at $U_{\text{init}}$ s.t. $\|(\mathbf{I} - U^*(U^*)^\top)U_{\text{init}}\|_F \leq \min(21/121, \widetilde{O}(1/\sqrt{\kappa}))$, and run for $K$ iterations outputs $U$ so that the following holds (w.p. $\geq 1 - K/(dr)^{10}$):*

$$\frac{\|(\mathbf{I} - U^*(U^*)^\top)U\|_F}{\sqrt{r}} \leq \widetilde{O}\left(\frac{\sqrt{\lambda_r^*/\lambda_1^*}}{\sqrt{r}2^K}\right). \tag{11}$$

**Remark 9 (Subset selection):** Note that when noise is very small $\sigma \ll O(\sqrt{\lambda_r^*}/\log t)$ or when the observations are noiseless ($\sigma = 0$), AltMin-S only needs $m \geq \widetilde{\Omega}(r^2)$ samples per task. Then, the per-task sample complexity does not grow with the number of tasks $t$. Again, we see that the sample complexity and iteration complexity of AltMin-S is smaller than AltMinGD-S. However, AltMinGD-S could still be faster than AltMin-S, due to its faster iterations.

Proofs of Theorems 5 & 6 are in the Appendices B.1 & D.1.

## A.3 Proof sketch for noiseless case

Here we provide proof sketches of Theorem 5. To highlight the main ideas behind our analysis, we start with the simplest case when there is no noise ($\sigma^2 = 0$) and all the task specific regression parameters lie on a single dimensional subspace ($r = 1$). The analysis gets quite challenging as we go to multi-dimensional shared subspace ($r > 1$), and we illustrate these challenges and how to resolve them in Section A.3.2.

### A.3.1 Proof sketch for the one-dimensional case

Let $u^* \in \mathbb{R}^d$ be the unit vector of the one-dimensional true subspace, and $v^* \in \mathbb{R}^t$ the vector of the true regression parameters of the $t$ tasks. In the noiseless setting ($\varepsilon_j^{(i)} = 0$), the $k$-th step of AltMin can be written as follows.

For all $i \in \mathcal{T}_k$
$$v^{(i)} \leftarrow (u^\top S_1^{(i)} u)^{-1} u^\top S_1^{(i)} (u^*) v^{*(i)},$$
$$\widehat{u} \leftarrow \left(\sum_{i \in \mathcal{T}_k} (v^{(i)})^2 S_2^{(i)}\right)^\dagger \left(\sum_{i \in \mathcal{T}_k} v^{*(i)} v^{(i)} S_2^{(i)} u^*\right), \ u^+ \leftarrow \frac{\widehat{u}}{\|\widehat{u}\|},$$

where $S_\ell^{(i)} = \frac{2}{m} \sum_{j=(\ell-1)m/2+1}^{\ell m/2} x_j^{(i)}(x_j^{(i)})^\top$ is the data covariance matrix of a half of the dataset $[m]$ of task $i \in [t]$. Our incoherence condition for rank-1 case simplifies to $\|v\|_\infty^2 \leq \frac{\mu}{t}\|v\|^2$. The distance between two unit norm vectors $u$ and $u^*$ is commonly measured by the angular distance defined as $\sin\theta(u, u^*) \triangleq \|(\mathbf{I} - u^*(u^*)^\top)u\|^{1/2}$, where $\mathbf{I} - u^*(u^*)^\top$ is the projection operator to the sub-space orthogonal to $u^*$. In the following we let $q \triangleq \langle u^*, u \rangle$ and use the relation $\sin\theta(u, u^*) = \|u - u^*q\|$ in the analysis. We use the fact that if we have a good previous iterate $u$ close to $u^*$, i.e. $\sin\theta(u, u^*) \leq 3/4$, then $1/2 \leq |q| \leq 1$.

Our analysis shows that we get geometrically closer to the true subspace $u^*$ at every iteration in this $\sin\theta$ distance, when initialized sufficiently close to $u^*$.

Our strategy is to show that the $v$-update achieves $|v^{(i)} - q^{-1}v^{*(i)}| \leq C\|v^{*(i)}\|\sin\theta(u, u^*)$ for some constant $C$, and the $u$-update achieves $\sin\theta(u^+, u^*) \leq (c/\|v^*\|)\|v - q^{-1}v^*\|)$ where the constant $c$ can be made as small as we want in the assumed sample regime. Together, they imply the desired theorem.

$v$**-update:** We can write $v^{(i)}q^{-1} - v^{*(i)}$ as

$$v^{(i)} - q^{-1}v^{*(i)} = u^\top S_1^{(i)}(qu^* - u)(u^\top S_1^{(i)} u)^{-1} q^{-1} v^{*(i)}.$$

In expectation, $\|\mathbb{E}[u^\top S_1^{(i)}(qu^* - u)]\| = \|u^\top(qu^* - u)\| = 1 - q^2 \leq (\sin\theta(u, u^*))^2$ and $\mathbb{E}[u^\top S_1^{(i)} u] = \|u\|^2 = 1$. Therefore, by Lemma B.2, if $\sin\theta(u, u^*) \leq \frac{1}{32}$ and there is enough samples per task, i.e. $m \geq \Omega(\log(t/K\,\delta))$, we can bound their deviations in terms of $\sin\theta(u, u^*)$. This implies that, with a probability of at least $1 - \delta/2$,

$$\frac{|v^{(i)} - q^{-1}v^{*(i)}|}{|v^{*(i)}|} \leq \frac{\sin\theta(u, u^*)}{4}, \text{ for all } i \in \mathcal{T}_k, \tag{12}$$

where we used the fact that $|q| \geq 1/2$. This in turn implies that $(1/4)|v^{*(i)}| \leq |v^{(i)}|$ and $v$ is incoherent.

$u$-**update:** We bound the distance between $\widehat{u}$ and $u^*$:

$$\widehat{u} - u^*q$$
$$= \Big( \underbrace{\sum_{i \in \mathcal{T}_k} \frac{(v^{(i)})^2}{\|v\|^2} S_2^{(i)}}_{:=A} \Big)^\dagger \Big( \underbrace{\sum_{i \in \mathcal{T}_k} \frac{v^{(i)}h^{(i)}}{\|v\|^2} S_2^{(i)}}_{:=\widehat{H}} u^*q \Big), \tag{13}$$

where $h^{(i)} = q^{-1}v^{*(i)} - v^{(i)}$. Notice that, in expectation, $\mathbb{E}[A] = \mathbf{I}$ and $\mathbb{E}[\widehat{H}u^*q] = \frac{v^\top h}{\|v\|^2}u^*q \leq \frac{\|h\|}{\|v\|}$. Therefore, by Lemma B.3, when there are enough samples, i.e. $mt \geq K\Omega(\mu d \log(\frac{1}{\delta}))$ deviations form these expected values can be bounded using the distance between $v$ and $v^*$, $\|h\|$. That is with a probability of at least $1 - \frac{\delta}{2}$, $A$ is invertible and well-conditioned,

$$A^{-1} = \mathbf{I} + E_1, \quad \text{and} \quad Hu^*q = \frac{v^\top h}{\|v\|^2}u^*q + e_2,$$

where $\|E_1\| \leq \frac{1}{16}$ and $\|e_2\| \leq \frac{1}{32}\Big(\frac{\|h\|}{\|v\|} + \sqrt{\frac{t}{\mu}}\frac{\|h\|_\infty}{\|v\|}\Big)$. Note that we had to critically use incoherence of intermediate $v$ to bound $e_2$. Therefore

$$\widehat{u} - u^*q = \underbrace{\frac{v^\top h}{\|v\|^2}u^*q}_{:=\widehat{u}_\parallel} + \underbrace{q\frac{v^\top h}{\|v\|^2}E_1 u^* + (\mathbf{I} + E_1)e_2}_{:=f}\,.$$

Notice that $\widehat{u}_\parallel$ is parallel to $u^*$. Rest of the terms are grouped together as $f$. The angle distance $\sin(u^+, u^*)$ only depends on the portion of $u^+$ which lie in the orthogonal subspace to $u^*$. Therefore, $\|\widehat{u}_\parallel\|$ does not directly contribute to the distance, and this is formalized below. Clearly, $\|(\mathbf{I} - u^*(u^*)^\top)u^+\| = \min_{q+}\|u^+ - u^*q^+\|$. This follows from the trivial solution of the scalar quadratic problem $\min_{q+\in\mathbb{R}}\|u - u^*q^+\|^2$. Thus,

$$\sin\theta(u^+, u^*) = \min_{q+}\|u^+ - u^*q^+\|$$
$$\leq \Big\|\frac{\widehat{u}}{\|\widehat{u}\|} - \Big(1 + \frac{h^\top v}{\|v\|^2}\Big)u^*\frac{q}{\|\widehat{u}\|}\Big\|$$
$$\leq \frac{\|f\|}{\|\widehat{u}\|} \leq \frac{\|f\|}{q\|u^*\| - \|f\| - \|h\|/\|v\|}\,. \tag{14}$$

**Putting them together:** We bound $f$ using definitions of $E_1$ and $e_2$, incoherence, and (12) as

$$\|f\| \leq \frac{1}{16}\frac{\|h\|}{\|v\|} + \frac{1}{32}\Big(\frac{\|h\|}{\|v\|} + \sqrt{\frac{t}{\mu}}\frac{\|h\|_\infty}{\|v\|}\Big) \leq \frac{1}{8}\sin\theta(u, u^*)\,.$$

Combining this with (14), we see that with a probability of at least $1 - \delta$, the angle distance geometrically decreases at each step, i.e.

$$\sin\theta(u^+, u^*) \leq \frac{1}{2}\sin\theta(u, u^*). \tag{15}$$

Finally, if the initialization is good, i.e. $\sin\theta(u_{\text{init}}, u^*) \leq \frac{1}{16}$, we can unroll the above inequality across iterations. Taking union bound over the iterations we get that, with a probability of at least $1 - K\delta$, the output $u$ after $K$ iterations satisfies

$$\sin\theta(u, u^*) \leq \frac{1}{2^K}\sin\theta(u_{\text{init}}, u^*). \tag{16}$$

To achieve this, we need at least $m \geq \Omega(\log(\frac{t}{K\delta}))$ samples per task and at least $mt \geq \Omega(K\mu d \log(\frac{1}{\delta}))$ total samples.

### A.3.2 Proof sketch for the $r$-dimensional case

Here we do not use $\sin\theta_1(U, u^*)$ distance, as the analysis of $\sin\theta_1$ gets more complicated in the general $r$-dimensional case. Therefore we use $\ell$-2 norm based error, $\Delta(U, U^*) := (\sum_{r'=1}^{r}\sin^2\theta_{r'}(U, U^*))^{1/2} := \|(\mathbf{I} - U^*(U^*)^\top)U\|_F$. Let $Q = (U^*)^\top U$, then $\Delta(U, U^*) = \|U - U^*Q\|_F$, and $1/2 \leq \|Q\| \leq 1$ if $\Delta(U, U^*) \leq 3/4$.

For all $i \in \mathcal{T}_k$

$$v^{(i)} \leftarrow (U^\top S_1^{(i)} U)^\dagger U^\top S_1^{(i)} U^* v^{*(i)} ,$$

$$\widehat{U} \leftarrow \Big(\mathcal{A}^\dagger\Big(\sum_{i \in \mathcal{T}_k} S_2^{(i)} U^* v^{*(i)} (v^{(i)} W^{-\frac{1}{2}})^\top\Big)\Big)W^{-\frac{1}{2}} ,$$

$$U \leftarrow \text{QR}(\widehat{U}) ,$$

where $W = V^\top V$, $\mathcal{A} : \mathbb{R}^{d \times r} \to \mathbb{R}^{d \times r}$ is linear operator such that $\mathcal{A}(U) = \sum_{i \in \mathcal{T}_k} S_2^{(i)} UW^{-\frac{1}{2}}v^{(i)}(v^{(i)})^\top W^{-\frac{1}{2}}$, and $S_\ell^{(i)}$ are defined as in the one-dimensional case.

$V$**-update:** We will prove that $\|v^{(i)} - Q^{-1}v^{*(i)}\| = O(\Delta(U, U^*))$. Let $h^{(i)} := v^{(i)}Q^{-1} - v^{*(i)}$, then

$$h^{(i)} = (U^\top S_1^{(i)} U)^\dagger \underbrace{U^\top S_1^{(i)}(U^*Q - U)Q^\dagger v^{*(i)}}_{:=G}.$$

Notice that, in expectation, $\|\mathbb{E}[U^\top S_1^{(i)} U]\| = 1$ and $\|\mathbb{E}[G]\| = \|U^\top(U^*Q - U)\| = \|Q^\top Q - \mathbf{I}\| = \Delta^2(U, U^*)$. Therefore, by Lemma B.2, if $\Delta^2(U, U^*) \leq \frac{1}{32}$ and there is enough samples per task, i.e. $m \geq \Omega(r\log(\frac{t}{K\delta}))$, we can bound their deviations in terms of $\sin\theta(u, u^*)$. This implies that, with a probability of at least $1 - \delta/2$,

$$\|h^{(i)}\| \leq \frac{\|v^{*(i)}\|\Delta^2(U, U^*)}{4} , \text{ for all } i \in \mathcal{T}_k. \tag{17}$$

Furthermore, $\|v^{(i)}\| \leq 4\|v^{*(i)}\|$ and $V$ is incoherent.

$U$**-update:** We bound the distance between $\widehat{U}$ and $U^*$:

$$(\widehat{U} - U^*Q)W^{\frac{1}{2}} = \mathcal{A}^\dagger\Big(\underbrace{\sum_{i \in \mathcal{T}_k} S_2^{(i)} U^* Q h^{(i)}(v^{(i)})^\top W^{-\frac{1}{2}}}_{:=-\widehat{\mathcal{H}}(U^*Q)}\Big).$$

Notice that, in expectation, $\mathbb{E}[\widehat{\mathcal{H}}(U^*Q)] = \mathcal{H}(U^*Q) := U^*Q\sum_{i \in \mathcal{T}_k} h^{(i)}(v^{(i)})^\top W^{-\frac{1}{2}}$ and $\mathcal{H}(U^*Q) \leq \|H\|_F$ and $\mathbb{E}[\mathcal{A}]$ is the identity map $\mathcal{I}$. Like in the 1-dimensional case, by Lemma B.3, when there are enough samples, i.e. $mt \geq K\Omega(\mu dr^2\log(\frac{1}{\delta}))$ deviations from these expected values can be bounded using the distance between $V$ and $V^*$, $\|H\|$. That is, with a probability of at least $1 - \delta/2$, $\mathcal{A}$ is invertible and well-conditioned in Frobenius operator norm,

$$\mathcal{A}^{-1} = \mathcal{I} + \mathcal{E}_1, \text{ and } \widehat{\mathcal{H}}(U^*Q) = \mathcal{H}(U^*Q) - E_2,$$

where $\|\mathcal{E}_1\|_F \leq 1/16$ and $\|E_2\|_F \leq 1/32(\|H\|_F + \sqrt{t/\mu}\|H\|_{\infty,2})$. Note that we had to critically use incoherence of intermediate $V$ to bound $E_2$. Therefore,

$$(\widehat{U} - U^*Q)W^{\frac{1}{2}} = -\mathcal{H}(U^*Q) - \underbrace{cE_1\mathcal{H}(U^*Q) + (\mathcal{I} + \mathcal{E}_1)E_2}_{:=F} .$$

Now, using similar arguments as in the one-dimensional case, we get

$$\Delta(U^+, U^*) \le \left\| \widehat{U}R^{-1} - U^*Q + \mathcal{H}(U^*Q) \right\|_F \|W^{-\frac{1}{2}}\|$$

$$\le \frac{\|F\|_F}{\|R^{-1}\|} \le \frac{\|F\|_F \lambda_r^{-\frac{1}{2}}}{\|QU^*\| - (\|F\|_F + \|H\|_F)\lambda_r^{-\frac{1}{2}}} \, .$$

**Putting them together:** Using similar arguments as in one-dimensional case, if the initialization is good, i.e. $\Delta(U_{\text{init}}, U^*) \le 1/16$, we can show that with a probability of at least $1 - \delta$, the next iterate $U^+$ satisfies: $\Delta(U^+, U^*) \le \frac{1}{2}\Delta(U, U^*)$. To achieve this, we need at least $\Omega(r \log(\frac{t}{K\delta}))$ samples per task ($m$) and at least $\Omega(K\mu dr^2 \log(\frac{1}{\delta}))$ total samples ($mt$). Result now follows by applying the above result $K$ times.

## B  Analysis of AltMin (Algorithm 3)

Initialized at $U$, the $k$-the step of alternating minimization-based AltMin (Algorithm 3) is:

$$v^{(i)} \leftarrow (U^\top S_1^{(i)}U)^\dagger((U^\top S_1^{(i)}U^*)v^{*(i)} + U^\top z^{(i)}), \quad \text{for } i \in \mathcal{T}_k = [1 + \frac{(k-1)t}{K}, \frac{tk}{K}] \ (18)$$

$$\widehat{U} \leftarrow \mathcal{A}^\dagger \Big( \sum_{i \in [t]} S_2^{(i)}U^*v^{*(i)}(v^{(i)})^\top + z^{(i)}(v^{(i)})^\top \Big), \tag{19}$$

$$U^+ \leftarrow \text{QR}(\widehat{U}), \tag{20}$$

where $U^+$ is the next iterate, $S_1^{(i)} = \frac{2}{m}\sum_{j \in [1, m/2]} x_j^{(i)}(x_j^{(i)})^\top$, $S_2^{(i)} = \frac{2}{m}\sum_{j \in [1+m/2, m]} x_j^{(i)}(x_j^{(i)})^\top$, $z^{(i)} \triangleq (1/m)\sum_{j \in [m]} \varepsilon_j^{(i)} x_j^{(i)}$ and $\mathcal{A} : \mathbb{R}^{d \times r} \to \mathbb{R}^{d \times r}$ is a self-adjoint linear operator such that $\mathcal{A}(U) = \sum_{i \in T} S^{(i)}Uv^{(i)}(v^{(i)})^\top$. The self-adjointness of $\mathcal{A}$ follows from the symmetry of $S^{(i)}$ when using cyclic property of trace as follows

$$\langle U_2, \mathcal{A}(U_1) \rangle = \sum_{i \in T} \left\langle U_2, S^{(i)}U_1 v^{(i)}(v^{(i)})^\top \right\rangle = \sum_{i \in T} \text{tr}(U_2^\top S^{(i)}U_1 v^{(i)}(v^{(i)})^\top)$$

$$= \sum_{i \in T} \text{tr}(v^{(i)}(v^{(i)})^\top U_2^\top S^{(i)}U_1) = \langle \mathcal{A}(U_2), U_1 \rangle$$

$$\tag{21}$$

**Incoherence.** $\max_i \|v^{*(i)}\|^2 \le (\mu r/t)\lambda_r(\sum_{i \in [t]} v^{*(i)}(v^{*(i)})^\top)$, and we define $\nu = (1/t)\lambda_r(\sum_{i \in [t]} v^{*(i)}(v^{*(i)})^\top)$. Notice that, this non-standard definition of incoherence is related to the standard definition: $W^* = (V^*)^\top V^* = \sum_{i \in [t]} v^{*(i)}(v^{*(i)})^\top$, $V^* = \tilde{V}^* R^*$ (QR-decomposition), $\max_i \|\tilde{v}^{*(i)}\|^2 \le \tilde{\mu} r/t$, as follows $\mu = \widehat{\mu}(\sigma_1^2(R^*)/\sigma_r^2(R^*))$.

**Theorem 8** (Formal version of Theorem 5 in Appendix A). *Let there be $t$ linear regression tasks, each with $m$ samples satisfying Assumptions 1 and 2, and $K = \lceil \log_2(\frac{mt}{(\lambda_1^*/\lambda_r^*)(\sigma/\sqrt{\lambda_r^*})\mu dr}) \rceil$, $\|(\mathbf{I} - U^*(U^*)^\top)U_{\text{init}}\|_F \le \min\left(\frac{21}{121}, O\left(\sqrt{\frac{\lambda_r^*}{\lambda_1^*}\frac{1}{\log(t/K)}}\right)\right)$, $m \ge \Omega\left((1 + r(\frac{\sigma}{\sqrt{\lambda_r^*}})^2)r\log(\frac{t}{\delta}) + r^2\log(\frac{K}{\delta})\right)$, $t \ge \Omega(\mu^2 r^3 K \log(\frac{K}{\delta}))$, and $mt \ge \Omega\left(\mu dr^2 K \frac{\lambda_1^*}{\lambda_r^*}\left(\log(\frac{t}{\delta}) + (\frac{\sigma}{\sqrt{\lambda_r^*}})^2 \log^2(\frac{t}{\delta})\log(\frac{rK}{\delta})\right)\right)$. Then, for any $0 < \delta < 1$, after $K$ iterations, AltMin (Algorithm 3) returns an orthonormal matrix $U \in \mathbb{R}^{d \times r}$, such that with a probability of at least $1 - \delta$*

$$\frac{1}{\sqrt{r}}\|(\mathbf{I} - U^*(U^*)^\top)U\|_F \le O\left(\frac{\sigma}{\sqrt{\lambda_r^*}}\sqrt{\frac{\mu dr K \log(\frac{t}{\delta})\log(\frac{rK}{\delta})}{mt}}\right) \tag{22}$$

*and the algorithm uses an additional memory of size $O(d^2 r^2)$.*

A proof is in Section B.1.

**Initialization.** If we initialize AltMin (Algorithm 3) with Method-of-Moments (Theorem 12), we need at least

$$mt \geq \widetilde{\Omega}\Big(\frac{\lambda_1^{*2}}{\lambda_r^{*2}}\mu dr^2 + \Big(\frac{\sigma}{\sqrt{\lambda_r^*}}\Big)^4 \frac{\lambda_1^*}{\lambda_r^*}dr^3\Big) \tag{23}$$

initial number of samples, where $\widetilde{\Omega}$ hides polylog factors.

## B.1 Proof of Theorem 8 (formal version of Theorem 5 in Appendix A)

**Proof sketch:** We first prove that distance between $U^*$ and $U$ decreases at each iteration up to some additional noise terms. Then this per iterate result is unrolled to obtained the final guarantees.

First we focus on the $k$-th iterate. In this analysis, unless specified $[t]$, represents the $k$-th $K$-way partition used for the $k$-th iterate. In the following lemma we prove that tasks subset used for each iteration, satisfy approximate incoherence.

**Lemma B.1** (Shuffling and partition of tasks). *Let $\mathcal{T}_k$ be the $k$-th subset ($k \in [K]$) of the $K$-way partition of the shuffled set of all $t$ tasks. If $t \geq \Omega(\mu^2 r^3 K \log(1/\delta))$, then with a probability of at least $1 - \delta/3$,*

$$\lambda_1\Big(\sum_{i\in\mathcal{T}_k} v^{*(i)}(v^{*(i)})^\top\Big) = \frac{1}{K}\Theta(\lambda_1((V^*)^\top V^*)) \quad and \tag{24}$$

$$\lambda_r\Big(\sum_{i\in\mathcal{T}_k} v^{*(i)}(v^{*(i)})^\top\Big) = \frac{1}{K}\Theta(\lambda_r((V^*)^\top V^*)) , \quad for\ all\ r' \in [r] \tag{25}$$

*where are $\lambda_1(\cdot)$ and $\lambda_r(\cdot)$ are the largest and smallest, respectively, eigenvalue operators of real-symmetric $r \times r$ matrix.*

A proof is in Section B.5.

In the analysis of an iterate we denote the current iterate using $U$ and the next iterate using $U^+$. First we prove that the distance between the true $v^{*(i)}$ and the current $v^{(i)}$ is approximately upper-bounded by multiple of distance between $U$ and $U^*$. Next we prove that distance between $U^+$ and $U^*$ is approximately a fraction of the distance between $v^{*(i)}$ and $v^{(i)}$. Finally, combining the above two results gives us desired result.

**Preliminaries:** Let $Q = (U^*)^\top U$. Using Lemma F.4, if $\|U - U^*(U^*)^\top U\|_F < 1$, $Q$ is invertible. Let $Q^{-1}$ be the right inverse of $Q$, i.e. $QQ^{-1} = \mathbf{I}$. Let $W = (V^*)^\top V^* = \sum_{i\in[t]} v^{*(i)}(v^{*(i)})^\top$, then using Assumption 2 we have that $\lambda_1^* = (r/t)\max_{\|z\|=1} z^\top W^* z$ and $\lambda_r^* = (r/t)\min_{\|z\|=1} z^\top W^* z$.

**Update on $V$:** Let $h^{(i)} = v^{(i)} - Q^{-1}v^{*(i)}$ and $H^T = [h^{(1)}h^{(2)}\ldots h^{(t)}]$. Let $\|H\|_F \triangleq \sqrt{\sum_{i\in[t]}\|h^{(i)}\|^2}$ and $\|H\|_{\infty,2} \triangleq \max_{i\in[t]}\|h^{(i)}\|$. Let $W = V^\top V = \sum_{i\in[t]} v^{(i)}(v^{(i)})^\top$, and $\lambda_1 = (r/t)\max_{\|z\|=1} z^\top W z$ and $\lambda_r = (r/t)\min_{\|z\|=1} z^\top W z$.

**Lemma B.2.** *Assume that all conditions and the large probability event in Lemma B.1 holds true. If $\|(\mathbf{I} - U^*(U^*)^\top)U\|_F \leq \min\Big(\frac{21}{121}, O\Big(\sqrt{\frac{\lambda_r^*}{\lambda_1^*}\frac{1}{\log(t/K)}}\Big)\Big)$ and $m \geq \Omega\Big(\Big(\frac{\sigma}{\sqrt{\lambda_r^*}}\Big)^2 r^2 \log(\frac{t}{K\delta}) + r\log(\frac{t}{K\delta})\Big)$, then with a probability of at least $1 - \delta/3$,*

$$\|v^{(i)}\| \leq O\Big(\mu\,\lambda_r\Big), \ \ \lambda_1 \leq 2\lambda_1^* , \ and \ \lambda_r^*/2 \leq \lambda_r \leq 2\lambda_r^* \tag{26}$$

*and*

$$\sqrt{\frac{rK}{t}}\frac{\|H\|_F}{\sqrt{\lambda_r}} \leq O\Big(\sqrt{\frac{\log(\frac{t}{K\delta})}{\log(\frac{1}{\delta})}}\sqrt{\frac{\lambda_1^*}{\lambda_r^*}}\|(\mathbf{I}-U^*(U^*)^\top)U\|_F + \frac{\sigma}{\sqrt{\lambda_r^*}}\sqrt{\frac{r^2\log(\frac{t}{K\delta})}{m}}\Big) \tag{27}$$

$$\sqrt{\frac{rK}{t}}\frac{\|H\|_{\infty,2}}{\sqrt{\lambda_r}} \leq O\Big(\sqrt{\frac{\log(\frac{t}{K\delta})}{\log(\frac{1}{\delta})}}\|(\mathbf{I}-U^*(U^*)^\top)U\|\sqrt{\frac{\mu rK}{t}} + \frac{\sigma}{\sqrt{\lambda_r^*}}\sqrt{\frac{r^2K\log(\frac{t}{K\delta})}{mt}}\Big) \tag{28}$$

A proof is in Section B.2.1.

**Update on $U$:** Let $\mathcal{W}, \mathcal{H}, \widehat{\mathcal{H}} : \mathbb{R}^{d\times r} \to \mathbb{R}^{d\times r}$ be three linear operators, such that $\mathcal{W}(U) = U\sum_{i\in\mathcal{T}_k}v^{(i)}(v^{(i)})^\top = UW, \mathcal{H}(U) = U\sum_{i\in\mathcal{T}_k}h^{(i)}(v^{(i)})^\top$ and $\widehat{\mathcal{H}}(U) = \sum_{i\in\mathcal{T}_k}S_2^{(i)}Uh^{(i)}(v^{(i)})^\top$, where $h^{(i)} = v^{(i)} - Q^{-1}v^{*(i)}$. $\mathcal{W}$ is invertible and self-adjoint. Therefore $\mathcal{W}^{-\frac{1}{2}}$ and $\mathcal{W}^{\frac{1}{2}}$ exist. Let $\mathcal{I} : \mathbb{R}^{d\times r} \to \mathbb{R}^{d\times r}$ be the identity mapping, such that $\mathcal{I}(U) = U$.

$$\widehat{U} - U^*Q = \mathcal{A}^\dagger\Big(\sum_{i\in\mathcal{T}_k}S_2^{(i)}U^*Q(Q^{-1}v^{*(i)} - v^{(i)})(v^{(i)})^\top + z^{(i)}(v^{(i)})^\top\Big) \tag{29}$$

$$= \mathcal{A}^\dagger\Big(-\widehat{\mathcal{H}}(U^*Q) + \sum_{i\in\mathcal{T}_k}z^{(i)}(v^{(i)})^\top\Big) \tag{30}$$

$$= \mathcal{W}^{-\frac{1}{2}}(\mathcal{W}^{\frac{1}{2}}\mathcal{A}^\dagger\mathcal{W}^{\frac{1}{2}})\mathcal{W}^{-\frac{1}{2}}\Big(-\widehat{\mathcal{H}}(U^*Q) + \sum_{i\in\mathcal{T}_k}z^{(i)}(v^{(i)})^\top\Big) \tag{31}$$

$$= \mathcal{W}^{-\frac{1}{2}}(\mathcal{I}+\mathcal{E}_1)\Big(-(\mathcal{W}^{-\frac{1}{2}}\mathcal{H}+\mathcal{E}_2)(U^*Q) + \mathcal{W}^{-\frac{1}{2}}\Big(\sum_{i\in\mathcal{T}_k}z^{(i)}(v^{(i)})^\top\Big)\Big) \tag{32}$$

where $\mathcal{E}_1 = (\mathcal{W}^{-\frac{1}{2}}\mathcal{A}\mathcal{W}^{-\frac{1}{2}})^\dagger - \mathcal{I}$ and $\mathcal{E}_2 = \mathcal{W}^{-\frac{1}{2}}\widehat{\mathcal{H}} - \mathcal{W}^{-\frac{1}{2}}\mathcal{H}$, and $F = \widehat{U} - U^*Q + \mathcal{W}^{-1}(\mathcal{H}(U^*Q))$. Let $F = \widehat{U} - U^*Q + \mathcal{W}^{-1}(\mathcal{H}(U^*Q))$

**Lemma B.3.** *Assume that all conditions and the large probability event in Lemma B.2 holds true. Then,*

$$\|\mathcal{W}^{-1}\mathcal{H}(U^*Q)\|_F \leq O\Big(\sqrt{\frac{\lambda_1^*}{\lambda_r^*}\log(\frac{t}{K})}\|(\mathbf{I}-U^*(U^*)^\top)U\|_F + \frac{\sigma}{\sqrt{\lambda_r^*}}\sqrt{\frac{r^2\log(\frac{t}{K\delta})}{m}}\Big) \tag{33}$$

*and if $mt \geq \Omega(\mu dr^2K\log(t/K\delta))$, then with probability at least $1-\delta/3$*

$$\|F\|_F \leq O\Big(\sqrt{\frac{\lambda_1^*}{\lambda_r^*}\frac{\mu dr^2K\log(\frac{t}{K\delta})}{mt}}\|(\mathbf{I}-U^*(U^*)^\top)U\|_F + \frac{\sigma}{\sqrt{\lambda_r^*}}\sqrt{\frac{\mu dr^2K\log(\frac{t}{K\delta})\log(\frac{r}{\delta})}{mt}}\Big) \tag{34}$$

A proof is in Section B.3.1.

**Lemma B.4.** *If $\frac{1}{2} \leq \sigma_{\min}(Q)$, $\|F\|_F \leq \frac{1}{8}$ and $\|\mathcal{W}^{-1}(\mathcal{H}(U^*Q))\|_F \leq \frac{1}{8}$, then $R$ is invertible and $\|R^{-1}\| \leq 4$.*

A proof is in Section B.4. Clearly, from (33) and (34), a sufficient condition for the above lemma is

$$O\Big(\sqrt{\frac{\lambda_1^*}{\lambda_r^*}\log(\frac{t}{K})}\|(\mathbf{I}-U^*(U^*)^\top)U\|_F + \frac{\sigma}{\sqrt{\lambda_r^*}}\sqrt{\frac{r^2\log(\frac{t}{K\delta})}{m}}\Big) \leq \frac{1}{8} \text{ , and} \tag{35}$$

$$O\Big(\sqrt{\frac{\lambda_1^*}{\lambda_r^*}\frac{\mu dr^2K\log(\frac{t}{K\delta})}{mt}}\|(\mathbf{I}-U^*(U^*)^\top)U\|_F + \frac{\sigma}{\sqrt{\lambda_r^*}}\sqrt{\frac{\mu dr^2K\log(\frac{t}{K\delta})\log(\frac{r}{\delta})}{mt}}\Big) \leq \frac{1}{8} \tag{36}$$

which can be satisfied with

$$\|(\mathbf{I}-U^*(U^*)^\top)U\|_F \leq O\Big(\sqrt{\frac{\lambda_r^*}{\lambda_1^*}\frac{1}{\log(t/K)}}\Big), \quad m \geq \Omega\Big(\Big(\frac{\sigma}{\sqrt{\lambda_r^*}}\Big)^2r^2\log(\frac{t}{K\delta}) + r^2\log(\frac{1}{\delta})\Big) \text{ , and} \tag{37}$$

$$mt \geq \Omega\Big(\mu dr^2K\Big(1 + \Big(\frac{\sigma}{\sqrt{\lambda_r^*}}\Big)^2\log(\frac{t}{K\delta})\log(\frac{r}{\delta})\Big)\Big) \tag{38}$$

Finally, we bound the Frobenius norm distance of the next iterate $U^+$ from the optimal $U^*$.

$$\|(\mathbf{I}-U^*(U^*)^\top)U^+\|_F \tag{39}$$

$$= \min_{Q^+} \|U^+ - U^*Q^+\|_F \tag{40}$$

$$\leq \|\widehat{U}R^{-1} - U^*QR^{-1} + (\mathcal{W}^{-1}\mathcal{H}(U^*Q))R^{-1}\| \tag{41}$$

$$\leq \|\widehat{U} - U^*Q + \mathcal{W}^{-1}\mathcal{H}(U^*Q)\|_F\|R^{-1}\| \tag{42}$$

$$= \|F\|_F\|R^{-1}\| \tag{43}$$

$$\leq O\Big(\sqrt{\frac{\lambda_1^*}{\lambda_r^*}\frac{\mu dr^2 K\log(\frac{t}{K\delta})}{mt}}\|(\mathbf{I}-U^*(U^*)^\top)U\|_F + \frac{\sigma}{\sqrt{\lambda_r^*}}\sqrt{\frac{\mu dr^2 K\log(\frac{t}{K\delta})\log(\frac{r}{\delta})}{mt}}\Big) \tag{44}$$

If

$$mt \geq \Omega\Big(\mu dr^2 K\frac{\lambda_1^*}{\lambda_r^*}\Big(\log(\frac{t}{K\delta}) + \Big(\frac{\sigma}{\sqrt{\lambda_r^*}}\Big)^2\log^2(\frac{t}{K\delta})\log(\frac{r}{\delta})\Big)\Big) \tag{45}$$

then,

$$\|(\mathbf{I}-U^*(U^*)^\top)U^+\|_F \leq \frac{1}{2}\|(\mathbf{I}-U^*(U^*)^\top)U\|_F + \frac{1}{2}\min\Big(\frac{21}{121}, O\Big(\sqrt{\frac{\lambda_r^*}{\lambda_1^*}\frac{1}{\log(t/K)}}\Big)\Big) \tag{46}$$

Thus if $\|(\mathbf{I} - U^*(U^*)^\top)U\|_F \leq \min\Big(\frac{21}{121}, O\Big(\sqrt{\frac{\lambda_r^*}{\lambda_1^*}\frac{1}{\log(t/K)}}\Big)\Big)$, then $\|(\mathbf{I} - U^*(U^*)^\top)U^+\|_F \leq \min\Big(\frac{21}{121}, O\Big(\sqrt{\frac{\lambda_r^*}{\lambda_1^*}\frac{1}{\log(t/K)}}\Big)\Big)$.

Therefore, using union-bound, we can un-roll the relation, between current iterate $U$ and the next iterate $U^+$, over $K$ iterations, starting from $U_{\text{init}}$ and ending at some $U$ iterations, to get

$$\|(\mathbf{I} - U^*(U^*)^\top)U\|_F \leq \frac{1}{2^K}\|(\mathbf{I} - U^*(U^*)^\top)U_{\text{init}}\|_F + O\Big(\frac{\sigma}{\sqrt{\lambda_r^*}}\sqrt{\frac{\mu dr^2 K\log(\frac{t}{K\delta})\log(\frac{r}{\delta})}{mt}}\Big) \tag{47}$$

with probability at least $1 - K\delta$. Finally setting $K = \lceil\log_2(\frac{mt}{(\lambda_1^*/\lambda_r^*)(\sigma/\sqrt{\lambda_r^*})\mu dr})\rceil$ we get that, with a probability of at least $1 - K\delta$

$$\|(\mathbf{I} - U^*(U^*)^\top)U\|_F \leq O\Big(\frac{\sigma}{\sqrt{\lambda_r^*}}\sqrt{\frac{\mu dr^2 K\log(\frac{t}{K\delta})\log(\frac{r}{\delta})}{mt}}\Big) \tag{48}$$

## B.2 Analysis of update on $V$

### B.2.1 Proof of Lemma B.2

*Proof of Lemma B.2.* In this proof for brevity, we will first set that $\mathcal{T}_k \leftarrow [t]$, $|\mathcal{T}_k| = t/K \leftarrow t$, $S_1^{(i)} \leftarrow S^{(i)} = \frac{1}{m}\sum_{j\in[m]} x_j^{(i)}(x_j^{(i)})^\top$. This can be done due to the approximate equivalence of the subset $\mathcal{T}_k$ and the set of all tasks $[t]$ by Lemma B.1 which requires that $t \geq \Omega(\mu^2 r^3 K\log(\frac{K}{\delta}))$. Finally at the end of the analysis we will reset $\mathcal{T}_k \leftarrow \mathcal{T}_k$, $|\mathcal{T}_k| = t/K \leftarrow t/K$, $S_1^{(i)} \leftarrow S_1^{(i)} = \frac{2}{m}\sum_{j\in[1,m/2]} x_j^{(i)}(x_j^{(i)})^\top$.

Recall the definition of $v^{(i)}$ from the update (18), and that $Q^{-1}$ is right inverse of $Q$, i.e. $QQ^{-1} = \mathbf{I}$.

$$v^{(i)} - Q^{-1}v^{*(i)} = (U^\top S^{(i)}U)^\dagger(U^\top S^{(i)}(U^*Q - U))Q^{-1}v^{*(i)} + (U^\top S^{(i)}U)^\dagger U^\top z^{(i)} \tag{49}$$

We can use re-write the first term as,

$$(U^\top S^{(i)} U)^\dagger U^\top S^{(i)} (U^* Q - U) Q^{-1} \tag{50}$$

$$= (U^\top S^{(i)} U)^\dagger U^\top S^{(i)} (U U^\top + U_\perp U_\perp^\top)(U^* Q - U) Q^{-1} \tag{51}$$

$$= U^\top (U^* Q - U) Q^{-1} + (U^\top S^{(i)} U)^\dagger U^\top S^{(i)} U_\perp U_\perp^\top (U^* Q - U) Q^{-1} \tag{52}$$

$$= -U^\top (\mathbf{I} - U^*(U^*)^\top)^2 U Q^{-1} + (U^\top S^{(i)} U)^\dagger U^\top S^{(i)} U_\perp U_\perp^\top U^* \tag{53}$$

$$= -(U - U^* Q)^\top (U - U^* Q) Q^{-1} + (U^\top S^{(i)} U)^\dagger U^\top S^{(i)} U_\perp U_\perp^\top U^* \tag{54}$$

where we used the fact that $Q = (U^*)^\top U$. Therefore

$$\|v^{(i)} - Q^{-1} v^{*(i)}\| \le$$
$$\|U - U^* Q\|\|(U - U^* Q) Q^{-1} v^{*(i)}\| + \|(U^\top S^{(i)} U)^\dagger\|(\|U^\top S^{(i)} U_\perp U_\perp^\top U^* v^{*(i)}\| + \|U^\top z^{(i)}\|) \tag{55}$$

If $m \ge \Omega(r \log(t/\delta))$, then $\alpha = c\sqrt{\frac{r \log(27t/\delta)}{m}} \le 1/2$ and by Lemma B.5, with a probability of at least $1 - \delta$,

$$\left.\begin{aligned}
\|(U^\top S^{(i)} U)^\dagger\| &\le (1 + 2\alpha), \\
\|U^\top S^{(i)} U_\perp U_\perp^\top U^* v^{*(i)}\| &\le \alpha \|U_\perp^\top U^* v^{*(i)}\|, \quad \text{and} \\
\|U^\top z^{(i)}\| &\le \sigma \alpha,
\end{aligned}\right\} \quad \text{for all } i \in [t] \tag{56}$$

Now if $m \ge \Omega(r \log(1/\delta))$ and $\|U^* Q - U\| \le O\left(\sqrt{\frac{\log(\frac{t}{\delta})}{\log(\frac{1}{\delta})}}\right)$, then

$$\|v^{(i)} - Q^{-1} v^{*(i)} \le O\left(\sqrt{\frac{\log(\frac{t}{\delta})}{\log(\frac{1}{\delta})}}(\|(U^* Q - U) Q^{-1} v^{*(i)}\| + \|U_\perp^\top U^* v^{*(i)}\|) + \sigma \sqrt{\frac{r \log(\frac{t}{\delta})}{m}}\right) \tag{57}$$

Next we bound $\|H\|_F$, which by definition is $\|H\|_F = \sqrt{\sum_{i \in [t]} \|h^{(i)}\|^2} = \sqrt{\sum_{i \in [t]} \|v^{(i)} - Q^{-1} v^{*(i)}\|^2}$. Using (57) and the fact that $(a^2 + b^2) \le 2(a^2 + b^2)$ we get

$$\|H\|_F^2 \le \frac{\log(\frac{t}{\delta})}{\log(\frac{1}{\delta})}\left[\sum_{i \in \mathcal{T}} O(\|(U^* Q - U) Q^{-1} v^{*(i)}\|^2 + \|U_\perp^\top U^* v^{*(i)}\|^2)\right] + t(\sigma \sqrt{\frac{r \log(\frac{t}{\delta})}{m}})^2) \tag{58}$$

Clearly $\|Q\| = \|(U^*)^\top U\| \le \|U^*\|\|U\| \le 1$. If $\|(\mathbf{I} - U^*(U^*)^\top)U\| \le \|(\mathbf{I} - U^*(U^*)^\top)U\|_F \le \frac{3}{4}$, then by using Lemma F.4, $\|Q^{-1}\| \le 2$.

$$\sum_{i \in [t]} \|(U^* Q - U) Q^{-1} v^{*(i)}\|^2 = \sum_{i \in [t]} \text{tr}((v^{*(i)})^\top ((U^* Q - U) Q^{-1})^\top (U^* Q - U) Q^{-1} v^{*(i)}) \tag{59}$$

$$= \text{tr}((U^* Q - U) Q^{-1})^\top (U^* Q - U) Q^{-1}) \sum_{i \in [t]} v^{*(i)}(v^{*(i)})^\top) \tag{60}$$

$$\le \|(U^* Q - U)\|_F^2 \|Q^{-1}\|^2 O(\lambda_1^*)(t/r) \tag{61}$$

$$\le 4\|(U^* Q - U)\|_F^2 O(\lambda_1^*)(t/r) \tag{62}$$

Similarly we can use Lemma F.4, to get

$$\sum_{i \in [t]} \|U_\perp^\top U^* v^{*(i)}\|^2 = \sum_{i \in [t]} \text{tr}((v^{*(i)})^\top (U_\perp^\top U^*)^\top U_\perp^\top U^* v^{*(i)}) \tag{63}$$

$$= \text{tr}((U_\perp^\top U^*)^\top (U_\perp^\top U^*) \sum_{i \in [t]} v^{*(i)}(v^{*(i)})^\top) \tag{64}$$

$$\le \|U_\perp^\top U^*\|_F^2 O(\lambda_1^*)(t/r) \tag{65}$$

$$\le \|(U^* Q - U)\|_F^2 O(\lambda_1^*)(t/r) \tag{66}$$

Therefore substituting the above two inequalities into (58) and using the fact that $\sqrt{a+b} \leq \sqrt{a}+\sqrt{b}$ for all $0 \leq a,b$ we get

$$\|H\|_F \leq O\left(\sqrt{\frac{\log(\frac{t}{\delta})}{\log(\frac{1}{\delta})}}\|U^*Q - U\|_F\sqrt{\lambda_1^*(t/r)} + \sqrt{t}\sigma\sqrt{\frac{r\log(\frac{t}{\delta})}{m}}\right) \tag{67}$$

Then as $\|(\mathbf{I} - U^*(U^*)^\top)U\|_F \leq O\left(\sqrt{\frac{\lambda_r^*}{\lambda_1^*}\frac{1}{\log(t)}}\right)$ and $m \geq \Omega\left(\left(\frac{\sigma}{\sqrt{\lambda_r^*}}\right)^2 r^2\log(\frac{t}{\delta})\right)$, $\|H\|_F \leq (1 - \frac{1}{\sqrt{2}})\sqrt{(t/r)\lambda_r^*}$. Using $\|Q^{-1}\| \leq 2$ in (57) we also get that

$$\|h^{(i)}\| = \|v^{(i)} - Q^{-1}v^{*(i)}\| \leq O\left(\sqrt{\frac{\log(\frac{t}{\delta})}{\log(\frac{1}{\delta})}}\|(U^*Q - U)\|\|v^{*(i)}\| + \sigma\sqrt{\frac{r\log(\frac{t}{\delta})}{m}}\right) \tag{68}$$

By definition is $\|H\|_{\infty,2} = \max_{i\in[t]}\|h^{(i)}\| = \max_{i\in[t]}\|v^{(i)} - Q^{-1}v^{*(i)}\|$. Then as $\|(\mathbf{I} - U^*(U^*)^\top)U\| \leq \|(\mathbf{I} - U^*(U^*)^\top)U\|_F \leq O\left(\sqrt{\frac{\lambda_r^*}{\lambda_1^*}\frac{1}{\log(t)}}\right) \leq O(1), m \geq \Omega\left(\left(\frac{\sigma}{\sqrt{\lambda_r^*}}\right)^2 r^2\log(\frac{t}{\delta})\right) \geq \Omega\left(\left(\frac{\sigma}{\sqrt{\lambda_r^*}}\right)^2 r\log(\frac{t}{\delta})\right)$, $\|H\|_{\infty,2} \leq O(\mu\lambda_r^*)$. Now, using $\|H\|_F \leq (1 - \frac{1}{\sqrt{2}})\sqrt{(t/r)\lambda_r^*}$, $\|H\|_{\infty,2} \leq O(\mu\lambda_r^*)$, $\|Q\| \leq 1$ and $\frac{10}{11} \leq \sigma_{\min}(Q)$, by Lemma B.6, we get the approximate incoherence relation for the intermediate $V$

$$\|v^{(i)}\| \leq O\left(\mu\lambda_r\right), \ \lambda_1 \leq 2\lambda_1^*, \text{ and } \lambda_r^* \leq 2\lambda_r \tag{69}$$

Using this we bound $\|H\|_{\infty,2}$. Using the above incoherence relation and (68), we get

$$\sqrt{\frac{r}{t}}\frac{\|H\|_{\infty,2}}{\sqrt{\lambda_r}} \leq 2\sqrt{\frac{r}{t}}\frac{\|H\|_{\infty,2}}{\sqrt{\lambda_r^*}}$$

$$\leq O\left(\sqrt{\frac{r}{t}}\sqrt{\frac{\log(\frac{t}{\delta})}{\log(\frac{1}{\delta})}}\|U^*Q - U\|\max_{i\in[t]}\frac{\|v^{*(i)}\|}{\sqrt{\lambda_r^*}} + 2\sqrt{\frac{r}{t}}\frac{2c\sigma}{\sqrt{\lambda_r^*}}\sqrt{\frac{r\log(\frac{27t}{\delta})}{m}}\right)$$

$$\leq O\left(\sqrt{\frac{\log(\frac{t}{\delta})}{\log(\frac{1}{\delta})}}\sqrt{\frac{\mu r}{t}}\|U^*Q - U\| + \frac{\sigma}{\sqrt{\lambda_r^*}}\sqrt{\frac{r^2\log(\frac{t}{\delta})}{mt}}\right) \tag{70}$$

Using (69) in (67), we get

$$\sqrt{\frac{r}{t}}\frac{\|H\|_F}{\sqrt{\lambda_r}} \leq 2\sqrt{\frac{r}{t}}\frac{\|H\|_F}{\sqrt{\lambda_r^*}} \leq O\left(\sqrt{\frac{\log(\frac{t}{\delta})}{\log(\frac{1}{\delta})}}\sqrt{\frac{\lambda_1^*}{\lambda_r^*}}\|(\mathbf{I} - U^*(U^*)^\top)U\|_F + \frac{\sigma}{\sqrt{\lambda_r^*}}\sqrt{\frac{r^2\log(\frac{t}{\delta})}{m}}\right) \tag{71}$$

Finally, by resetting $\mathcal{T}_k \leftarrow \mathcal{T}_k, |\mathcal{T}_k| = t/K \leftarrow t/K, S_1^{(i)} \leftarrow S_1^{(i)} = \frac{2}{m}\sum_{j\in[1,m/2]}x_j^{(i)}(x_j^{(i)})^\top$, we obtain the desired result. □

### B.2.2 Supporting lemmas for the analysis of update on $V$

Here we bound the linear operators in the $v^{(i)}$ update.

**Lemma B.5.** *Let* $\alpha = c\sqrt{\frac{r\log(27t/\delta)}{m}}$. *With a probability of at least* $1 - \delta$*, the following are true for all* $i \in [t]$

$$\|(U^\top S^{(i)}U)^\dagger\| \leq (1 + 2\alpha), \tag{72}$$

$$\|(U^\top S^{(i)}(U^*Q - U)Q^{-1}v^{*(i)}\| \leq (\|(\mathbf{I} - U^*(U^*)^\top)U\| + \alpha)\|(U^*Q - U)Q^{-1}v^{*(i)}\| \tag{73}$$

$$\leq (1 + \alpha)\|(U^*Q - U)Q^{-1}v^{*(i)}\| \tag{74}$$

$$\|U^\top S^{(i)}U_\perp U_\perp^\top U^*v^{*(i)}\| \leq \alpha\|U_\perp^\top U^*v^{*(i)}\|, \text{ and} \tag{75}$$

$$\|U^\top z^{(i)}\| \leq \sigma\alpha \tag{76}$$

*Proof of Lemma B.5.* Let $i \in [t]$.

Let $\mathcal{S} = \{v \in \mathbb{R}^r \mid \|v\| = 1\}$ be the set of all real vectors of dimension $r$ with unit Euclidean norm. For $\epsilon \leq 1$, there exists an $\epsilon$-net, $N_\epsilon \subset \mathcal{S}$, of size $(1 + 2/\epsilon)^r$ with respect to the Euclidean norm [49, Lemma 5.2]. That is for any $v' \in \mathcal{S}$, there exists some $v \in N_\epsilon$ such that $\|v' - v\|_F \leq \epsilon$.

Consider a $v \in N_\epsilon$, such that $\|v\|_F = 1$. Now we will prove with high-probability that $\langle ((U^\top S^{(i)} U) - \mathbf{I})v, v \rangle$ is small. Consider the the following quadratic form

$$v^\top (U^\top S^{(i)} U) v = \frac{1}{m} \sum_{j \in [m]} \mathrm{tr}(v^\top (U^\top x_j^{(i)} (x_j^{(i)})^\top U) v) = \frac{1}{m} \sum_{j \in [m]} \mathrm{tr}((x_j^{(i)})^\top U v v^\top U^\top x_j^{(i)}) \quad (77)$$

$x_j^{(i)} \sim \mathcal{N}(0, \mathbf{I}_{d \times d})$ are i.i.d. standard Gaussian random vectors. We will use Hanson-Wright inequality (Lemma F.5) to prove that the above quadratic form concentrates around its mean. In Lemma F.6 (which is a straightforward Corollary of Hanson-Wright inequality), by setting $a \leftarrow Uv, b \leftarrow Uv$, we get that with a probability of at least $1 - \delta$

$$\left| v^\top ((U^\top S^{(i)} U) - \mathbf{I}) v \right| \leq c \max \left( \sqrt{\frac{\log(1/\delta)}{m}}, \frac{\log(1/\delta)}{m} \right) := \Delta_\epsilon \quad (78)$$

For brevity, let $E = (U^\top S^{(i)} U) - \mathbf{I}$. Notice that $E$ is a real symmetric matrix, therefore it has an eigen decomposition. Then, let $v' \in \mathcal{S} \subset \mathbb{R}^r$ be the largest "eigenvector" of $E$, such that $(v')^\top E v' = \|E\| = \max_{\|\tilde{v}\|=1} \tilde{v}^\top E \tilde{v} = \max_{\|\tilde{v}\|=\|\tilde{v}'\|_F=1} \tilde{v}^\top E \tilde{v}'$. Then there exists some $v \in N_\epsilon$ such that $\|v' - v\| \leq \epsilon$.

$$\|E\|_F = (v')^\top E v = v^\top E v + (v' - v)^\top E v + (v')^\top E (v' - v) \quad (79)$$

$$\leq v^\top E v + \|v' - v\| \|E\| \|v\| + \|v'\| \|E\| \|v' - v\| \quad (80)$$

$$\leq v^\top E v + 2\epsilon \|E\| \quad (81)$$

Re-arranging and setting $\epsilon = 1/4$, and $c \leftarrow 2c$, we get

$$\|(U^\top S^{(i)} U) - \mathbf{I}\| = \|E\| \leq \Delta_{\frac{1}{4}} = \Delta. \quad (82)$$

where $\Delta = c \max \left( \sqrt{\frac{r \log(9/\delta)}{m}}, \frac{r \log(9/\delta)}{m} \right)$. If $m \geq \max(1, 4c^2) r \log(27t/\delta)$, then $\Delta \leq \alpha \leq 1/2$.

Thus with a probability of at least is is also implies that

$$\|(U^\top S^{(i)} U)^\dagger\| = (\sigma_{\min}(U^\top S^{(i)} U))^{-1} \leq \frac{1}{1 - \alpha} \leq 2. \quad (83)$$

Using similar arguments we can also prove that with a probability of at least $1 - \delta$

$$\|(U^\top S^{(i)} (U^* Q - U) Q^{-1} v^{*(i)}\|$$

$$\leq \|U^\top (U^* Q - U) Q^{-1} v^{*(i)}\| + \alpha \|(U^* Q - U) Q^{-1} v^{*(i)}\| \quad (84)$$

$$\leq \|U^\top (\mathbf{I} - U^* (U^*)^\top) U Q^{-1} v^{*(i)}\| + \alpha \|(U^* Q - U) Q^{-1} v^{*(i)}\| \quad (85)$$

$$\leq \|U^\top (\mathbf{I} - U^* (U^*)^\top)^2 U Q^{-1} v^{*(i)}\| + \alpha \|(U^* Q - U) Q^{-1} v^{*(i)}\| \quad (86)$$

$$\leq \|U^\top (\mathbf{I} - U^* (U^*)^\top)(U^* Q - U) Q^{-1} v^{*(i)}\| + \alpha \|(U^* Q - U) Q^{-1} v^{*(i)}\| \quad (87)$$

$$\leq \|(\mathbf{I} - U^* (U^*)^\top) U\| \|(U^* Q - U) Q^{-1} v^{*(i)}\| + \alpha \|(U^* Q - U) Q^{-1} v^{*(i)}\| \quad (88)$$

$$\leq (\|(\mathbf{I} - U^* (U^*)^\top) U\| + \alpha) \|(U^* Q - U) Q^{-1} v^{*(i)}\| \quad (89)$$

$$\leq (1 + \alpha) \|(U^* Q - U) Q^{-1} v^{*(i)}\|, \quad (90)$$

Using similar arguments we can also prove that with a probability of at least $1 - \delta$

$$\left\| U^\top S^{(i)} U_\perp U_\perp^\top U^* v^{*(i)} \right\| \leq \alpha \left\| U_\perp^\top U^* v^{*(i)} \right\| \quad (91)$$

and with a probability of at least $1 - \delta$

$$\left\| U^\top z^{(i)} \right\| \leq \sigma \alpha \quad (92)$$

Finally setting $\delta \leftarrow \delta/3/t$ and taking the union bound over three bounds over all the tasks in $[t]$ gets us the desired result. □

Here we prove the approximate incoherence of the intermediate $V$ and the spectrum of intermediate $W$.

**Lemma B.6** (Incoherence of intermediate $v^{(i)}$). *If* $\|H\|_F \le (1 - \frac{1}{\sqrt{2}})\sqrt{(t/r)\lambda_r((r/t)W^*)}$, $\|H\|_{\infty,2}^2 \le O(\mu\lambda_r((r/t)W^*))$, $\|Q\| \le 1$ *and* $\frac{10}{11} \le \sigma_{\min}(Q)$, *and* (67) *and* (68) *are true, then*

$$\|v^{(i)}\| \le O\left(\mu\,\lambda_r((r/t)W)\right),\ \lambda_1((r/t)W) \le 2\lambda_1((r/t)W^*)\,,\ and \tag{93}$$

$$(1/2)\lambda_r((r/t)W^*) \le \lambda_r((r/t)W) \le (4/3)\lambda_r((r/t)W^*) \tag{94}$$

*Proof of Lemma B.6.*

$$\|v^{(i)}\| \le \|Q^{-1}v^{*(i)}\| + \|v^{(i)} - Q^{-1}v^{*(i)}\| \le 2\|v^{*(i)}\| + \|h^{(i)}\| \tag{95}$$

$$\implies \|v^{(i)}\|^2 \le O(\|V^*\|_{\infty,2}^2) + O(\|H\|_{\infty,2}^2) \le O\left(\mu\lambda_r((r/t)W^*)\right) \tag{96}$$

where the second inequality use the definition $h^{(i)} = v^{(i)} - Q^{-1}v^{*(i)}$ and $\|Q^{-1}\| \le 2$ (as $\sigma_{\min}(Q) \ge \frac{1}{2}$), the third inequality use the fact that $a + b \le 2a^2 + 2b^2$ an (68), and the final inequality uses $\|H\|_{\infty,2} \le \|V\|_{\infty,2}$.

Notice that $W = V^\top V$ and $W^* = (V^*)^\top V^*$. Thus $\sqrt{\lambda_r((r/t)W)} = \sqrt{(r/t)}\sigma_r(V)$ and $\sqrt{\lambda_r((r/t)W^*)} = \sqrt{(r/t)}\sigma_r(W^*)$, and both $W$ and $W^*$ are positive semi-definite (PSD). Similarly, using $\sigma_{\min}(Q^{-1}) = \sigma_{\min}(((U^*)^\top U)^{-1}) \ge 1$ and Lemma F.1 we can get that

$$\sqrt{\lambda_r((r/t)W^*)} \le \sqrt{\sigma_{\min}^2(Q^{-1})\lambda_r((r/t)W^*)} \le \sqrt{(r/t)\lambda_r(Q^{-1}(V^*)^\top V^* Q^{-\top})}$$

$$\le \sqrt{(r/t)}\sigma_r(V^* Q^{-\top}) \tag{97}$$

Therefore, instead of analyzing the relation between $\lambda_r(W)$ and $\lambda_r(W^*)$, we can analyze the relation between $\sigma_r(V)$ and $\sigma_r(V^*)$. Notice that $V^* Q^{-\top} = V + V^* Q^{-\top} - V$. Then by Weyl's inequality (Lemma F.2, by setting $A \leftarrow V^* Q^{-\top}$, $B \leftarrow V$, and $C \leftarrow V^* Q^{-\top} - V$) we get that

$$\sqrt{\lambda_r((r/t)W^*)} \le \sqrt{(r/t)}\sigma_r(V^* Q^{-\top}) \le \sqrt{(r/t)}\sigma_r(V) + \sqrt{(r/t)}\|V - V^* Q^{-\top}\| \tag{98}$$

$$\le \sqrt{\lambda_r((r/t)W)} + \sqrt{(r/t)}\|H\| \tag{99}$$

$$\le \sqrt{\lambda_r((r/t)W)} + \sqrt{(r/t)}\|H\|_F \tag{100}$$

$$\le \sqrt{\lambda_r((r/t)W)} + (1 - \frac{1}{\sqrt{2}})\sqrt{\lambda_r((r/t)W^*)} \tag{101}$$

where the last inequality uses $\|H\|_F \le (1 - \frac{1}{\sqrt{2}})\sqrt{(t/r)\lambda_r((r/t)W^*)}$. Finally we get the desired result: $\lambda_r((r/t)W^*) \le 2\lambda_r((r/t)W^*)$ by re-arranging the terms. Using similar arguments we can show that $\lambda_r((r/t)W^*) \le 2\lambda_r((r/t)W)$ as follows.

$$\sqrt{\lambda_r((r/t)W)} \le \sqrt{(r/t)}\sigma_r(V) \le \sqrt{(r/t)}\sigma_r(V^* Q^{-\top}) + \sqrt{(r/t)}\|V - V^* Q^{-\top}\| \tag{102}$$

$$\le \|Q^{-\top}\|\sqrt{(r/t)}\sigma_r(V^*) + \sqrt{(r/t)}\|H\| \tag{103}$$

$$\le (1 + \frac{1}{10})\sqrt{\lambda_r((r/t)W^*)} + (1 - \frac{1}{\sqrt{2}})\sqrt{\lambda_r((r/t)W^*)} \tag{104}$$

$$\le \sqrt{2\lambda_r((r/t)W^*)} \tag{105}$$

Similarly, we derive a relation between $\lambda_r(W)$ and $\lambda_r(W^*)$. Notice that $V^* Q^{-\top} = V + V^* Q^{-\top} - V$. Then by Weyl's inequality (Lemma F.2, by setting $B \leftarrow V^* Q^{-\top}$, $A \leftarrow V$, and $C \leftarrow V^* Q^{-\top} - V$) we get that

$$\sqrt{\lambda_1((r/t)W)} \le \sqrt{(r/t)}\sigma_1(V) \le \sqrt{(r/t)}\sigma_1(V^* Q^{-\top}) + \sqrt{(r/t)}\|V - V^* Q^{-T}\| \tag{106}$$

$$\le \|Q^{-1}\|\sqrt{\lambda_1((r/t)W^*)} + \sqrt{(r/t)}\|H\| \tag{107}$$

$$\le (1 + \frac{1}{10})\sqrt{\lambda_1((r/t)W^*)} + \sqrt{(r/t)}\|H\|_F \tag{108}$$

$$\le (1 + \frac{1}{10})\sqrt{\lambda_1((r/t)W^*)} + (1 - \frac{1}{\sqrt{2}})\sqrt{\lambda_r((r/t)W^*)} \tag{109}$$

$$\le \sqrt{2}\sqrt{\lambda_1((r/t)W^*)} \tag{110}$$

where the last second inequality uses $\|H\|_F \leq (1 - \frac{1}{\sqrt{2}})\sqrt{(t/r)\lambda_r((r/t)W^*)}$, and the last inequality uses $\lambda_r(\cdot) \leq \lambda_1(\cdot)$. Finally we get the desired result by re-arranging the terms. $\qquad \square$

## B.3 Analysis of update on $U$

### B.3.1 Proof of Lemma B.3

*Proof of Lemma B.3.* In this proof for brevity, we will first set that $\mathcal{T}_k \leftarrow [t]$, $|\mathcal{T}_k| = t/K \leftarrow t$, $S_2^{(i)} \leftarrow S^{(i)} = \frac{1}{m}\sum_{j\in[m]} x_j^{(i)}(x_j^{(i)})^\top$. This can be done due to the approximate equivalence of the subset $\mathcal{T}_k$ and the set of all task $[t]$ by Lemma B.1, which requires that $t \geq \Omega(\mu^2 r^3 K \log(\frac{K}{\delta}))$. Finally at the end of the analysis we will reset $\mathcal{T}_k \leftarrow \mathcal{T}_k$, $|\mathcal{T}_k| = t/K \leftarrow t/K$, $S_2^{(i)} \leftarrow S_2^{(i)} = \frac{2}{m}\sum_{j\in[m/2+1,m]} x_j^{(i)}(x_j^{(i)})^\top$.

Recall that

$$\widehat{U} - U^*Q = \mathcal{W}^{-\frac{1}{2}}(\mathcal{I} + \mathcal{E}_1)(-(\mathcal{W}^{-\frac{1}{2}}\mathcal{H} + \mathcal{E}_2)(U^*Q) + \mathcal{W}^{-\frac{1}{2}}(\sum_{i\in[t]} z^{(i)}(v^{(i)})^\top)) \qquad (111)$$

where $\mathcal{E}_1 = (\mathcal{W}^{-\frac{1}{2}}\mathcal{A}\mathcal{W}^{-\frac{1}{2}})^\dagger - \mathcal{I}$ and $\mathcal{E}_2 = \mathcal{W}^{-\frac{1}{2}}\widehat{\mathcal{H}} - \mathcal{W}^{-\frac{1}{2}}\mathcal{H}$, and $F = \widehat{U} - U^*Q + \mathcal{W}^{-1}(\mathcal{H}(U^*Q))$. Therefore

$$\|F\|_F \leq \|\mathcal{W}^{-\frac{1}{2}}\|_F(\|\mathcal{E}_1\|_F\|\mathcal{W}^{-\frac{1}{2}}\mathcal{H}(U^*Q)\|_F + \|\mathcal{I} +$$
$$\mathcal{E}_1\|_F(\|\mathcal{E}_2(U^*Q)\|_F + \|\mathcal{W}^{-\frac{1}{2}}(\sum_{i\in[t]} z^{(i)}(v^{(i)})^\top))\|_F)) \qquad (112)$$

We can trivially bound $\|\mathcal{W}^{-\frac{1}{2}}\|_F$ as follows. For all $\|U\|_F = 1$, the following is true.

$$\|\mathcal{W}^{-\frac{1}{2}}(U)\|_F = \|UW^{-\frac{1}{2}}\|_F \leq \|U\|_F\|W^{-\frac{1}{2}}\| \leq \sqrt{\frac{r/t}{\lambda_r}} \qquad (113)$$

$\Omega(\mu dr^2 \log(1/\delta)) \leq mt$ and approximate incoherence of intermediate $V$ (26) implies that $\Omega(dr\frac{\|V\|_{\infty,2}^2}{\lambda_r(W)/t}\log(1/\delta)) \leq \Omega(\mu dr^2\log(1/\delta)) \leq mt$, then by Lemma B.7 we have that, with a probability of at least $1 - \delta/3$

$$\|\mathcal{E}_1\|_F \leq 3c\sqrt{\frac{dr\|V\|_{\infty,2}^2\log(27/\delta)}{m\,\lambda_r(W)}} \leq 3c\sqrt{\frac{\mu dr^2\log(27/\delta)}{mt}} \leq \frac{1}{2} \qquad (114)$$

This also implies that

$$\|\mathcal{I} + \mathcal{E}_1\|_F \leq \|\mathcal{I}\| + \|\mathcal{E}_1\|_F \leq 1 + \Delta \leq \frac{3}{2} \qquad (115)$$

By Lemma B.8,

$$\|(\mathcal{W}^{-\frac{1}{2}}\mathcal{H})(U^*Q)\|_F \leq \|H\|_F \qquad (116)$$

and with a probability of at least $1 - \delta/3$

$$\|\mathcal{E}_2(U^*Q)\|_F \leq c(\min(\|H\|_F\frac{\|V\|_{\infty,2}}{\sqrt{\lambda_r(W)}}, \|H\|_{\infty,2})\sqrt{\frac{dr\,\log(15/\delta)}{m}} +$$
$$\|H\|_{\infty,2}\frac{\|V\|_{\infty,2}}{\sqrt{\lambda_r(W)}}\frac{dr\,\log(15/\delta)}{m}) \qquad (117)$$

Using the approximate incoherence of $V$ (26) in the above inequality, we get that

$$\|\mathcal{E}_2(U^*Q)\|_F \leq c(\min(\|H\|_F\sqrt{\frac{\mu r}{t}}, \|H\|_{\infty,2})\sqrt{\frac{dr\,\log(15/\delta)}{m}} + \|H\|_{\infty,2}\sqrt{\frac{\mu r}{t}}\cdot\frac{dr\,\log(15/\delta)}{m}) \qquad (118)$$

By Lemma B.9 with a probability of at least $1 - \delta/3$

$$\|\sum_{i \in [t]} \mathcal{W}^{-\frac{1}{2}}(z^{(i)}(v^{(i)})^\top))\|_F \leq O\Big(\sigma\sqrt{\frac{dr}{m}\log\Big(\frac{t}{\delta}\Big)\log\Big(\frac{r}{\delta}\Big)}\Big) \tag{119}$$

Finally taking union bound over the above results and using Lemma B.2, we can bound each of the terms constituting $F$. Using (113), (116) and (27) (recall that we set $t \leftarrow t/K$) we get

$$\|\mathcal{W}^{-1}\mathcal{H}(U^*Q)\|_F \leq \|\mathcal{W}^{-\frac{1}{2}}\|_F\|\mathcal{W}^{-\frac{1}{2}}\mathcal{H}(U^*Q)\|_F \tag{120}$$

$$\leq \sqrt{\frac{r}{t}}\frac{\|H\|_F}{\sqrt{\lambda_r}} \leq O\Big(\sqrt{\frac{\lambda_1^*}{\lambda_r^*}}\sqrt{\frac{\log(\frac{t}{\delta})}{\log(\frac{1}{\delta})}}\|(\mathbf{I} - U^*(U^*)^\top)U\|_F + \frac{\sigma}{\sqrt{\lambda_r^*}}\sqrt{\frac{r^2\log(\frac{t}{\delta})}{m}}\Big) \tag{121}$$

Using (113), (115), (116), and (27) we get

$$\|\mathcal{W}^{-\frac{1}{2}}\|_F\|\mathcal{E}_1\|_F\|\mathcal{W}^{-\frac{1}{2}}\mathcal{H}(U^*Q)\|_F \tag{122}$$

$$\leq O\Big(\sqrt{\frac{\mu dr^2\log(\frac{1}{\delta})}{mt}}\sqrt{\frac{r}{t}}\frac{\|H\|_F}{\sqrt{\lambda_r}}\Big) \tag{123}$$

$$\leq O\Big(\sqrt{\frac{\lambda_1^*}{\lambda_r^*}\frac{\mu dr^2\log(\frac{t}{\delta})}{mt}}\|(\mathbf{I} - U^*(U^*)^\top)U\|_F + \sqrt{\frac{\mu dr^2\log(\frac{1}{\delta})}{mt}}\frac{\sigma}{\sqrt{\lambda_r^*}}\sqrt{\frac{r^2\log(\frac{t}{\delta})}{m}}\Big) \tag{124}$$

Using (113), (115), (118), (27) and (28) we get

$$\|\mathcal{W}^{-\frac{1}{2}}\|_F\|\mathcal{I} + \mathcal{E}_1\|_F\|\mathcal{E}_2(U^*Q)\|_F \tag{125}$$

$$\leq O\Big(\sqrt{\frac{r}{t}}\min\Big(\frac{\|H\|_F}{\sqrt{\lambda_r}}\sqrt{\frac{\mu r}{t}}, \frac{\|H\|_{\infty,2}}{\sqrt{\lambda_r}}\Big)\sqrt{\frac{dr\,\log(\frac{1}{\delta})}{m}} + \sqrt{\frac{r}{t}}\frac{\|H\|_{\infty,2}}{\sqrt{\lambda_r}}\sqrt{\frac{\mu r}{t}}\frac{dr}{m}\frac{\log(\frac{1}{\delta})}{m}\Big) \tag{126}$$

$$\leq O\Big(\min\Big(\sqrt{\frac{\lambda_1^*}{\lambda_r^*}\frac{\mu dr^2\log(\frac{t}{\delta})}{mt}}\|(\mathbf{I} - U^*(U^*)^\top)U\|_F + \sqrt{\frac{\mu dr^2\log(\frac{1}{\delta})}{mt}}\frac{\sigma}{\sqrt{\lambda_r^*}}\sqrt{\frac{r^2\log(\frac{t}{\delta})}{m}},$$

$$\sqrt{\frac{\mu dr^2\log(\frac{t}{\delta})}{mt}}\|(\mathbf{I} - U^*(U^*)^\top)U\| + \sqrt{\frac{dr\log(\frac{1}{\delta})}{m}}\frac{\sigma}{\sqrt{\lambda_r^*}}\sqrt{\frac{r^2\log(\frac{t}{\delta})}{mt}}\Big) +$$

$$\frac{\mu dr^2\log(\frac{t}{\delta})}{mt}\|(\mathbf{I} - U^*(U^*)^\top)U\| + \frac{\sqrt{\mu}dr\sqrt{r}\log(\frac{1}{\delta})}{m\sqrt{t}}\frac{\sigma}{\sqrt{\lambda_r^*}}\sqrt{\frac{r^2\log(\frac{t}{\delta})}{mt}}\Big) \tag{127}$$

$$\leq O\Big(\sqrt{\frac{\mu dr^2\log(\frac{t}{\delta})}{mt}}\|(\mathbf{I} - U^*(U^*)^\top)U\| + \sqrt{\frac{dr\log(\frac{1}{\delta})}{m}}\frac{\sigma}{\sqrt{\lambda_r^*}}\sqrt{\frac{r^2\log(\frac{t}{\delta})}{mt}}\Big) + \tag{128}$$

$$\frac{\mu dr^2\log(\frac{t}{\delta})}{mt}\|(\mathbf{I} - U^*(U^*)^\top)U\| + \frac{\sqrt{\mu}dr\sqrt{r}\log(\frac{1}{\delta})}{m\sqrt{t}}\frac{\sigma}{\sqrt{\lambda_r^*}}\sqrt{\frac{r^2\log(\frac{t}{\delta})}{mt}}\Big) \tag{129}$$

Using (113), (115), (119), and (26) we get

$$\|\mathcal{W}^{-\frac{1}{2}}\|_F\|\mathcal{I} + \mathcal{E}_1\|_F\|\sum_{i \in [t]} \mathcal{W}^{-\frac{1}{2}}(z^{(i)}(v^{(i)})^\top))\|_F \leq O\Big(\frac{\sigma}{\sqrt{\lambda_r^*}}\sqrt{\frac{dr^2\log(\frac{t}{\delta})\log(\frac{r}{\delta})}{mt}}\Big) \tag{130}$$

Substituting (121), (124), (129), and (130) in (112) we get

$$\|F\|_F \leq \|\mathcal{W}^{-\frac{1}{2}}\|_F (\|\mathcal{E}_1\|_F \|\mathcal{W}^{-\frac{1}{2}}\mathcal{H}(U^*Q)\|_F + \|\mathcal{I} + \mathcal{E}_1\|_F (\|\mathcal{E}_2(U^*Q)\|_F + \|\sum_{i\in[t]} \mathcal{W}^{-\frac{1}{2}}(z^{(i)}(v^{(i)})^\top)\|_F)) \tag{131}$$

$$\leq O\Big(\sqrt{\frac{\lambda_1^*}{\lambda_r^*}\frac{\mu dr^2 \log(\frac{t}{\delta})}{mt}}\|(\mathbf{I} - U^*(U^*)^\top)U\|_F + \sqrt{\frac{\mu dr^2 \log(\frac{1}{\delta})}{mt}}\frac{\sigma}{\sqrt{\lambda_r^*}}\sqrt{\frac{r^2 \log(\frac{t}{\delta})}{m}}\Big) + \tag{132}$$

$$O\Big(\frac{\mu dr^2 \log(\frac{t}{\delta})}{mt}\|(\mathbf{I} - U^*(U^*)^\top)U\| + \frac{\sqrt{\mu}dr\sqrt{r}\log(\frac{1}{\delta})}{mt}\frac{\sigma}{\sqrt{\lambda_r^*}}\sqrt{\frac{r^2 \log(\frac{t}{\delta})}{m}}\Big) +$$

$$O\Big(\frac{\sigma}{\sqrt{\lambda_r^*}}\sqrt{\frac{dr^2 \log(\frac{t}{\delta})\log(\frac{r}{\delta})}{mt}}\Big) \tag{133}$$

$$\leq O\Big(\sqrt{\frac{\lambda_1^*}{\lambda_r^*}\frac{\mu dr^2 \log(\frac{t}{\delta})}{mt}}\|(\mathbf{I} - U^*(U^*)^\top)U\|_F + \frac{\sigma}{\sqrt{\lambda_r^*}}\sqrt{\frac{\mu dr^2 \log(\frac{t}{\delta})\log(\frac{r}{\delta})}{mt}}\Big) \tag{134}$$

where the second-last inequality used the fact that $mt \geq \Omega(\mu dr^2 \log(\frac{t}{\delta}))$. Finally, by resetting $\mathcal{T}_k \leftarrow \mathcal{T}_k, |\mathcal{T}_k| = t/K \leftarrow t/K, S_2^{(i)} \leftarrow S_2^{(i)} = \frac{2}{m}\sum_{j\in[m/2+1,m]} x_j^{(i)}(x_j^{(i)})^\top$, we obtain the desired result. $\square$

### B.3.2 Supporting lemmas for the analysis of update on $U$

**Lemma B.7.** If $\max(1, 4c^2)dr\frac{\|V\|_{\infty,2}^2}{\lambda_r(W)/t}\log(27/\delta) \leq mt$, then with a probability of at least $1 - \delta/3$,

$$\|\mathcal{E}_1\|_F \leq 3c\sqrt{\frac{dr\|V\|_{\infty,2}^2 \log(27/\delta)}{m\,\lambda_r(W)}} \tag{135}$$

*Proof of Lemma B.7.* Let $\mathcal{S}_F = \{U \in \mathbb{R}^{d\times r} \mid \|U\|_F = 1\}$ be the set of all real matrices of dimensions $d \times r$ with unit Frobenius norm. For $\epsilon \leq 1$, there exists an $\epsilon$-net, $N_\epsilon \subset \mathcal{S}_F$, of size $(1 + 2/\epsilon)^{dr}$ with respect to the Frobenius norm [49, Lemma 5.2]. That is for any $U' \in \mathcal{S}_F$, there exists some $U \in N_\epsilon$ such that $\|U' - U\|_F \leq \epsilon$.

Consider a $U \in N_\epsilon$, such that $\|U\|_F = 1$. Now we will prove with high-probability that $\langle(\mathcal{W}^{-\frac{1}{2}}\mathcal{A}\mathcal{W}^{-\frac{1}{2}} - \mathcal{I})(U), U\rangle$ is small. Consider the the following quadratic form

$$\langle(\mathcal{W}^{-\frac{1}{2}}\mathcal{A}\mathcal{W}^{-\frac{1}{2}})(U), U\rangle = \Big\langle \sum_{i\in[t]} S^{(i)}UW^{-\frac{1}{2}}v^{(i)}(v^{(i)})^\top W^{-\frac{1}{2}}, U\Big\rangle \tag{136}$$

$$= \sum_{i\in[t]}\frac{1}{m}\sum_{j\in[m]}(x_j^{(i)})^\top(UW^{-\frac{1}{2}}v^{(i)}(v^{(i)})^\top W^{-\frac{1}{2}}U^\top)x_j^{(i)} \tag{137}$$

where $S^{(i)} = \frac{1}{m}\sum_{j\in[m]} x_j^{(i)}(x_j^{(i)})^\top$ and $x_j^{(i)} \sim \mathcal{N}(0, \mathbf{I}_{d\times d})$ are i.i.d. standard Gaussian random vectors and $W = \sum_{i\in[t]} v^{(i)}(v^{(i)})^\top$ is rank-$r$ matrix. We will use Hanson-Wright inequality (Lemma F.5) to prove that the above quadratic form concentrates around its mean. Notice that the the expectation of $\langle(\mathcal{W}^{-\frac{1}{2}}\mathcal{A}\mathcal{W}^{-\frac{1}{2}})(U), U\rangle$ is $\langle\mathcal{I}(U), U\rangle$.

$$\sum_{i\in[t]}\mathbb{E}\Big[\Big\langle S^{(i)}UW^{-\frac{1}{2}}v^{(i)}(v^{(i)})^\top W^{-\frac{1}{2}}, U\Big\rangle\Big] = \Big\langle UW^{-\frac{1}{2}}\sum_{i\in[t]}v^{(i)}(v^{(i)})^\top W^{-\frac{1}{2}}, U\Big\rangle \tag{138}$$

$$= \langle U, U\rangle = \|U\|_F^2 = 1. \tag{139}$$

We will also need the following bounds to apply the Hanson-Wright inequality. Recall that $\|V\|_{\infty,2} = \max_{i\in[t]}\|v^{(i)}\|$. Then,

$$\max_{i\in[t]}\|UW^{-\frac{1}{2}}v^{(i)}(v^{(i)})^\top W^{-\frac{1}{2}}U^\top\| = \max_{i\in[t]}\|UW^{-\frac{1}{2}}v^{(i)}\|^2 \leq \max_{i\in[t]}\|U\|^2\|W^{-1}\|^2\|v^{(i)}\|^2 \quad (140)$$

$$\leq \frac{\|V\|_{\infty,2}^2}{\lambda_r(W)} \quad (141)$$

Also note that,

$$\sum_{i\in[t]}\|UW^{-\frac{1}{2}}v^{(i)}(v^{(i)})^\top W^{-\frac{1}{2}}U^\top\|_F^2 = \sum_{i\in[t]}\|UW^{-\frac{1}{2}}v^{(i)}\|^4 \quad (142)$$

$$= \max_{i\in[t]}\|UW^{-\frac{1}{2}}v^{(i)}\|^2 \sum_{i\in[t]}\left\langle UW^{-\frac{1}{2}}v^{(i)}, UW^{-\frac{1}{2}}v^{(i)}\right\rangle \quad (143)$$

$$\leq \frac{\|V\|_{\infty,2}^2}{\lambda_r(W)} \quad (144)$$

where the last inequality used (138) and (141). Then by Hanson-Wright inequality (Lemma F.5), with probability at least $1 - \delta/|N_\epsilon|$

$$\left|\left\langle(\mathcal{W}^{-\frac{1}{2}}\mathcal{A}\mathcal{W}^{-\frac{1}{2}} - \mathcal{I})(U), U\right\rangle\right| \quad (145)$$

$$= \left|\left\langle\sum_{i\in[t]}\frac{1}{m}\sum_{j\in[m]}x_j^{(i)}(x_j^{(i)})^\top UW^{-\frac{1}{2}}v^{(i)}(v^{(i)})^\top W^{-\frac{1}{2}}, U\right\rangle - \langle U, U\rangle\right| \leq \Delta_\epsilon \quad (146)$$

where $\Delta_\epsilon = c\max(\sqrt{\frac{\|V\|_{\infty,2}^2\log(|N_\epsilon|/\delta)}{m\,\lambda_r(W)}}, \frac{\|V\|_{\infty,2}^2\log(|N_\epsilon|/\delta)}{m\,\lambda_r(W)})$. Taking union bound over all $U \in N_\epsilon$ implies that with probability at least $1 - \delta$

$$\left|\left\langle(\mathcal{W}^{-\frac{1}{2}}\mathcal{A}\mathcal{W}^{-\frac{1}{2}} - \mathcal{I})(U), U\right\rangle\right| \leq \Delta_\epsilon , \text{ for all } U \in N_\epsilon . \quad (147)$$

For brevity, let $\mathcal{E}_1'(U) = (\mathcal{W}^{-\frac{1}{2}}\mathcal{A}\mathcal{W}^{-\frac{1}{2}} - \mathcal{I})(U)$. Notice that $\mathcal{E}_1'$ is self-adjoint, therefore it has an eigen decomposition with respect to the Frobenius norm. Then, let $U' \in \mathcal{S}_F \subset \mathbb{R}^{d\times r}$ be the largest "eigenmatrix" of $\mathcal{E}_1$, such that $\langle\mathcal{E}_1'(U), U\rangle = \|\mathcal{E}_1'\|_F = \max_{\|\widetilde{U}\|_F=1}\left\langle\mathcal{E}_1'(\widetilde{U}), \widetilde{U}\right\rangle = \max_{\|\widetilde{U}\|_F=\|\widetilde{U}'\|_F=1}\left\langle\mathcal{E}_1'(\widetilde{U}), \widetilde{U}'\right\rangle$. Then there exists some $U \in N_\epsilon$ such that $\|U' - U\|_F \leq \epsilon$.

$$\|\mathcal{E}_1'\|_F = \langle\mathcal{E}_1'(U'), U'\rangle = \langle\mathcal{E}_1'(U), U\rangle + \langle\mathcal{E}_1'(U' - U), U\rangle + \langle\mathcal{E}_1'(U'), U' - U\rangle \quad (148)$$

$$\leq \langle\mathcal{E}_1'(U), U\rangle + \|\mathcal{E}_1'\|_F\|U' - U\|_F(\|U\|_F + \|U'\|_F) \quad (149)$$

$$\leq \langle\mathcal{E}_1'(U), U\rangle + 2\epsilon\|\mathcal{E}_1'\|_F \quad (150)$$

Re-arranging and setting $\epsilon = 1/4$, and $c \leftarrow 2c$, we get

$$\|\mathcal{W}^{-\frac{1}{2}}\mathcal{A}\mathcal{W}^{-\frac{1}{2}} - \mathcal{I}\|_F = \|\mathcal{E}_1'\|_F \leq \Delta_{\frac{1}{4}} = \Delta. \quad (151)$$

where $\Delta = c\max\left(\sqrt{\frac{dr\,\|V\|_{\infty,2}^2\log(9/\delta)}{m\,\lambda_r(W)}}, \frac{dr\,\|V\|_{\infty,2}^2\log(9/\delta)}{m\,\lambda_r(W)}\right)$.

For brevity, let $\widehat{\mathcal{A}}(U) = (\mathcal{W}^{-\frac{1}{2}}\mathcal{A}\mathcal{W}^{-\frac{1}{2}})(U)$. Notice that $\widehat{\mathcal{A}}$ is self-adjoint, therefore it has an eigen decomposition with respect to the Frobenius norm. Then, let $U' \in \mathcal{S}_F \subset \mathbb{R}^{d\times r}$ be the smallest "eigenmatrix" of $\widehat{\mathcal{A}}$, such that $\left\langle\widehat{\mathcal{A}}(U), U\right\rangle = \lambda_{\min}(\widehat{\mathcal{A}}) = \min_{\|\widetilde{U}\|_F=1}\left\langle\widehat{\mathcal{A}}(\widetilde{U}), \widetilde{U}\right\rangle = \min_{\|\widetilde{U}\|_F=\|\widetilde{U}'\|_F=1}\left\langle\widehat{\mathcal{A}}(\widetilde{U}), \widetilde{U}'\right\rangle$. Then there exists some $U \in N_\epsilon$ such that $\|U' - U\|_F \leq \epsilon$.

$$\lambda_{\min}(\widehat{\mathcal{A}}) = \langle\widehat{\mathcal{A}}(U'), U'\rangle = \langle\mathcal{I}(U), U\rangle + \langle(\widehat{\mathcal{A}} - \mathcal{I})(U), U\rangle + \langle\widehat{\mathcal{A}}(U' - U), U\rangle + \langle\widehat{\mathcal{A}}(U'), U' - U\rangle \quad (152)$$

$$\geq 1 - \left|\langle(\widehat{\mathcal{A}} - \mathcal{I})(U), U\rangle\right| - \lambda_{\min}(\widehat{\mathcal{A}})\|U' - U\|_F(\|U\|_F + \|U'\|_F) \quad (153)$$

$$\geq 1 - \Delta_\epsilon - 2\epsilon\lambda_{\min}(\widehat{\mathcal{A}}) \quad (154)$$

Re-arranging and setting $\epsilon = 1/4$, and $c \leftarrow 2c$, we get that $\lambda_{\min}(\widehat{\mathcal{A}}) \geq \frac{2}{3}(1 - \Delta)$. Therefore,

$$\|(\mathcal{W}^{-\frac{1}{2}}\mathcal{A}\mathcal{W}^{-\frac{1}{2}})^{\dagger}\|_F = \frac{1}{\lambda_{\min}(\widehat{\mathcal{A}})} \leq \frac{3}{2(1 - \Delta)}. \tag{155}$$

where $\Delta = c\max\left(\sqrt{\frac{dr\,\|V\|_{\infty,2}^2\,\log(9/\delta)}{m\,\lambda_r(W)}}, \frac{dr\,\|V\|_{\infty,2}^2\,\log(9/\delta)}{m\,\lambda_r(W)}\right).$ If $\max(1, 4c^2)dr\frac{\|V\|_{\infty,2}^2}{\lambda_r(W)/t}\log(27/\delta) \leq mt$, we get that $\Delta \leq c\sqrt{\frac{dr\,\|V\|_{\infty,2}^2\,\log(9/\delta)}{m\,\lambda_r(W)}} \leq \frac{1}{2}.$

By setting $A + B = \mathcal{W}^{-\frac{1}{2}}\mathcal{A}\mathcal{W}^{-\frac{1}{2}}$ and $A = \mathcal{I}$ such that $\mathcal{E}_1 = (A+B)^{-1} - B^{-1}$, in the Woodburry matrix inverse identity (359) (Lemma F.3) we get that, with a probability of at least $1 - \delta$

$$\|(A+B)^{-1} - A^{-1}\|_F \leq \|A^{-1}\|_F\|B\|_F\|(A+B)^{-1}\|_F \tag{156}$$

$$\implies \|\mathcal{E}_1\|_F \leq \left\|(\mathcal{W}^{-\frac{1}{2}}\mathcal{A}\mathcal{W}^{-\frac{1}{2}})^{\dagger} - \mathcal{I}\right\|_F \leq \|\mathcal{I}^{\dagger}\|_F\|\mathcal{W}^{-\frac{1}{2}}\mathcal{A}\mathcal{W}^{-\frac{1}{2}} - \mathcal{I}\|_F\|(\mathcal{W}^{-\frac{1}{2}}\mathcal{A}\mathcal{W}^{-\frac{1}{2}})^{\dagger}\|_F \tag{157}$$

$$\leq 1 \cdot \Delta \cdot \frac{3}{2(1 - \Delta)} \leq 3\Delta \leq 3c\sqrt{\frac{dr\,\|V\|_{\infty,2}^2\,\log(9/\delta)}{m\,\lambda_r(W)}} \tag{158}$$

Finally, setting $\delta \leftarrow \delta/3$ get us the desired result. $\qquad\square$

**Lemma B.8.** $\|(\mathcal{W}^{-\frac{1}{2}}\mathcal{H})(U^*Q)\|_F \leq \|H\|_F$ and with a probability of at least $1 - \delta/3$

$$\|\mathcal{E}_2(U^*Q)\|_F \leq c(\min(\|H\|_F\frac{\|V\|_{\infty,2}}{\sqrt{\lambda_r(W)}}, \|H\|_{\infty,2})\sqrt{\frac{dr\,\log(15/\delta)}{m}} +$$

$$\|H\|_{\infty,2}\frac{\|V\|_{\infty,2}}{\sqrt{\lambda_r(W)}}\frac{dr\,\log(15/\delta)}{m}) \tag{159}$$

*Proof of Lemma B.8.* First we prove that the expected value $\mathbb{E}[(\mathcal{W}^{-\frac{1}{2}}\widehat{\mathcal{H}})(U^*Q)] = (\mathcal{W}^{-\frac{1}{2}}\mathcal{H})(U^*Q)$ is bounded.

$$\|(\mathcal{W}^{-\frac{1}{2}}\mathcal{H})(U^*Q)\|_F = \max_{\|U\|_F = 1}\left\langle(\mathcal{W}^{-\frac{1}{2}}\mathcal{H})(U^*Q), U\right\rangle \tag{160}$$

$$= \max_{\|U\|_F = 1}\sum_{i\in[t]}\left\langle U^*Qh^{(i)}(v^{(i)})^\top W^{-\frac{1}{2}}, U\right\rangle \tag{161}$$

$$= \max_{\|U\|_F = 1}\sum_{i\in[t]}\left\langle U^*Qh^{(i)}, UW^{-\frac{1}{2}}v^{(i)}\right\rangle \tag{162}$$

$$\leq \max_{\|U\|_F = 1}\sqrt{\sum_{i\in[t]}\|U^*Qh^{(i)}\|^2}\sqrt{\sum_{i\in[t]}\left\langle UW^{-\frac{1}{2}}v^{(i)}, UW^{-\frac{1}{2}}v^{(i)}\right\rangle} \tag{163}$$

$$\leq \max_{\|U\|_F = 1}\|Q\|\sqrt{\sum_{i\in[t]}\|h^{(i)}\|^2}\sqrt{\left\langle U\sum_{i\in[t]}W^{-\frac{1}{2}}v^{(i)}(v^{(i)})^\top W^{-\frac{1}{2}}, U\right\rangle} \tag{164}$$

$$\leq \max_{\|U\|_F = 1}\|H\|_F\|U\|_F = \|H\|_F \tag{165}$$

where used the fact that $\langle AB, C\rangle = \langle A, CB^\top\rangle$ and $(U^*)^\top U^* = \mathbf{I}$.

Let $\mathcal{S}_F = \{U \in \mathbb{R}^{d\times r} \mid \|U\|_F = 1\}$ be the set of all real matrices of dimensions $d \times r$ with unit Frobenius norm. For $\epsilon \leq 1$, there exists an $\epsilon$-net, $N_\epsilon \subset \mathcal{S}_F$, of size $(1 + 2/\epsilon)^{dr}$ with respect to the Frobenius norm [49, Lemma 5.2]. That is for any $U' \in \mathcal{S}_F$, there exists some $U \in N_\epsilon$ such that $\|U' - U\|_F \leq \epsilon$.

Consider a $U \in N_\epsilon$, such that $\|U\|_F = 1$. Now we will prove with high-probability that $\left\langle(\mathcal{W}^{-\frac{1}{2}}\mathcal{H})(U^*Q)(U) - \mathcal{W}^{-\frac{1}{2}}(\sum_{i\in[t]}S^{(i)}U^*Qh^{(i)}(v^{(i)})^\top), U\right\rangle$ is small. Consider the the following

quadratic form

$$\left\langle \mathcal{W}^{-\frac{1}{2}}(\sum_{i\in[t]} S^{(i)}U^*Qh^{(i)}(v^{(i)})^\top), U\right\rangle = \left\langle \sum_{i\in[t]} S^{(i)}U^*Qh^{(i)}(v^{(i)})^\top W^{-\frac{1}{2}}, U\right\rangle \tag{166}$$

$$= \sum_{i\in[t]} \frac{1}{m} \sum_{j\in[m]} (x_j^{(i)})^\top (U^*Qh^{(i)}(v^{(i)})^\top W^{-\frac{1}{2}}U^\top)x_j^{(i)} \tag{167}$$

where $S^{(i)} = \frac{1}{m}\sum_{j\in[m]} x_j^{(i)}(x_j^{(i)})^\top$ and $x_j^{(i)} \sim \mathcal{N}(0, \mathbf{I}_{d\times d})$ are i.i.d. standard Gaussian random vectors and $W = \sum_{i\in[t]} v^{(i)}(v^{(i)})^\top$ is rank-$r$ matrix. We will use Hanson-Wright inequality (Lemma F.5) to prove that the above quadratic form concentrates around its mean. Notice that the the the expectation of $\left\langle \mathcal{W}^{-\frac{1}{2}}(\sum_{i\in[t]} S^{(i)}U^*Qh^{(i)}(v^{(i)})^\top), U\right\rangle$ is $\left\langle W^{-\frac{1}{2}}\mathcal{H}(U), U\right\rangle$.

$$\mathbb{E}[\mathcal{W}^{-\frac{1}{2}}(\sum_{i\in[t]} S^{(i)}U^*Qh^{(i)}(v^{(i)})^\top)] = \mathcal{W}^{-\frac{1}{2}}(\sum_{i\in[t]} U^*Qh^{(i)}(v^{(i)})^\top) = (\mathcal{W}^{-\frac{1}{2}}\mathcal{H})(U^*Q). \tag{168}$$

We will also need the following bounds to apply the Hanson-Wright inequality. Recall that $\|H\|_{\infty,2} = \max_{i\in[t]}\|h^{(i)}\|$ and $\|V\|_{\infty,2} = \max_{i\in[t]}\|v^{(i)}\|$. Then,

$$\max_{i\in[t]} \|U^*Qh^{(i)}(v^{(i)})^\top W^{-\frac{1}{2}}U^\top\| \le \max_{i\in[t]}\|U^*\|\|Q\|\|h^{(i)}\|\max_{i\in[t]}\frac{\|v^{(i)}\|}{\sqrt{\lambda_r(W)}}\|U\| \le \|H\|_{\infty,2}\frac{\|V\|_{\infty,2}}{\sqrt{\lambda_r(W)}} \tag{169}$$

Also note that

$$\sum_{i\in[t]} \|U^*Qh^{(i)}(v^{(i)})^\top W^{-\frac{1}{2}}U^\top\|_F^2 = \sum_{i\in[t]} \|U^*Qh^{(i)}\|^2\|UW^{-\frac{1}{2}}v^{(i)}\|^2 \tag{170}$$

$$\le (\sum_{i\in[t]}\|U^*Qh^{(i)}\|^2)(\max_{i\in[t]}\|UW^{-\frac{1}{2}}v^{(i)}\|^2) \tag{171}$$

$$\le (\|Q\|^2 \sum_{i\in[t]}\|h^{(i)}\|^2)(\max_{i\in[t]}\|U\|^2\|W^{-\frac{1}{2}}\|^2\|v^{(i)}\|^2) \tag{172}$$

$$\le \|H\|_F^2 \frac{\|V\|_{\infty,2}^2}{\lambda_r(W)} \tag{173}$$

and

$$\sum_{i\in[t]} \|U^*Qh^{(i)}(v^{(i)})^\top W^{-\frac{1}{2}}U^\top\|_F^2 = \sum_{i\in[t]} \|U^*Qh^{(i)}\|^2\|UW^{-\frac{1}{2}}v^{(i)}\|^2 \tag{174}$$

$$\le (\max_{i\in[t]}\|U^*Qh^{(i)}\|^2)\mathrm{tr}(UW^{-\frac{1}{2}}\sum_{i\in[t]} v^{(i)}(v^{(i)})^\top W^{-\frac{1}{2}}U^\top) \tag{175}$$

$$\le \|Q\|\max_{i\in[t]}\|h^{(i)}\|^2\|U\|_F^2 \tag{176}$$

$$= \|H\|_{\infty,2}^2. \tag{177}$$

Therefore, $\sum_{i\in[t]} \|U^*Qh^{(i)}(v^{(i)})^\top W^{-\frac{1}{2}}U^\top\|_F^2 \le \min\{\|H\|_F^2\frac{\|V\|_{\infty,2}^2}{\lambda_r(W)}, \|H\|_{\infty,2}^2\}$. For brevity, let $\mathcal{E}_2(U) = \mathcal{W}^{-\frac{1}{2}}(\sum_{i\in[t]} S^{(i)}Uh^{(i)}(v^{(i)})^\top) - (\mathcal{W}^{-\frac{1}{2}}\mathcal{H})(U)$. Then by Hanson-Wright inequality (Lemma F.5), with probability at least $1 - \delta/|N_\epsilon|$

$$|\langle\mathcal{E}_2(U^*Q), U\rangle| \tag{178}$$

$$= |\left\langle \sum_{i\in[t]} \frac{1}{m}\sum_{j\in[m]} x_j^{(i)}(x_j^{(i)})^\top U^*Qh^{(i)}(v^{(i)})^\top W^{-\frac{1}{2}}, U\right\rangle - \left\langle (\mathcal{W}^{-\frac{1}{2}}\mathcal{H})(U^*Q), U\right\rangle| \le \Delta_\epsilon \tag{179}$$

where $\Delta_\epsilon = c(\min(\|H\|_F \frac{\|V\|_{\infty,2}}{\sqrt{\lambda_r(W)}}, \|H\|_{\infty,2}) \sqrt{\frac{\log(|N_\epsilon|/\delta)}{m}} + \|H\|_{\infty,2} \frac{\|V\|_{\infty,2}}{\sqrt{\lambda_r(W)}} \frac{\log(|N_\epsilon|/\delta)}{m})$. Taking union bound over all $U \in N_\epsilon$ implies that with probability at least $1 - \delta$

$$\left|\langle \mathcal{E}_2(U), U\rangle\right| \leq \Delta_\epsilon \ , \ \text{ for all } U \in N_\epsilon . \tag{180}$$

Let $U' \in \mathcal{S}_F \subset \mathbb{R}^{d\times r}$ be the matrix "parallel" to $\mathcal{E}_2(U^*Q)$, that is $\|\mathcal{E}_2(U^*Q)\|_F = \max_{\|\widetilde{U}\|_F=1} \left\langle \mathcal{E}_1(U^*Q), \widetilde{U}\right\rangle = \langle \mathcal{E}_2(U^*Q), U'\rangle$. Then there exists some $U \in N_\epsilon$ such that $\|U' - U\|_F \leq \epsilon$.

$$\|\mathcal{E}_2(U^*Q)\|_F = \langle \mathcal{E}_2(U^*Q), U'\rangle = \langle \mathcal{E}_2(U^*Q), U\rangle + \langle \mathcal{E}_2(U^*Q), U' - U\rangle \tag{181}$$
$$\leq \langle \mathcal{E}_1(U), U\rangle + \|\mathcal{E}_2(U^*Q)\|_F \|U' - U\|_F \tag{182}$$
$$\leq \langle \mathcal{E}_1(U), U\rangle + \epsilon\|\mathcal{E}_2(U^*Q)\|_F \tag{183}$$

Re-arranging and setting $\epsilon = 1/2$, and $c \leftarrow 2c$, we get

$$\|\mathcal{E}_2(U^*Q)\|_F \leq \Delta_{\frac{1}{2}} = c(\min(\|H\|_F \frac{\|V\|_{\infty,2}}{\sqrt{\lambda_r(W)}}, \|H\|_{\infty,2}) \sqrt{\frac{dr \, \log(5/\delta)}{m}} +$$
$$\|H\|_{\infty,2} \frac{\|V\|_{\infty,2}}{\sqrt{\lambda_r(W)}} \frac{dr \, \log(5/\delta)}{m}) \tag{184}$$

Finally setting $\delta \leftarrow \delta/3$ get us the desired result.

$\square$

**Lemma B.9.** *With a probability of at least* $1 - \delta/3$

$$\|\sum_{i\in[t]} \mathcal{W}^{-\frac{1}{2}}(z^{(i)}(v^{(i)})^\top))\|_F \leq O\left(\sigma\sqrt{\frac{dr}{m} \log\left(\frac{t}{\delta}\right) \log\left(\frac{r}{\delta}\right)}\right) \tag{185}$$

*Proof of Lemma B.9.* Notice that $z^{(i)}$ (defined in Appendix B) is a Gaussian random vector of the following form

$$z^{(i)} = \frac{1}{m} \sum_{j\in[m]} \varepsilon_j^{(i)} x_j^{(i)} = \frac{1}{m}\|\varepsilon^{(i)}\|g^{(i)}, g^{(i)} \sim \mathcal{N}(0, \mathbf{I}_{d\times d}) \tag{186}$$

Using Hanson-Wright inequality (Lemma F.5, by setting $m \leftarrow 1$, $x_1 \leftarrow \varepsilon^{(i)}$, and $A_1 \leftarrow \mathbf{I}_{m\times m}$) and taking union bound over all tasks, we get that, with probability of at least $1 - \frac{\delta}{2}$

$$\|\varepsilon^{(i)}\|^2 \leq \sigma^2 m(1 + c\sqrt{\frac{\log(\frac{2t}{\delta})}{m}} + c\frac{\log(\frac{2t}{\delta})}{m}) \leq 2c\,\sigma^2 m \log\left(\frac{2t}{\delta}\right), \ \text{ for all } i \in [t] \tag{187}$$

where used the fact that $m \geq 1$ and $\log\left(\frac{2t}{\delta}\right) \geq 1$.

Let $\widehat{v}^{(i)} = W^{-\frac{1}{2}}v^{(i)}$, then
$$\sum_{i\in[t]} \|\widehat{v}^{(i)}\|^2 = \sum_{i\in[t]} \text{tr}((v^{(i)})^\top W^{-1} v^{(i)}) = \sum_{i\in[t]} \text{tr}(W^{-1} v^{(i)}(v^{(i)})^\top) = r \tag{188}$$

Notice that $\sum_{i\in[t]} \frac{1}{m}\|\varepsilon^{(i)}\|g^{(i)}\widehat{v}_j^{(i)}$ is a Gaussian random vector of the following form

$$\sum_{i\in[t]} \frac{1}{m}\|\varepsilon^{(i)}\|g^{(i)}\widehat{v}_j^{(i)} = \frac{1}{m} \sqrt{\sum_{i\in[t]} \|\varepsilon^{(i)}\|^2(\widehat{v}_j^{(i)})^2} \ \widehat{g}_j \, , \widehat{g}_j \sim \mathcal{N}(0, \mathbf{I}_{d\times d}) \tag{189}$$

Using Hanson-Wright inequality (Lemma F.5, by setting $m \leftarrow 1$, $x_1 \leftarrow \widehat{g}_j$, and $A_1 \leftarrow \mathbf{I}_{d\times d}$) and taking union bound over all $j \in [r]$, we get that, with probability of at least $1 - \frac{\delta}{2}$

$$\|\widehat{g}_j\|^2 \leq d(1 + c\sqrt{\frac{\log(\frac{2r}{\delta})}{d}} + c\frac{\log(\frac{2r}{\delta})}{d}) \leq 2cd \log\left(\frac{2r}{\delta}\right), \ \text{ for all } j \in [r] \tag{190}$$

where used the fact that $d \geq 1$ and $\log\left(\frac{2r}{\delta}\right) \geq 1$.

Combining the above results and using union bound, we get that, with a probability of at least $1 - \delta$,

$$\left\| \sum_{i \in [t]} \mathcal{W}^{-\frac{1}{2}}(z^{(i)}(v^{(i)})^\top)) \right\|_F^2 = \left\| \sum_{i \in [t]} z^{(i)}(v^{(i)})^\top W^{-\frac{1}{2}} \right\|_F^2 \tag{191}$$

$$= \left\| \sum_{i \in [t]} \frac{1}{m} \|\varepsilon^{(i)}\| g^{(i)}(\widehat{v}^{(i)})^\top \right\|_F^2 \tag{192}$$

$$= \sum_{j \in [r]} \left\| \sum_{i \in [t]} \frac{1}{m} \|\varepsilon^{(i)}\| g^{(i)}\widehat{v}_j^{(i)} \right\|^2 \tag{193}$$

$$\leq \sum_{j \in [r]} \sum_{i \in [t]} \frac{\|\varepsilon^{(i)}\|^2}{m^2}(\widehat{v}_j^{(i)})^2 \|\widehat{g}_j\|^2 \tag{194}$$

$$\leq \sum_{j \in [r]} \sum_{i \in [t]} O\left(\frac{m\sigma^2}{m^2}\log\left(\frac{t}{\delta}\right)\right)(\widehat{v}_j^{(i)})^2 O\left(d\log\left(\frac{r}{\delta}\right)\right) \tag{195}$$

$$\leq O\left(\frac{d\sigma^2}{m}\log\left(\frac{t}{\delta}\right)\log\left(\frac{r}{\delta}\right)\right) \sum_{i \in [t]} \|\widehat{v}^{(i)}\|^2 \tag{196}$$

$$\leq O\left(\frac{\sigma^2 dr}{m}\log\left(\frac{t}{\delta}\right)\log\left(\frac{r}{\delta}\right)\right). \tag{197}$$

Finally, we get the desired result by setting $\delta \leftarrow \delta/3$. $\qquad\square$

## B.4 Analysis of QR decomposition

*Proof of Lemma B.4.*

$$\sigma_{\min}(R) \geq \min_{\|z\|=1} \|Rz\| = \min_{\|z\|=1} \|U^+Rz\| = \min_{\|z\|=1} \|\widehat{U}z\| \tag{198}$$

$$\geq \min_{\|z\|=1} \|(U^*Q - \mathcal{W}^\dagger \mathcal{H}(U^*Q) + F)z\| \tag{199}$$

$$\geq \min_{\|z\|=1} \sqrt{z^\top Q^\top Qz} - \|\mathcal{W}^\dagger \mathcal{H}(U^*Q)\| - \|F\| \tag{200}$$

$$\geq \min_{\|z\|=1} \sigma_{\min}(Q) - \|\mathcal{W}^\dagger \mathcal{H}(U^*Q)\| - \|F\| \tag{201}$$

$$\geq \frac{1}{2} - \frac{1}{8} - \frac{1}{8} \geq \frac{1}{4} \tag{202}$$

There fore $R$ is invertible and $\|R^{-1}\| = (\sigma_{\min}(R))^{-1} \leq 4$ $\qquad\square$

## B.5 Analysis of shuffling and partitioning

*Proof of Lemma B.1.* We will assume that the set of tasks $[t]$ is shuffled. We will prove that incoherence holds for the all subset $\mathcal{T}_k = [1 + \frac{t(k-1)}{K}, \frac{tk}{K}]$ of size $t/K$. Shuffling and $K$-way partitioning to get $\mathcal{T}_k$ is equivalent to uniformly sampling without replacement $t/K$ elements from $[t]$. We prove that incoherrence holds for the first subset $\mathcal{T}_1$, then this is equivalent to proving that incoherence holds for the $k$-th partition $\mathcal{T}_k$ by symmetry. Let the tasks sampled for $\mathcal{T}_1$ without replacement be $\{i_l\}_{l=1}^{t/k}$, where $i_l$ is the $l$-th sample.

Let $\mathcal{S}_F = \{z \in \mathbb{R}^r \mid \|z\| = 1\}$ be the set of all real vectors of dimensions $r$ with unit Euclidean norm. For $\epsilon \leq 1$, there exists an $\epsilon$-net, $N_\epsilon \subset \mathcal{S}_F$, of size $(1 + 2/\epsilon)^r$ with respect to the Euclidean norm [49, Lemma 5.2]. That is for any $z' \in \mathcal{S}_F$, there exists some $z \in N_\epsilon$ such that $\|z' - z\| \leq \epsilon$.

Consider a $z \in N_\epsilon$, such that $\|z\| = 1$. Now we will prove with high-probability that $z^\top(\sum_{l=1}^{t/K} v^{*(i_l)}(v^{*(i_l)})^\top)z$ is approximately equal to $z^\top \mathbb{E}[\sum_{l=1}^{t/K} v^{*(i_l)}(v^{*(i_l)})^\top]z$. Now consider the martingale $X_l$, such that $X_0 = 0$ and $X_l = X_{l-1} + z^\top(v^{*(i_l)}(v^{*(i_l)})^\top -$

$\mathbb{E}[v^{*(i_l)}(v^{*(i_l)})^\top | X_0, \ldots, X_{l-1}])z$, for all $l \in [t/K]$. Clearly this is a martginagle as $\mathbb{E}[X_l | X_0, \ldots, X_{l-1}] = 0$, for all $l \in [t/K]$. The maximum difference two consecutive steps is $\max_l |X_l - X_{l-1}| \leq 2\|v^{*(i_l)}\|^2 \leq 2\|V^*\|_{\infty,2}^2$. Therefore by Azuma-Hoeffding martingale inequality,

$$|\sum_{l=1}^{t/K} z^\top v^{*(i_l)}(v^{*(i_l)})^\top z - z^\top \mathbb{E}[\sum_{l=1}^{t/K} v^{*(i_l)}(v^{*(i_l)})^\top]z| = |X_{t/K}| \leq \sqrt{\frac{2t}{K}\|V\|_{\infty,2}^4 \log(\frac{2|N_\epsilon|}{\delta})}$$

(203)

with a probability of at least $1 - \delta/|N_\epsilon|$.

For brevity, let $E = \sum_{l=1}^{t/K} v^{*(i_l)}(v^{*(i_l)})^\top - \mathbb{E}[\sum_{l=1}^{t/K} v^{*(i_l)}(v^{*(i_l)})^\top]$. Notice that $E$ is a real symmetric matrix, therefore it has an eigen decomposition. Then, let $v' \in \mathcal{S} \subset \mathbb{R}^r$ be the largest "eigenvector" of $E$, such that $(v')^\top E v' = \|E\| = \max_{\|\widetilde{v}\|=1} \widetilde{v}^\top E \widetilde{v} = \max_{\|\widetilde{v}\|=\|\widetilde{v}'\|_F=1} \widetilde{v}^\top E \widetilde{v}'$. Then there exists some $v \in N_\epsilon$ such that $\|v' - v\| \leq \epsilon$.

$$\|E\|_F = (v')^\top E v = v^\top E v + (v'-v)^\top E v + (v')^\top E(v'-v) \qquad (204)$$

$$\leq v^\top E v + \|v'-v\|\|E\|\|v\| + \|v'\|\|E\|\|v'-v\| \qquad (205)$$

$$\leq v^\top E v + 2\epsilon\|E\| \qquad (206)$$

Re-arranging and setting $\epsilon = 1/4$, and $c \leftarrow 2c$, we get

$$\|\sum_{l=1}^{t/K} v^{*(i_l)}(v^{*(i_l)})^\top - \mathbb{E}[\sum_{l=1}^{t/K} v^{*(i_l)}(v^{*(i_l)})^\top]\| = \|E\| \leq \sqrt{\frac{2tr}{K}\|V\|_{\infty,2}^4 \log(\frac{18}{\delta})} \qquad (207)$$

$$\leq \frac{1}{2}\lambda_r(\mathbb{E}[\sum_{l=1}^{t/K} v^{*(i_l)}(v^{*(i_l)})^\top]). \qquad (208)$$

with probability at least $1 - \delta/k$, where the last inequality used the fact that $t \geq \Omega(\mu^2 r^3 K \log(1/\delta))$. Additionally note that $\mathbb{E}[\sum_{l=1}^{t/k} v^{*(i_l)}(v^{*(i_l)})^\top] = \frac{1}{K}\sum_{i=1}^{t} v^{*(i)}(v^{*(i)})^\top = \frac{1}{K}(V^*)^\top V^*$, Therefore

$$\lambda_{r'}(\sum_{i\in\mathcal{T}_k} v^{*(i)}(v^{*(i)})^\top) = \frac{1}{K}\Theta(\lambda_{r'}((V^*)^\top V^*)) \text{ for all } r' \in [r] \qquad (209)$$

where $\lambda_i(\cdot)$ is the $r'$-th largest eigenvalue matrix operator.

$\square$

## C   Analysis of AltMinGD (Algorithm 1)

Initialized at $U$, the $k$-the step of alternating minimization-based AltMin (Algorithm 1) is:

$$v^{(i)} \leftarrow (U^\top S_1^{(i)} U)^\dagger((U^\top S_1^{(i)} U^*)v^{*(i)} + U^\top z^{(i)}), \text{ for } i \in \mathcal{T}_k = [1 + \frac{(k-1)t}{K}, \frac{tk}{K}] \quad (210)$$

$$\widetilde{U} \leftarrow U - \eta\Big(\sum_{i\in[t]} S_2^{(i)}(Uv^{(i)} - U^*v^{*(i)})(v^{(i)})^\top + z^{(i)}(v^{(i)})^\top\Big), \qquad (211)$$

$$U^+ \leftarrow \text{QR}(\widetilde{U}), \qquad (212)$$

where $U^+$ is the next iterate, and $S_1^{(i)} = \frac{2}{m}\sum_{j\in[1,m/2]} x_j^{(i)}(x_j^{(i)})^\top$, $S_2^{(i)} = \frac{2}{m}\sum_{j\in[1+m/2,m]} x_j^{(i)}(x_j^{(i)})^\top$, and $z^{(i)} \triangleq (1/m)\sum_{j\in[m]} \varepsilon_j^{(i)} x_j^{(i)}$ and $\mathcal{A} : \mathbb{R}^{d\times r} \to \mathbb{R}^{d\times r}$ is a self-adjoint linear operator such that $\mathcal{A}(U) = \sum_{i\in T} S^{(i)} U v^{(i)}(v^{(i)})^\top$. The self-adjointness of $\mathcal{A}$ follows from the symmetry of $S^{(i)}$ when using cyclic property of trace as follows

$$\langle U_2, \mathcal{A}(U_1)\rangle = \sum_{i\in T}\langle U_2, S^{(i)}U_1 v^{(i)}(v^{(i)})^\top)\rangle = \sum_{i\in T}\text{tr}(U_2^\top S^{(i)}U_1 v^{(i)}(v^{(i)})^\top)$$

$$= \sum_{i\in T}\text{tr}(v^{(i)}(v^{(i)})^\top U_2^\top S^{(i)}U_1) = \langle\mathcal{A}(U_2), U_1\rangle$$

(213)

QR-decomposition after every update is required to ensure that magnitude of $U$ and $V$ does not stray far away from that of true $U^*$ and $V^*$, respectively. Otherwise, the sample complexity requirements of our algorithm increase in the condition number factors.

**Incoherence.** $\max_i \|v^{*(i)}\|^2 \leq (\mu\,r/t)\lambda_r(\sum_{i\in[t]} v^{*(i)}(v^{*(i)})^\top)$, and we define $\nu = (1/t)\lambda_r(\sum_{i\in[t]} v^{*(i)}(v^{*(i)})^\top)$. Notice that, this non-standard definition of incoherence is related to the standard definition: $W^* = (V^*)^\top V^* = \sum_{i\in[t]} v^{*(i)}(v^{*(i)})^\top$, $V^* = \tilde{V}^* R^*$ (QR-decomposition), $\max_i \|\widetilde{v}^{*(i)}\|^2 \leq \widetilde{\mu}\,r/t$, as follows $\mu = \widehat{\mu}(\sigma_1^2(R^*)/\sigma_r^2(R^*))$.

**Theorem 9** (Formal version of Theorem 1 in Section 4)**.** *Let there be $t$ linear regression tasks, each with $m$ samples satisfying Assumptions 1 and 2, and number of iterations $K = \Omega(\lceil \frac{\lambda_1^*}{\lambda_r^*}\log(\frac{mt}{(\lambda_1^*/\lambda_r^*)(\sigma/\sqrt{\lambda_r^*})\mu dr})\rceil)$, $\|(\mathbf{I} - U^*(U^*)^\top)U_{\text{init}}\|_F \leq \min\left(\frac{21}{121}, O\left(\frac{\lambda_r^*}{\lambda_1^*}\sqrt{\frac{1}{\log(t/K)}}\right)\right)$, $m \geq \Omega\left((1 + r\frac{\lambda_1^*}{\lambda_r^*}\left(\frac{\sigma}{\sqrt{\lambda_r^*}}\right)^2)r\log(\frac{t}{\delta}) + r^2\log(\frac{K}{\delta})\right)$, $t \geq \Omega(\mu^2 r^3 K \log(\frac{K}{\delta}))$, and $mt \geq \Omega\left(\mu dr^2 K \log(\frac{t}{\delta})\left(1 + \left(\frac{\lambda_1^*}{\lambda_r^*}\right)^2\left(\frac{\sigma}{\sqrt{\lambda_r^*}}\right)^2\log(\frac{t}{\delta})\log(\frac{rK}{\delta})\right)\right)$. Then, for any $0 < \delta < 1$, after $K$ iterations and using the stepsize $\eta = (r/t)/2\lambda_1^*$, AltMinGD (Algorithm 1) returns an orthonormal matrix $U \in \mathbb{R}^{d\times r}$, such that with a probability of at least $1 - \delta$*

$$\frac{1}{\sqrt{r}}\|(\mathbf{I} - U^*(U^*)^\top)U\|_F \leq O\left(\frac{\sigma}{\sqrt{\lambda_r^*}}\sqrt{\frac{\mu dr^2 K \log(\frac{t}{\delta})\log(\frac{rK}{\delta})}{mt}}\right) \tag{214}$$

A proof is in Section C.1.

**Initialization.** If we initialize AltMinGD (Algorithm 1) with Method-of-Moments (Theorem 12), we need at least

$$mt \geq \widetilde{\Omega}\left(\left(\frac{\lambda_1^*}{\lambda_r^*}\right)^3 \mu dr^2 + \left(\frac{\sigma}{\sqrt{\lambda_r^*}}\right)^4\left(\frac{\lambda_1^*}{\lambda_r^*}\right)^2 dr^3\right) \tag{215}$$

initial number of samples, where $\widetilde{\Omega}$ hides $\mathrm{polylog}$ factors.

### C.1 Proof of Theorem 9 (formal version of Theorem 1 in Section 4)

**Proof sketch:** We first prove that distance between $U^*$ and $U$ decreases at each iteration up to some additional noise terms. Then this per iterate result is unrolled to obtained the final guarantees.

First we focus on the $k$-th iterate. In this analysis, unless specified $[t]$, represents the $k$-th $K$-way partition used for the $k$-th iterate. Same result as Lemma B.1 for AltMin (Algorithm 3), holds for AltMinGD too. Therefore the tasks subset used for each iteration, satisfy approximate incoherence.

In the analysis of an iterate we denote the current iterate using $U$ and the next iterate using $U^+$. First we prove that the distance between the true $v^{*(i)}$ and the current $v^{(i)}$ is approximately upper-bounded by multiple of distance between $U$ and $U^*$. Next we prove that distance between $U^+$ and $U^*$ is approximately a fraction of the distance between $v^{*(i)}$ and $v^{(i)}$. Finally, combining the above two results gives us desired result.

**Preliminaries:** Let $Q = (U^*)^\top U$. Using Lemma F.4, if $\|U - U^*(U^*)^\top U\|_F < 1$, $Q$ is invertible. Let $Q^{-1}$ be the right inverse of $Q$, i.e. $QQ^{-1} = \mathbf{I}$. Let $W = (V^*)^\top V^* = \sum_{i\in[t]} v^{*(i)}(v^{*(i)})^\top$, then using Assumption 2 we have that $\lambda_1^* = (r/t)\max_{\|z\|=1} z^\top W^* z$ and $\lambda_r^* = (r/t)\min_{\|z\|=1} z^\top W^* z$.

**Update on $V$:** Let $h^{(i)} = v^{(i)} - Q^{-1}v^{*(i)}$ and $H^T = [h^{(1)}h^{(2)}\ldots h^{(t)}]$. Let $\|H\|_F \triangleq \sqrt{\sum_{i\in[t]}\|h^{(i)}\|^2}$ and $\|H\|_{\infty,2} \triangleq \max_{i\in[t]}\|h^{(i)}\|$. Let $W = V^\top V = \sum_{i\in[t]} v^{(i)}(v^{(i)})^\top$, and $\lambda_1 = (r/t)\max_{\|z\|=1} z^\top W z$ and $\lambda_r = (r/t)\min_{\|z\|=1} z^\top W z$.

Same result as Lemma B.2 for AltMin (Algorithm 3), holds for AltMinGD too. Therefore $V$ update of AltMinGD satisfies Lemma B.2.

**Update on** $U$**:** Let $\mathcal{W}, \mathcal{H}, \widehat{\mathcal{H}} : \mathbb{R}^{d \times r} \to \mathbb{R}^{d \times r}$ be three linear operators, such that $\mathcal{W}(U) = U \sum_{i \in \mathcal{T}_k} v^{(i)}(v^{(i)})^\top = UW, \mathcal{H}(U) = U \sum_{i \in \mathcal{T}_k} h^{(i)}(v^{(i)})^\top$ and $\widehat{\mathcal{H}}(U) = \sum_{i \in \mathcal{T}_k} S_2^{(i)} U h^{(i)}(v^{(i)})^\top$, where $h^{(i)} = v^{(i)} - Q^{-1}v^{*(i)}$. $\mathcal{W}$ is invertible and self-adjoint. Therefore $\mathcal{W}^{-\frac{1}{2}}$ and $\mathcal{W}^{\frac{1}{2}}$ exist. Let $\mathcal{I} : \mathbb{R}^{d \times r} \to \mathbb{R}^{d \times r}$ be the identity mapping, such that $\mathcal{I}(U) = U$.

$$\widetilde{U} - U^*Q$$

$$= U - U^*Q - \eta \Big( \sum_{i \in \mathcal{T}_k} S_2^{(i)}(Uv^{(i)} - U^*v^{*(i)})(v^{(i)})^\top + z^{(i)}(v^{(i)})^\top \Big)$$

$$= U - U^*Q - \eta \big( \sum_{i \in \mathcal{T}_k} S_2^{(i)}(U - U^*Q)v^{(i)}(v^{(i)})^\top - S_2^{(i)}U^*Q(Q^{-1}v^{*(i)} - v^{(i)})(v^{(i)})^\top + z^{(i)}(v^{(i)})^\top \big)$$

$$= (\mathcal{I} - \eta\mathcal{A})(U - U^*Q) + \eta \big( -\widehat{\mathcal{H}}(U^*Q) + \sum_{i \in \mathcal{T}_k} z^{(i)}(v^{(i)})^\top \big)$$

$$= (\mathcal{I} - \eta\mathcal{W})(U - U^*Q) + \eta\mathcal{E}_1(U - U^*Q) + \eta(-\mathcal{H} + \mathcal{E}_2)(U^*Q) + \eta \sum_{i \in \mathcal{T}_k} z^{(i)}(v^{(i)})^\top \quad (216)$$

where $\mathcal{E}_1 = \mathcal{A} - \mathcal{W}$ and $\mathcal{E}_2 = \widehat{\mathcal{H}} - \mathcal{H}$. Let $F = \widetilde{U} - U^*Q + \eta\mathcal{H}(U^*Q)$

**Lemma C.1.** *Assume that all conditions and the large probability event in Lemma B.2 holds true. Then,*

$$\|\mathcal{H}(U^*Q)\|_F \leq \lambda_r(W) O\Big( \frac{\lambda_1^*}{\lambda_r^*} \sqrt{\frac{\log(\frac{t}{K\delta})}{\log(\frac{1}{\delta})}} \|(\mathbf{I} - U^*(U^*)^\top)U\|_F + \frac{\sigma}{\sqrt{\lambda_r^*}} \sqrt{\frac{\lambda_1^*}{\lambda_r^*} \frac{r^2 \log(\frac{t}{K\delta})}{m}} \Big)$$

(217)

*and if $mt \geq \Omega(\mu dr^2 K \log(t/K\delta))$, $m \geq \Omega(r^2 \log(1/\delta))$, and $\eta \leq \frac{1}{\lambda_1(W)}$, then with probability at least $1 - \delta/3$*

$$\|F\|_F \leq (1 - \frac{\eta}{2}\lambda_r(W))\|(\mathbf{I} - U^*(U^*)^\top)U\|_F + \eta\lambda_r(W) O\Big( \frac{\sigma}{\sqrt{\lambda_r^*}} \sqrt{\frac{\mu dr^2 K \log(\frac{t}{K\delta}) \log(\frac{r}{\delta})}{mt}} \Big)$$

(218)

A proof is in Section C.2.1.

**Lemma C.2.** *If $\eta \leq \frac{1}{\lambda_1(W)}$, $1 - \frac{1}{21}\eta\lambda_r(W) \leq \sigma_{\min}(Q)$, $\|F\|_F \leq \frac{1}{21}\eta\lambda_r(W)$ and $\eta\|\mathcal{H}(U^*Q)\|_F \leq \frac{1}{21}\eta\lambda_r(W)$, then $R$ is invertible and $\|R^{-1}\| \leq (1 + \frac{1}{6}\eta\lambda_r(W)) \leq \frac{7}{6}$.*

A proof is in Section C.3. Clearly, from (217) and (218), a sufficient condition for the above lemma is

$$1 - \frac{1}{21}\eta\lambda_r(W) \leq \sigma_{\min}(Q) \quad (219)$$

$$\eta\lambda_r(W) O\Big( \frac{\lambda_1^*}{\lambda_r^*} \sqrt{\frac{\log(\frac{t}{K\delta})}{\log(\frac{1}{\delta})}} \|(\mathbf{I} - U^*(U^*)^\top)U\|_F + \frac{\sigma}{\sqrt{\lambda_r^*}} \sqrt{\frac{\lambda_1^*}{\lambda_r^*} \frac{r^2 \log(\frac{t}{K\delta})}{m}} \Big) \leq \frac{1}{21}\eta\lambda_r(W) \text{ , and}$$

(220)

$$(1 - \frac{\eta}{2}\lambda_r(W))\|(\mathbf{I} - U^*(U^*)^\top)U\|_F + \eta\lambda_r(W) O\Big( \frac{\sigma}{\sqrt{\lambda_r^*}} \sqrt{\frac{\mu dr^2 K \log(\frac{t}{K\delta}) \log(\frac{r}{\delta})}{mt}} \Big) \leq \frac{1}{21}\eta\lambda_r(W)$$

(221)

which can be satisfied with

$$\|(\mathbf{I} - U^*(U^*)^\top)U\|_F \leq \min\Big( O(\eta\lambda_r(W)), O\Big(\frac{\lambda_r^*}{\lambda_1^*}\sqrt{\frac{1}{\log(t/K)}}\Big) \Big), \quad (222)$$

$$m \geq \Omega\Big(\frac{\lambda_1^*}{\lambda_r^*}\Big(\frac{\sigma}{\sqrt{\lambda_r^*}}\Big)^2 r^2 \log\Big(\frac{t}{K\delta}\Big)\Big) \text{ , and} \quad (223)$$

$$mt \geq \Omega\Big(\mu dr^2 K \Big(\frac{\sigma}{\sqrt{\lambda_r^*}}\Big)^2 \log\Big(\frac{t}{K\delta}\Big) \log\Big(\frac{r}{\delta}\Big)\Big) \quad (224)$$

Finally, we bound the Frobenius norm distance of the next iterate $U^+$ from the optimal $U^*$.

$$
\begin{aligned}
\|(\mathbf{I} - U^*(U^*)^\top)U^+\|_F &= \min_{Q^+} \|U^+ - U^* Q^+\|_F \\
&\leq \|\widetilde{U}R^{-1} - U^* Q R^{-1} + \eta(\mathcal{H}(U^* Q))R^{-1}\|_F \\
&\leq \|\widetilde{U} - U^* Q + \eta\mathcal{H}(U^* Q)\|_F \|R^{-1}\| \\
&= \|F\|_F \|R^{-1}\| \\
&\leq (1 - \frac{\eta}{2}\lambda_r(W))(1 + \frac{\eta}{6}\lambda_r(W))\|(\mathbf{I} - U^*(U^*)^\top)U\|_F + \\
&\quad \frac{7}{6}\eta\lambda_r(W)O\Big(\frac{\sigma}{\sqrt{\lambda_r^*}}\sqrt{\frac{\mu d r^2 K \log(\frac{t}{K\delta})\log(\frac{r}{\delta})}{mt}}\Big) \\
&\leq (1 - \frac{\eta}{3}\lambda_r(W))\|(\mathbf{I} - U^*(U^*)^\top)U\|_F + \\
&\quad \eta\lambda_r(W)O\Big(\frac{\sigma}{\sqrt{\lambda_r^*}}\sqrt{\frac{\mu d r^2 K \log(\frac{t}{K\delta})\log(\frac{r}{\delta})}{mt}}\Big)
\end{aligned}
\tag{225}
$$

Finally setting $\eta = \frac{1}{2\lambda_1(W^*)} \leq \frac{1}{\lambda_1(W)}$, we get

$$
\begin{aligned}
\|(\mathbf{I} - U^*(U^*)^\top)U^+\|_F &\leq (1 - \frac{\lambda_r}{6\lambda_1^*})\|(\mathbf{I} - U^*(U^*)^\top)U\|_F + \\
&\quad \frac{\lambda_r}{6\lambda_1^*}O\Big(\frac{\sigma}{\sqrt{\lambda_r^*}}\sqrt{\frac{\mu d r^2 K \log(\frac{t}{K\delta})\log(\frac{r}{\delta})}{mt}}\Big)
\end{aligned}
\tag{226}
$$

If

$$
mt \geq \Omega\Big(\mu d r^2 K\Big(\frac{\sigma}{\sqrt{\lambda_r^*}}\Big)^2 \log(\frac{t}{K\delta})\log(\frac{r}{\delta})\Big(1 + \Big(\frac{\lambda_1^*}{\lambda_r^*}\Big)^2\log(\frac{t}{K\delta})\Big)\Big)
\tag{227}
$$

then,

$$
\|(\mathbf{I} - U^*(U^*)^\top)U^+\|_F \leq (1 - \frac{1}{6}\frac{\lambda_r}{\lambda_1^*})\|(\mathbf{I} - U^*(U^*)^\top)U\|_F + \frac{1}{6}\frac{\lambda_r}{\lambda_1^*}\min\Big(\frac{21}{121}, O\Big(\frac{\lambda_r^*}{\lambda_1^*}\sqrt{\frac{1}{\log(t/K)}}\Big)\Big)
\tag{228}
$$

Thus if $\|(\mathbf{I} - U^*(U^*)^\top)U\|_F \leq \min\Big(\frac{21}{121}, O\Big(\frac{\lambda_r^*}{\lambda_1^*}\sqrt{\frac{1}{\log(t/K)}}\Big)\Big)$, then $\|(\mathbf{I} - U^*(U^*)^\top)U^+\|_F \leq \min\Big(\frac{21}{121}, O\Big(\frac{\lambda_r^*}{\lambda_1^*}\sqrt{\frac{1}{\log(t/K)}}\Big)\Big)$.

Therefore, using Lemma B.1 (which requires that $t \geq \Omega(\mu^2 r^3 K \log(\frac{K}{\delta}))$) and union-bound and $\lambda_r^*/2 \leq \lambda_r \leq 2\lambda_r^*$ (Lemma B.2), we can un-roll the relation, between current iterate $U$ and the next iterate $U^+$, over $K$ iterations, starting from $U_{\text{init}}$ and ending at some $U$ iterations, to get

$$
\begin{aligned}
\|(\mathbf{I} - U^*(U^*)^\top)U\|_F &\leq (1 - \frac{1}{12}\frac{\lambda_r^*}{\lambda_1^*})^K\|(\mathbf{I} - U^*(U^*)^\top)U_{\text{init}}\|_F + \\
&\quad O\Big(\frac{\sigma}{\sqrt{\lambda_r^*}}\sqrt{\frac{\mu d r^2 K \log(\frac{t}{K\delta})\log(\frac{r}{\delta})}{mt}}\Big)
\end{aligned}
\tag{229}
$$

with probability at least $1 - K\delta$. Finally setting the number of iterations as $K := \Theta(\lceil\frac{\lambda_1^*}{\lambda_r^*}\log(\frac{mt}{(\lambda_1^*/\lambda_r^*)(\sigma/\sqrt{\lambda_r^*})\mu d r})\rceil)$ we get that, with a probability of at least $1 - K\delta$

$$
\|(\mathbf{I} - U^*(U^*)^\top)U\|_F \leq O\Big(\frac{\sigma}{\sqrt{\lambda_r^*}}\sqrt{\frac{\mu d r^2 K \log(\frac{t}{K\delta})\log(\frac{r}{\delta})}{mt}}\Big)
\tag{230}
$$

## C.2 Analysis of update on $U$

### C.2.1 Proof of Lemma C.1

*Proof of Lemma C.1.* In this proof for brevity, we will first set that $\mathcal{T}_k \leftarrow [t]$, $|\mathcal{T}_k| = t/K \leftarrow t$, $S_2^{(i)} \leftarrow S^{(i)} = \frac{1}{m}\sum_{j\in[m]} x_j^{(i)}(x_j^{(i)})^\top$. This can be done due to the approximate equivalence of the subset $\mathcal{T}_k$ and the set of all tasks $[t]$ by Lemma B.1, which requires that $t \geq \Omega(\mu^2 r^3 K \log(\frac{K}{\delta}))$. Finally at the end of the analysis we will reset $\mathcal{T}_k \leftarrow \mathcal{T}_k$, $|\mathcal{T}_k| = t/K \leftarrow t/K$, $S_2^{(i)} \leftarrow S_2^{(i)} = \frac{2}{m}\sum_{j\in[m/2+1,m]} x_j^{(i)}(x_j^{(i)})^\top$.

Recall that

$$\widetilde{U} - U^*Q = (\mathcal{I} - \eta\mathcal{W})(U - U^*Q) + \eta\mathcal{E}_1(U - U^*Q) +$$
$$\eta(-\mathcal{H} + \mathcal{E}_2)(U^*Q) + \eta\sum_{i\in\mathcal{T}_k} z^{(i)}(v^{(i)})^\top \tag{231}$$

where $\mathcal{E}_1 = \mathcal{A} - \mathcal{W}$ and $\mathcal{E}_2 = \mathcal{W}^{-\frac{1}{2}}\widehat{\mathcal{H}} - \mathcal{W}^{-\frac{1}{2}}\mathcal{H}$, and $F = \widehat{U} - U^*Q + \eta\mathcal{H}(U^*Q)$. Assume that $0 < 1 - \eta\lambda_1(W) < 1 - \eta\lambda_r(W)$. Therefore

$$\|F\|_F \leq (1 - \eta\lambda_r(W) + \eta\|\mathcal{E}_1\|_F)\|U - U^*Q\|_F +$$
$$\eta(\|\mathcal{E}_2(U^*Q)\|_F + \|\sum_{i\in[t]} z^{(i)}(v^{(i)})^\top\|_F) \tag{232}$$

$\Omega(\mu dr^2 \log(1/\delta)) \leq mt$ and approximate incoherence of intermediate $V$ (26) implies that $\Omega(dr\frac{\|V\|_{\infty,2}^2}{\lambda_r(W)/t}\log(1/\delta)) \leq \Omega(\mu dr^2 \log(1/\delta)) \leq mt$, then by Lemma C.3 we have that, with a probability of at least $1 - \delta/3$

$$\|\mathcal{E}_1\|_F \leq \lambda_r(W)O\left(\sqrt{\frac{\lambda_1^*}{\lambda_r^*}\frac{\mu dr^2 \log(27/\delta)}{mt}}\right) \tag{233}$$

By Lemma C.4,

$$\|\mathcal{H}(U^*Q)\|_F \leq \sqrt{\lambda_1(W)}\|H\|_F \tag{234}$$

and with a probability of at least $1 - \delta/3$

$$\|\mathcal{E}_2(U^*Q)\|_F$$
$$\leq c(\min(\|H\|_F\|V\|_{\infty,2}, \|H\|_{\infty,2}\sqrt{\lambda_1(W)})\sqrt{\frac{dr \log(5/\delta)}{m}} + \|H\|_{\infty,2}\|V\|_{\infty,2}\frac{dr \log(5/\delta)}{m}) \tag{235}$$

Using the approximate incoherence of $V$ (26) in the above inequality, we get that

$$\|\mathcal{E}_2(U^*Q)\|_F \leq \sqrt{\lambda_r(W)}O(\min(\|H\|_F\sqrt{\frac{\mu r}{t}}, \|H\|_{\infty,2}\sqrt{\frac{\lambda_1^*}{\lambda_r^*}})\sqrt{\frac{dr \log(15/\delta)}{m}} +$$
$$\|H\|_{\infty,2}\sqrt{\frac{\mu r}{t}} \cdot \frac{dr \log(15/\delta)}{m}) \tag{236}$$

By Lemma C.5 with a probability of at least $1 - \delta/3$

$$\|\sum_{i\in[t]} z^{(i)}(v^{(i)})^\top\|_F \leq O\left(\sigma\sqrt{\frac{d\mathrm{tr}(W)}{m}}\log\left(\frac{t}{\delta}\right)\log\left(\frac{r}{\delta}\right)\right) \tag{237}$$

Finally taking union bound over the above results and using Lemma B.2, we can bound each of the terms constituting $F$. Using the definitions of $\lambda_1 = (r/t)\lambda_1(W)$ and $\lambda_r = (r/t)\lambda_r(W)$, (26), and

(27) (recall that we set $t \leftarrow t/K$) in (234) we get

$$\|\mathcal{H}(U^*Q)\|_F \leq \sqrt{\lambda_1(W)}\|H\|_F \tag{238}$$

$$\leq \lambda_r(W)\sqrt{\frac{\lambda_1}{\lambda_r}}\sqrt{\frac{r}{t}}\frac{\|H\|_F}{\sqrt{\lambda_r}} \tag{239}$$

$$\leq \lambda_r(W)O\Big(\frac{\lambda_1^*}{\lambda_r^*}\sqrt{\frac{\log(\frac{t}{\delta})}{\log(\frac{1}{\delta})}}\|(\mathbf{I} - U^*(U^*)^\top)U\|_F + \frac{\sigma}{\sqrt{\lambda_r^*}}\sqrt{\frac{\lambda_1^*}{\lambda_r^*}\frac{r^2\log(\frac{t}{\delta})}{m}}\Big) \tag{240}$$

Using the definitions of $\lambda_1 = (r/t)\lambda_1(W)$ and $\lambda_r = (r/t)\lambda_r(W)$, (27) and (28) in (236) we get

$$\|\mathcal{E}_2(U^*Q)\|_F$$

$$\leq \sqrt{\lambda_r(W)}O(\min(\|H\|_F\sqrt{\frac{\mu r}{t}}, \|H\|_{\infty,2}\sqrt{\frac{\lambda_1^*}{\lambda_r^*}})\sqrt{\frac{dr\,\log(15/\delta)}{m}} + \|H\|_{\infty,2}\sqrt{\frac{\mu r}{t}}\cdot\frac{dr\,\log(15/\delta)}{m})$$

$$\leq \lambda_r(W)O(\sqrt{\frac{r}{t}}\min(\frac{\|H\|_F}{\sqrt{\lambda_r^*}}\sqrt{\frac{\mu r}{t}}, \frac{\|H\|_{\infty,2}}{\sqrt{\lambda_r^*}}\sqrt{\frac{\lambda_1^*}{\lambda_r^*}})\sqrt{\frac{dr\,\log(15/\delta)}{m}} +$$

$$\sqrt{\frac{r}{t}}\frac{\|H\|_{\infty,2}}{\sqrt{\lambda_r^*}}\sqrt{\frac{\mu r}{t}}\cdot\frac{dr\,\log(15/\delta)}{m})$$

$$\leq \lambda_r(W)O\Big(\min\Big(\sqrt{\frac{\lambda_1^*}{\lambda_r^*}\frac{\mu dr^2\log(\frac{t}{\delta})}{mt}}\|(\mathbf{I} - U^*(U^*)^\top)U\|_F + \sqrt{\frac{\mu dr^2\log(\frac{1}{\delta})}{mt}}\frac{\sigma}{\sqrt{\lambda_r^*}}\sqrt{\frac{r^2\log(\frac{t}{\delta})}{m}},$$

$$\sqrt{\frac{\lambda_1^*}{\lambda_r^*}\frac{\mu dr^2\log(\frac{t}{\delta})}{mt}}\|(\mathbf{I} - U^*(U^*)^\top)U\| + \sqrt{\frac{\lambda_1^*}{\lambda_r^*}\frac{dr\log(\frac{1}{\delta})}{m}}\frac{\sigma}{\sqrt{\lambda_r^*}}\sqrt{\frac{r^2\log(\frac{t}{\delta})}{mt}}\Big) +$$

$$\frac{\mu dr^2\log(\frac{t}{\delta})}{mt}\|(\mathbf{I} - U^*(U^*)^\top)U\| + \frac{\sqrt{\mu}dr\sqrt{r}\log(\frac{1}{\delta})}{m\sqrt{t}}\frac{\sigma}{\sqrt{\lambda_r^*}}\sqrt{\frac{r^2\log(\frac{t}{\delta})}{mt}}\Big)$$

$$\leq \lambda_r(W)O\Big(\sqrt{\frac{\lambda_1^*}{\lambda_r^*}\frac{\mu dr^2\log(\frac{t}{\delta})}{mt}}\|(\mathbf{I} - U^*(U^*)^\top)U\| + \sqrt{\frac{\lambda_1^*}{\lambda_r^*}\frac{dr\log(\frac{1}{\delta})}{m}}\frac{\sigma}{\sqrt{\lambda_r^*}}\sqrt{\frac{r^2\log(\frac{t}{\delta})}{mt}} +$$

$$\frac{\mu dr^2\log(\frac{t}{\delta})}{mt}\|(\mathbf{I} - U^*(U^*)^\top)U\| + \frac{\sqrt{\mu}dr\sqrt{r}\log(\frac{1}{\delta})}{m\sqrt{t}}\frac{\sigma}{\sqrt{\lambda_r^*}}\sqrt{\frac{r^2\log(\frac{t}{\delta})}{mt}}\Big)$$

$$\leq \lambda_r(W)O\Big(\sqrt{\frac{\lambda_1^*}{\lambda_r^*}\frac{\mu dr^2\log(\frac{t}{\delta})}{mt}}\|(\mathbf{I} - U^*(U^*)^\top)U\| + \sqrt{\frac{\mu dr^2\log(\frac{t}{\delta})}{mt}}\frac{\sigma}{\sqrt{\lambda_r^*}}\sqrt{\frac{r^2\log(\frac{1}{\delta})}{m}}\Big) \tag{241}$$

where the second-last inequality used the fact that $mt \geq \Omega(\mu dr^2\log(\frac{t}{\delta}))$ and last inequality uses $\lambda_1^*/\lambda_r^* \leq \mu r$ (which follows from Assumption 2). Using the definitions of $\lambda_1 = (r/t)\lambda_1(W)$ and $\lambda_r = (r/t)\lambda_r(W)$, and (26) in (237) we get

$$\|\sum_{i\in[t]} z^{(i)}(v^{(i)})^\top)\|_F \leq \lambda_r(W)O\Big(\frac{\sigma}{\sqrt{\lambda_r^*}}\sqrt{\frac{\text{tr}(W)}{\lambda_r(W)}\frac{dr\log(\frac{t}{\delta})\log(\frac{r}{\delta})}{mt}}\Big) \tag{242}$$

$$\leq \lambda_r(W)O\Big(\frac{\sigma}{\sqrt{\lambda_r^*}}\sqrt{\frac{\mu dr^2\log(\frac{t}{\delta})\log(\frac{r}{\delta})}{mt}}\Big) \tag{243}$$

where the last inequality uses $\text{tr}(W)/\lambda_r(W) \le \mu r$ (which follows from Assumption 2 and (26)) Substituting (233), (241), (243), and (26) in (232) and using $m \ge \Omega(r^2 \log(1/\delta))$ we get

$$\|F\|_F \le (1 - \eta\lambda_r(W) + \eta\|\mathcal{E}_1\|_F)\|U - U^*Q\|_F +$$
$$\eta(\|\mathcal{E}_2(U^*Q)\|_F + \|\mathcal{W}^{\frac{1}{2}}\|\|\mathcal{W}^{-\frac{1}{2}}(\sum_{i\in[t]} z^{(i)}(v^{(i)})^\top)\|_F) \tag{244}$$

$$\le \left(1 - \eta\lambda_r(W)\left(1 - O\left(\sqrt{\frac{\lambda_1^*}{\lambda_r^*}\frac{\mu d r^2 \log(t/\delta)}{mt}}\right)\right)\right)\|(\mathbf{I} - U^*(U^*)^\top)U\|_F +$$
$$\eta\lambda_r(W)O\left(\sqrt{\frac{\mu d r^2 \log(\frac{t}{\delta})}{mt}}\frac{\sigma}{\sqrt{\lambda_r^*}}\sqrt{\frac{r^2 \log(\frac{1}{\delta})}{m}} + \frac{\sigma}{\sqrt{\lambda_r^*}}\sqrt{\frac{\mu d r^2 \log(\frac{t}{\delta})\log(\frac{r}{\delta})}{mt}}\right) \tag{245}$$

$$\le (1 - \frac{\eta}{2}\lambda_r(W))\|(\mathbf{I} - U^*(U^*)^\top)U\|_F + \eta\lambda_r(W)O\left(\frac{\sigma}{\sqrt{\lambda_r^*}}\sqrt{\frac{\mu d r^2 \log(\frac{t}{\delta})\log(\frac{r}{\delta})}{mt}}\right) \tag{246}$$

Finally, by resetting $\mathcal{T}_k \leftarrow \mathcal{T}_k$, $|\mathcal{T}_k| = t/K \leftarrow t/K$, $S_2^{(i)} \leftarrow S_2^{(i)} = \frac{2}{m}\sum_{j\in[m/2+1,m]} x_j^{(i)}(x_j^{(i)})^\top$, we obtain the desired result. $\qquad\square$

### C.2.2 Supporting lemmas for the analysis of update on $U$

**Lemma C.3.** *If $\Omega(\mu d r^2 \log(27/\delta)) \le mt$, then with a probability of at least $1 - \delta/3$,*

$$\|\mathcal{E}_1\|_F \le \lambda_r(W)O\left(\sqrt{\frac{\lambda_1^*}{\lambda_r^*}\frac{\mu d r^2 \log(27/\delta)}{m\,t}}\right) \tag{247}$$

*Proof of Lemma C.3.* Let $\mathcal{S}_F = \{U \in \mathbb{R}^{d\times r} \mid \|U\|_F = 1\}$ be the set of all real matrices of dimensions $d \times r$ with unit Frobenius norm. For $\epsilon \le 1$, there exists an $\epsilon$-net, $N_\epsilon \subset \mathcal{S}_F$, of size $(1 + 2/\epsilon)^{dr}$ with respect to the Frobenius norm [49, Lemma 5.2]. That is for any $U' \in \mathcal{S}_F$, there exists some $U \in N_\epsilon$ such that $\|U' - U\|_F \le \epsilon$.

Consider a $U \in N_\epsilon$, such that $\|U\|_F = 1$. Now we will prove with high-probability that $\langle(\mathcal{A} - \mathcal{W})(U), U\rangle$ is small. Consider the the following quadratic form

$$\langle(\mathcal{A})(U), U\rangle = \left\langle \sum_{i\in[t]} S^{(i)}Uv^{(i)}(v^{(i)})^\top, U \right\rangle \tag{248}$$

$$= \sum_{i\in[t]}\frac{1}{m}\sum_{j\in[m]}(x_j^{(i)})^\top(Uv^{(i)}(v^{(i)})^\top U^\top)x_j^{(i)} \tag{249}$$

where $S^{(i)} = \frac{1}{m}\sum_{j\in[m]} x_j^{(i)}(x_j^{(i)})^\top$ and $x_j^{(i)} \sim \mathcal{N}(0, \mathbf{I}_{d\times d})$ are i.i.d. standard Gaussian random vectors and $W = \sum_{i\in[t]} v^{(i)}(v^{(i)})^\top$ is rank-$r$ matrix. We will use Hanson-Wright inequality (Lemma F.5) to prove that the above quadratic form concentrates around its mean. Notice that the the expectation of $\langle\mathcal{A}(U), U\rangle$ is $\langle\mathcal{W}(U), U\rangle$.

$$\sum_{i\in[t]}\mathbb{E}\left[\left\langle S^{(i)}Uv^{(i)}(v^{(i)})^\top, U\right\rangle\right] = \left\langle U\sum_{i\in[t]} v^{(i)}(v^{(i)})^\top, U\right\rangle = \langle UW, U\rangle = \langle\mathcal{W}(U), U\rangle . \tag{250}$$

We will also need the following bounds to apply the Hanson-Wright inequality. Recall that $\|V\|_{\infty,2} = \max_{i\in[t]}\|v^{(i)}\|$. Then,

$$\max_{i\in[t]}\|Uv^{(i)}(v^{(i)})^\top U^\top\| = \max_{i\in[t]}\|Uv^{(i)}\|^2 \le \max_{i\in[t]}\|U\|^2\|v^{(i)}\|^2 \le \|V\|_{\infty,2}^2 \tag{251}$$

Also note that,

$$\sum_{i\in[t]}\|Uv^{(i)}(v^{(i)})^\top U^\top\|_F^2 = \sum_{i\in[t]}\|Uv^{(i)}\|^4 = \max_{i\in[t]}\|Uv^{(i)}\|^2 \sum_{i\in[t]}\left\langle Uv^{(i)}, Uv^{(i)}\right\rangle \tag{252}$$

$$= \max_{i\in[t]}\|U\|^2\|v^{(i)}\|^2 \sum_{i\in[t]}\left\langle UU^\top, \sum_{i\in[t]} v^{(i)}(v^{(i)})^\top\right\rangle \tag{253}$$

$$\leq \|V\|_{\infty,2}^2 \lambda_1(W) \tag{254}$$

where the last inequality used (250) and (251). Then by Hanson-Wright inequality (Lemma F.5), with probability at least $1 - \delta/|N_\epsilon|$

$$\left|\left\langle (\mathcal{A}-\mathcal{W})(U), U\right\rangle\right| = \left|\left\langle \sum_{i\in[t]}\frac{1}{m}\sum_{j\in[m]} x_j^{(i)}(x_j^{(i)})^\top Uv^{(i)}(v^{(i)})^\top, U\right\rangle - \left\langle \mathcal{W}(U), U\right\rangle\right| \leq \Delta_\epsilon \tag{255}$$

where $\Delta_\epsilon = c\max(\sqrt{\frac{\|V\|_{\infty,2}^2 \lambda_1(W)\log(|N_\epsilon|/\delta)}{m}}, \frac{\|V\|_{\infty,2}^2 \log(|N_\epsilon|/\delta)}{m})$. Taking union bound over all $U \in N_\epsilon$ implies that with probability at least $1 - \delta$

$$\left|\left\langle (\mathcal{A}-\mathcal{W})(U), U\right\rangle\right| \leq \Delta_\epsilon \ , \ \text{for all } U \in N_\epsilon. \tag{256}$$

For brevity, let $\mathcal{E}_1(U) = (\mathcal{A}-\mathcal{W})(U)$. Notice that $\mathcal{E}_1$ is self-adjoint, therefore it has an eigen decomposition with respect to the Frobenius norm. Then, let $U' \in \mathcal{S}_F \subset \mathbb{R}^{d\times r}$ be the largest "eigenmatrix" of $\mathcal{E}_1$, such that $\langle \mathcal{E}_1(U), U\rangle = \|\mathcal{E}_1\|_F = \max_{\|\widetilde{U}\|_F=1}\left\langle \mathcal{E}_1(\widetilde{U}), \widetilde{U}\right\rangle = \max_{\|\widetilde{U}\|_F=\|\widetilde{U}'\|_F=1}\left\langle \mathcal{E}_1(\widetilde{U}), \widetilde{U}'\right\rangle$. Then there exists some $U \in N_\epsilon$ such that $\|U'-U\|_F \leq \epsilon$.

$$\|\mathcal{E}_1\|_F = \langle \mathcal{E}_1(U'), U'\rangle = \langle \mathcal{E}_1(U), U\rangle + \langle \mathcal{E}_1(U'-U), U\rangle + \langle \mathcal{E}_1(U'), U'-U\rangle \tag{257}$$

$$\leq \langle \mathcal{E}_1(U), U\rangle + \|\mathcal{E}_1\|_F\|U'-U\|_F(\|U\|_F + \|U'\|_F) \tag{258}$$

$$\leq \langle \mathcal{E}_1(U), U\rangle + 2\epsilon\|\mathcal{E}_1\|_F \tag{259}$$

Re-arranging and setting $\epsilon = 1/4$, and $c \leftarrow 2c$, we get

$$\|\mathcal{A}-\mathcal{W}\|_F = \|\mathcal{E}_1'\|_F \leq \Delta_{\frac{1}{4}} \leq O\left(\sqrt{\frac{\lambda_1^*}{\lambda_r^*}\frac{\mu dr^2\log(9/\delta)}{m\,t}}\right). \tag{260}$$

where we use the approximate incoherence of intermediate variable $V$ Lemma B.2 and the fact that $\Omega(\mu dr^2\log(9/\delta)) \leq mt$, which implies that $\Delta_{\frac{1}{4}} = c\max\left(\sqrt{\frac{dr\,\|V\|_{\infty,2}^2\lambda_1(W)\log(9/\delta)}{m}}, \frac{dr\,\|V\|_{\infty,2}^2\log(9/\delta)}{m}\right) \leq \lambda_r(W)O\left(\sqrt{\frac{\lambda_1^*}{\lambda_r^*}}\max\left(\sqrt{\frac{\mu dr^2\log(9/\delta)}{mt}}, \frac{\mu dr^2\log(9/\delta)}{mt}\right)\right) \leq \lambda_r(W)O\left(\sqrt{\frac{\lambda_1^*}{\lambda_r^*}\frac{\mu dr^2\log(9/\delta)}{m\,t}}\right)$. Finally, setting $\delta \leftarrow \delta/3$ get us the desired result. $\square$

**Lemma C.4.** $\|(\mathcal{W}^{-\frac{1}{2}}\mathcal{H})(U^*Q)\|_F \leq \sqrt{\lambda_1(W)}\|H\|_F$ and with a probability of at least $1 - \delta/3$

$$\|\mathcal{E}_2(U^*Q)\|_F$$

$$\leq c(\min(\|H\|_F\|V\|_{\infty,2}, \|H\|_{\infty,2}\sqrt{\lambda_1(W)})\sqrt{\frac{dr\,\log(5/\delta)}{m}} + \|H\|_{\infty,2}\|V\|_{\infty,2}\frac{dr\,\log(5/\delta)}{m}) \tag{261}$$

*Proof of Lemma C.4.* First we prove that the expected value $\mathbb{E}[(\widehat{\mathcal{H}})(U^*Q)] = (\mathcal{H})(U^*Q)$ is bounded.

$$\|\mathcal{H}(U^*Q)\|_F = \max_{\|U\|_F=1} \langle \mathcal{H}(U^*Q), U \rangle \tag{262}$$

$$= \max_{\|U\|_F=1} \sum_{i \in [t]} \langle U^*Qh^{(i)}(v^{(i)})^\top, U \rangle \tag{263}$$

$$= \max_{\|U\|_F=1} \sum_{i \in [t]} \langle U^*Qh^{(i)}, Uv^{(i)} \rangle \tag{264}$$

$$\leq \max_{\|U\|_F=1} \sqrt{\sum_{i \in [t]} \|U^*Qh^{(i)}\|^2} \sqrt{\sum_{i \in [t]} \langle Uv^{(i)}, Uv^{(i)} \rangle} \tag{265}$$

$$\leq \max_{\|U\|_F=1} \|Q\| \sqrt{\sum_{i \in [t]} \|h^{(i)}\|^2} \sqrt{\langle U \sum_{i \in [t]} v^{(i)}(v^{(i)})^\top, U \rangle} \tag{266}$$

$$\leq \max_{\|U\|_F=1} \|H\|_F \|U\|_F \sqrt{\lambda_1(W)} = \sqrt{\lambda_1(W)} \|H\|_F \tag{267}$$

where used the fact that $\langle AB, C \rangle = \langle A, CB^\top \rangle$ and $(U^*)^\top U^* = \mathbf{I}$.

Let $\mathcal{S}_F = \{U \in \mathbb{R}^{d \times r} \mid \|U\|_F = 1\}$ be the set of all real matrices of dimensions $d \times r$ with unit Frobenius norm. For $\epsilon \leq 1$, there exists an $\epsilon$-net, $N_\epsilon \subset \mathcal{S}_F$, of size $(1 + 2/\epsilon)^{dr}$ with respect to the Frobenius norm [49, Lemma 5.2]. That is for any $U' \in \mathcal{S}_F$, there exists some $U \in N_\epsilon$ such that $\|U' - U\|_F \leq \epsilon$.

Consider a $U \in N_\epsilon$, such that $\|U\|_F = 1$. Now we will prove with high-probability that $\langle \mathcal{H}(U^*Q)(U) - \sum_{i \in [t]} S^{(i)}U^*Qh^{(i)}(v^{(i)})^\top, U \rangle$ is small. Consider the the following quadratic form

$$\langle \sum_{i \in [t]} S^{(i)}U^*Qh^{(i)}(v^{(i)})^\top, U \rangle = \langle \sum_{i \in [t]} S^{(i)}U^*Qh^{(i)}(v^{(i)})^\top, U \rangle \tag{268}$$

$$= \sum_{i \in [t]} \frac{1}{m} \sum_{j \in [m]} (x_j^{(i)})^\top (U^*Qh^{(i)}(v^{(i)})^\top U^\top) x_j^{(i)} \tag{269}$$

where $S^{(i)} = \frac{1}{m} \sum_{j \in [m]} x_j^{(i)}(x_j^{(i)})^\top$ and $x_j^{(i)} \sim \mathcal{N}(0, \mathbf{I}_{d \times d})$ are i.i.d. standard Gaussian random vectors. We will use Hanson-Wright inequality (Lemma F.5) to prove that the above quadratic form concentrates around its mean. Notice that the the expectation of $\langle \sum_{i \in [t]} S^{(i)}U^*Qh^{(i)}(v^{(i)})^\top, U \rangle$ is $\langle \mathcal{H}(U), U \rangle$.

$$\mathbb{E}[\sum_{i \in [t]} S^{(i)}U^*Qh^{(i)}(v^{(i)})^\top] = \sum_{i \in [t]} U^*Qh^{(i)}(v^{(i)})^\top = \mathcal{H}(U^*Q) . \tag{270}$$

We will also need the following bounds to apply the Hanson-Wright inequality. Recall that $\|H\|_{\infty,2} = \max_{i \in [t]} \|h^{(i)}\|$ and $\|V\|_{\infty,2} = \max_{i \in [t]} \|v^{(i)}\|$. Then,

$$\max_{i \in [t]} \|U^*Qh^{(i)}(v^{(i)})^\top U^\top\| \leq \max_{i \in [t]} \|U^*\|\|Q\|\|h^{(i)}\| \max_{i \in [t]} \|v^{(i)}\|\|U\| \leq \|H\|_{\infty,2}\|V\|_{\infty,2} \tag{271}$$

Also note that

$$\sum_{i \in [t]} \|U^*Qh^{(i)}(v^{(i)})^\top U^\top\|_F^2 = \sum_{i \in [t]} \|U^*Qh^{(i)}\|^2 \|Uv^{(i)}\|^2 \tag{272}$$

$$\leq (\sum_{i \in [t]} \|U^*Qh^{(i)}\|^2)(\max_{i \in [t]} \|Uv^{(i)}\|^2) \tag{273}$$

$$\leq (\|Q\|^2 \sum_{i \in [t]} \|h^{(i)}\|^2)(\max_{i \in [t]} \|U\|^2\|v^{(i)}\|^2) \tag{274}$$

$$\leq \|H\|_F^2 \|V\|_{\infty,2}^2 \tag{275}$$

and

$$\sum_{i \in [t]} \|U^* Q h^{(i)} (v^{(i)})^\top U^\top\|_F^2 = \sum_{i \in [t]} \|U^* Q h^{(i)}\|^2 \|U v^{(i)}\|^2 \tag{276}$$

$$\leq (\max_{i \in [t]} \|U^* Q h^{(i)}\|^2) \mathrm{tr}(U \sum_{i \in [t]} v^{(i)} (v^{(i)})^\top U^\top) \tag{277}$$

$$\leq (\max_{i \in [t]} \|U^* Q h^{(i)}\|^2) \langle U U^\top, W \rangle \tag{278}$$

$$\leq \|Q\| \max_{i \in [t]} \|h^{(i)}\|^2 \|U\|_F^2 \lambda_1(W) \tag{279}$$

$$= \|H\|_{\infty,2}^2 \lambda_1(W) . \tag{280}$$

Therefore, $\sum_{i \in [t]} \|U^* Q h^{(i)} (v^{(i)})^\top U^\top\|_F^2 \leq \min\{\|H\|_F^2 \|V\|_{\infty,2}^2, \|H\|_{\infty,2}^2 \lambda_1(W)\}$. For brevity, let $\mathcal{E}_2(U) = \sum_{i \in [t]} S^{(i)} U h^{(i)} (v^{(i)})^\top - \mathcal{H}(U)$. Then by Hanson-Wright inequality (Lemma F.5), with probability at least $1 - \delta/|N_\epsilon|$

$$\left|\langle \mathcal{E}_2(U^* Q), U \rangle\right| = \left|\left\langle \sum_{i \in [t]} \frac{1}{m} \sum_{j \in [m]} x_j^{(i)} (x_j^{(i)})^\top U^* Q h^{(i)} (v^{(i)})^\top, U \right\rangle - \langle \mathcal{H}(U^* Q), U \rangle\right| \leq \Delta_\epsilon \tag{281}$$

where $\Delta_\epsilon = c(\min(\|H\|_F \|V\|_{\infty,2}, \|H\|_{\infty,2} \sqrt{\lambda_1(W)}) \sqrt{\frac{\log(|N_\epsilon|/\delta)}{m}} + \|H\|_{\infty,2} \|V\|_{\infty,2} \frac{\log(|N_\epsilon|/\delta)}{m})$. Taking union bound over all $U \in N_\epsilon$ implies that with probability at least $1 - \delta$

$$\left|\langle \mathcal{E}_2(U), U \rangle\right| \leq \Delta_\epsilon , \quad \text{for all } U \in N_\epsilon . \tag{282}$$

Let $U' \in \mathcal{S}_F \subset \mathbb{R}^{d \times r}$ be the matrix "parallel" to $\mathcal{E}_2(U^* Q)$, that is $\|\mathcal{E}_2(U^* Q)\|_F = \max_{\|\widetilde{U}\|_F = 1} \left\langle \mathcal{E}_1(U^* Q), \widetilde{U} \right\rangle = \langle \mathcal{E}_2(U^* Q), U' \rangle$. Then there exists some $U \in N_\epsilon$ such that $\|U' - U\|_F \leq \epsilon$.

$$\|\mathcal{E}_2(U^* Q)\|_F = \langle \mathcal{E}_2(U^* Q), U' \rangle = \langle \mathcal{E}_2(U^* Q), U \rangle + \langle \mathcal{E}_2(U^* Q), U' - U \rangle \tag{283}$$

$$\leq \langle \mathcal{E}_1(U), U \rangle + \|\mathcal{E}_2(U^* Q)\|_F \|U' - U\|_F \tag{284}$$

$$\leq \langle \mathcal{E}_1(U), U \rangle + \epsilon \|\mathcal{E}_2(U^* Q)\|_F \tag{285}$$

Re-arranging and setting $\epsilon = 1/2$, and $c \leftarrow 2c$, we get

$$\|\mathcal{E}_2(U^* Q)\|_F \leq \Delta_{\frac{1}{2}}$$

$$= c(\min(\|H\|_F \|V\|_{\infty,2}, \|H\|_{\infty,2} \sqrt{\lambda_1(W)}) \sqrt{\frac{dr \log(5/\delta)}{m}} + \|H\|_{\infty,2} \|V\|_{\infty,2} \frac{dr \log(5/\delta)}{m}) \tag{286}$$

Finally setting $\delta \leftarrow \delta/3$ get us the desired result.

$\square$

**Lemma C.5.** *With a probability of at least $1 - \delta/3$*

$$\|\sum_{i \in [t]} z^{(i)} (v^{(i)})^\top)\|_F \leq O\Big(\sigma \sqrt{\frac{d \mathrm{tr}(W)}{m} \log\Big(\frac{t}{\delta}\Big) \log\Big(\frac{r}{\delta}\Big)}\Big) \tag{287}$$

*Proof of Lemma C.5.* Notice that $z^{(i)}$ (defined in Appendix C) is a Gaussian random vector of the following form

$$z^{(i)} = \frac{1}{m} \sum_{j \in [m]} \varepsilon_j^{(i)} x_j^{(i)} = \frac{1}{m} \|\varepsilon^{(i)}\| g^{(i)}, g^{(i)} \sim \mathcal{N}(0, \mathbf{I}_{d \times d}) \tag{288}$$

Using Hanson-Wright inequality (Lemma F.5, by setting $m \leftarrow 1$, $x_1 \leftarrow \varepsilon^{(i)}$, and $A_1 \leftarrow \mathbf{I}_{m \times m}$) and taking union bound over all tasks, we get that, with probability of at least $1 - \frac{\delta}{2}$

$$\|\varepsilon^{(i)}\|^2 \leq \sigma^2 m(1 + c\sqrt{\frac{\log(\frac{2t}{\delta})}{m}} + c\frac{\log(\frac{2t}{\delta})}{m}) \leq 2c\,\sigma^2 m \log\left(\frac{2t}{\delta}\right), \quad \text{for all } i \in [t] \tag{289}$$

where used the fact that $m \geq 1$ and $\log\left(\frac{2t}{\delta}\right) \geq 1$.

Now it is easy check that

$$\sum_{i \in [t]} \|v^{(i)}\|^2 = \sum_{i \in [t]} \operatorname{tr}((v^{(i)})^\top v^{(i)}) = \sum_{i \in [t]} \operatorname{tr}(v^{(i)}(v^{(i)})^\top) = \operatorname{tr}(W) \leq \mu r \lambda_r(W) \tag{290}$$

Notice that $\sum_{i \in [t]} \frac{1}{m}\|\varepsilon^{(i)}\| g^{(i)} v_j^{(i)}$ is a Gaussian random vector of the following form

$$\sum_{i \in [t]} \frac{1}{m}\|\varepsilon^{(i)}\| g^{(i)} v_j^{(i)} = \frac{1}{m}\sqrt{\sum_{i \in [t]} \|\varepsilon^{(i)}\|^2 (v_j^{(i)})^2}\ \widehat{g}_j, \quad \widehat{g}_j \sim \mathcal{N}(0, \mathbf{I}_{d \times d}) \tag{291}$$

Using Hanson-Wright inequality (Lemma F.5, by setting $m \leftarrow 1$, $x_1 \leftarrow \widehat{g}_j$, and $A_1 \leftarrow \mathbf{I}_{d \times d}$) and taking union bound over all $j \in [r]$, we get that, with probability of at least $1 - \frac{\delta}{2}$

$$\|\widehat{g}_j\|^2 \leq d(1 + c\sqrt{\frac{\log(\frac{2r}{\delta})}{d}} + c\frac{\log(\frac{2r}{\delta})}{d}) \leq 2cd \log\left(\frac{2r}{\delta}\right), \quad \text{for all } j \in [r] \tag{292}$$

where used the fact that $d \geq 1$ and $\log\left(\frac{2r}{\delta}\right) \geq 1$.

Combining the above results and using union bound, we get that, with a probability of at least $1 - \delta$,

$$\left\| \sum_{i \in [t]} z^{(i)}(v^{(i)})^\top) \right\|_F^2 = \left\| \sum_{i \in [t]} z^{(i)}(v^{(i)})^\top \right\|_F^2 \tag{293}$$

$$= \left\| \sum_{i \in [t]} \frac{1}{m}\|\varepsilon^{(i)}\| g^{(i)}(v^{(i)})^\top \right\|_F^2 \tag{294}$$

$$= \sum_{j \in [r]} \left\| \sum_{i \in [t]} \frac{1}{m}\|\varepsilon^{(i)}\| g^{(i)} v_j^{(i)} \right\|^2 \tag{295}$$

$$\leq \sum_{j \in [r]} \sum_{i \in [t]} \frac{\|\varepsilon^{(i)}\|^2}{m^2}(v_j^{(i)})^2 \|\widehat{g}_j\|^2 \tag{296}$$

$$\leq \sum_{j \in [r]} \sum_{i \in [t]} O\left(\frac{m\sigma^2}{m^2} \log\left(\frac{t}{\delta}\right)\right)(v_j^{(i)})^2 O\left(d \log\left(\frac{r}{\delta}\right)\right) \tag{297}$$

$$\leq O\left(\frac{d\sigma^2}{m} \log\left(\frac{t}{\delta}\right) \log\left(\frac{r}{\delta}\right)\right) \sum_{i \in [t]} \|v^{(i)}\|^2 \tag{298}$$

$$\leq O\left(\frac{\sigma^2 d\operatorname{tr}(W)}{m} \log\left(\frac{t}{\delta}\right) \log\left(\frac{r}{\delta}\right)\right). \tag{299}$$

Finally, we get the desired result by setting $\delta \leftarrow \delta/3$. $\qquad\square$

## C.3 Analysis of QR decomposition

*Proof of Lemma C.2.*

$$\sigma_{\min}(R) \geq \min_{\|z\|=1} \|Rz\| = \min_{\|z\|=1} \|U^+Rz\| = \min_{\|z\|=1} \|\widehat{U}z\| \tag{300}$$

$$\geq \min_{\|z\|=1} \|(U^*Q - \eta\mathcal{H}(U^*Q) + F)z\| \tag{301}$$

$$\geq \min_{\|z\|=1} \sqrt{z^\top Q^\top Qz} - \eta\|\mathcal{H}(U^*Q)\| - \|F\| \tag{302}$$

$$\geq \min_{\|z\|=1} \sigma_{\min}(Q) - \eta\|\mathcal{H}(U^*Q)\| - \|F\| \tag{303}$$

$$\geq 1 - \frac{1}{21}\frac{\lambda_r^*}{\lambda_1^*} - \frac{1}{21}\frac{\lambda_r^*}{\lambda_1^*} - \frac{1}{21}\frac{\lambda_r^*}{\lambda_1^*} \geq 1 - \frac{1}{7}\frac{\lambda_r^*}{\lambda_1^*} \tag{304}$$

There fore $R$ is invertible and $\|R^{-1}\| = (\sigma_{\min}(R))^{-1} \leq \frac{1}{1-\frac{1}{7}\frac{\lambda_r^*}{\lambda_1^*}} \leq 1 + \frac{1}{6}\frac{\lambda_r^*}{\lambda_1^*}$  □

# D  Analysis of AltMinGD-S (Algorithm 2) and AltMin-S (Algorithm 4) with subset selection

In this section analyze the task subset selection-based algorithms: AltMinGD-S (Algorithm 2) and AltMin-S (Algorithm 4).

**AltMin-S**: Initialized at $U$, the $k$-the step of alternating minimization-based AltMin-S (Algorithm 4) is:

$$\mathcal{T}_k = \left\{ i \in [1 + \frac{(k-1)t}{K}, \frac{tk}{K}] \mid \sigma_{\min}(U^\top S^{(i)}U) \geq 1/2 \text{ and } \sigma_{\max}(U^\top S^{(i)}U) \leq 2 \right\} \tag{305}$$

$$v^{(i)} \leftarrow (U^\top S^{(i)}U)^\dagger((U^\top S^{(i)}U^*)v^{*(i)} + U^\top z^{(i)}), \qquad \text{for } i \in \mathcal{T}_k \tag{306}$$

$$\widehat{U} \leftarrow \mathcal{A}^\dagger\left( \sum_{i \in \mathcal{T}} S^{(i)}U^*v^{*(i)}(v^{(i)})^\top + z^{(i)}(v^{(i)})^\top \right), \tag{307}$$

$$U^+ \leftarrow \text{QR}(\widehat{U}), \tag{308}$$

where $U^+$ is the next iterate, $S_1^{(i)} = \frac{2}{m}\sum_{j \in [1,m/2]} x_j^{(i)}(x_j^{(i)})^\top$, $S_2^{(i)} = \frac{2}{m}\sum_{j \in [1+m/2,m]} x_j^{(i)}(x_j^{(i)})^\top$, $z^{(i)} \triangleq (1/m)\sum_{j \in [m]} \varepsilon_j^{(i)} x_j^{(i)}$ and $\mathcal{A} : \mathbb{R}^{d \times r} \to \mathbb{R}^{d \times r}$ is a self-adjoint linear operator such that $\mathcal{A}(U) = \sum_{i \in T} S^{(i)}Uv^{(i)}(v^{(i)})^\top$.

**Theorem 10** (Formal version of Theorem 6 in Appendix A)**.** *Let there be $t$ linear regression tasks, each with $m$ samples satisfying Assumptions 1 and 2, and $K = \lceil \log_2(\frac{(\lambda_r^*/\lambda_1^*)mt}{\mu dr^2}) \rceil$, $\|(\mathbf{I} - U^*(U^*)^\top)U_{\text{init}}\|_F \leq \min\left( \frac{21}{121}, O\left(\sqrt{\frac{\lambda_r^*}{\lambda_1^*}\frac{1}{\log(t/K)}}\right) \right)$, $m \geq \Omega\left( \left(\frac{\sigma}{\sqrt{\lambda_r^*}}\right)^2 r^2 \log(\frac{t}{\delta}) + r^2 \log(\frac{K}{\delta}) + \log(\mu r) \right)$, $t \geq \Omega(\mu^2 r^3 K \log(\frac{K}{\delta}))$ and $mt \geq \Omega\left( \mu dr^2 K \frac{\lambda_1^*}{\lambda_r^*}\left( \log(\frac{t}{\delta}) + \left(\frac{\sigma}{\sqrt{\lambda_r^*}}\right)^2 \log^2(\frac{t}{\delta})\log(\frac{rK}{\delta}) \right) \right)$. Then, for any $0 < \delta < 1$, after $K$ iterations, AltMin-S (Algorithm 4) returns an orthonormal matrix $U \in \mathbb{R}^{d \times r}$, such that with a probability of at least $1 - \delta$*

$$\frac{1}{\sqrt{r}}\|(\mathbf{I} - U^*(U^*)^\top)U\|_F \leq O\left( \frac{\sigma}{\sqrt{\lambda_r^*}}\sqrt{\frac{\mu dr K \log(\frac{t}{\delta})\log(\frac{rK}{\delta})}{mt}} \right) \tag{309}$$

*and the algorithm uses an additional memory of size $O(d^2r^2)$.*

A proof is in Section D.1.

**AltMin-S**: Initialized at $U$, the $k$-the step of alternating minimization-based AltMinGD-S (Algorithm 2) is:

$$\mathcal{T}_k = \Big\{ i \in [1 + \frac{(k-1)t}{K}, \frac{tk}{K}] \mid \sigma_{\min}(U^\top S^{(i)}U) \geq 1/2 \text{ and } \sigma_{\max}(U^\top S^{(i)}U) \leq 2 \Big\} \tag{310}$$

$$v^{(i)} \leftarrow (U^\top S^{(i)}U)^\dagger((U^\top S^{(i)}U^*)v^{*(i)} + U^\top z^{(i)}), \qquad \text{for } i \in \mathcal{T}_k \tag{311}$$

$$\widetilde{U} \leftarrow U - \eta\Big( \sum_{i \in [t]} S_2^{(i)}(Uv^{(i)} - U^*v^{*(i)})(v^{(i)})^\top + z^{(i)}(v^{(i)})^\top \Big), \tag{312}$$

$$U^+ \leftarrow \mathrm{QR}(\widetilde{U}), \tag{313}$$

where $U^+$ is the next iterate, $S_1^{(i)}$, $S_2^{(i)}$, and $\mathcal{A}$ are defined in the same way as above for AltMin-S.

**Theorem 11** (Formal version of Theorem 3 in Section 4)). *Let there be $t$ linear regression tasks, each with $m$ samples satisfying Assumptions 1 and 2, and $K = \Omega(\lceil \frac{\lambda_1^*}{\lambda_r^*} \log(\frac{mt}{(\lambda_1^*/\lambda_r^*)(\sigma/\sqrt{\lambda_r^*})\mu dr}) \rceil)$,*

$\|(\mathbf{I} - U^*(U^*)^\top)U_{\mathrm{init}}\|_F \leq \min\left( \frac{21}{121}, O\Big(\frac{\lambda_r^*}{\lambda_1^*}\sqrt{\frac{1}{\log(t/K)}}\Big)\right)$, $m \geq \Omega\Big(r^2\frac{\lambda_1^*}{\lambda_r^*}\Big(\frac{\sigma}{\sqrt{\lambda_r^*}}\Big)^2 \log(\frac{t}{\delta}) + r^2\log(\frac{K}{\delta}) + \log(\mu r)\Big)$, $t \geq \Omega(\mu^2 r^3 K \log(\frac{K}{\delta}))$ *and* $mt \geq \Omega\Big(\mu dr^2 K \log(\frac{t}{\delta})\Big(1 + \Big(\frac{\lambda_1^*}{\lambda_r^*}\Big)^2\Big(\frac{\sigma}{\sqrt{\lambda_r^*}}\Big)^2 \log(\frac{t}{\delta})\log(\frac{rK}{\delta})\Big)\Big)$. *Then, for any $0 < \delta < 1$, after $K$ iterations and using the stepsize $\eta = (r/t)/2\lambda_1^*$, AltMinGD-S (Algorithm 2) returns an orthonormal matrix $U \in \mathbb{R}^{d\times r}$, such that with a probability of at least $1 - \delta$*

$$\frac{1}{\sqrt{r}}\|(\mathbf{I} - U^*(U^*)^\top)U\|_F \leq O\Big(\frac{\sigma}{\sqrt{\lambda_r^*}}\sqrt{\frac{\mu dr^2 K \log(\frac{t}{\delta})\log(\frac{rK}{\delta})}{mt}}\Big) \tag{314}$$

A proof is in Section D.1.

### D.1 Proofs of Theorem 10 (formal version of Theorem 6 in Appendix A) and Theorem 11 (formal version of Theorem 3 in Section 4)

Here we provide only the proof of Theorem 10 as the proof of Theorem 11 is very similar and straightforward, given the former.

First, in the following lemma, we prove that the task subset $\mathcal{T}_k$ has similar properties as the full task partition $[1 + t(k-1)/K, tk/K]$.

**Lemma D.1** (Subset selection). *If $m \geq \Omega(r + \log(\mu r))$ and $t \geq \Omega(\mu^2 r^2 K \log(\frac{1}{\delta}))$, then with a probability of at least $1 - \delta/3$,*

$$|\mathcal{T}_k| = \Theta\Big(\frac{t}{K}\Big), \quad , \text{ and } \quad \|V^*\|_{\infty,2}^2 \leq O\Big(\frac{\mu r}{|\mathcal{T}_k|}\lambda_r(\sum_{i\in\mathcal{T}}v^{*(i)}(v^{*(i)})^\top)\Big) \tag{315}$$

$$\lambda_r(\sum_{i\in\mathcal{T}}v^{*(i)}(v^{*(i)})^\top) = \Theta(\lambda_r(\sum_{i\in\mathcal{P}_k}v^{*(i)}(v^{*(i)})^\top)), \text{ and} \tag{316}$$

$$\lambda_1(\sum_{i\in\mathcal{T}}v^{*(i)}(v^{*(i)})^\top) = \Theta(\lambda_1(\sum_{i\in\mathcal{P}_k}v^{*(i)}(v^{*(i)})^\top)), \tag{317}$$

*where $\mathcal{P}_k = [1 + t(k-1)/K, tk/K]$ is the $k$-th $K$-way partition of $[t]$ after shuffling.*

A proof is in Section D.2. Therefore, assuming that the above high-probability event holds, in the rest of the proof we can consider that $\mathcal{T}_k$ is equivalent to $\mathcal{P}_k$.

In the rest of the proof, when compared to the proof of Theorem 8, only the following Lemma (corresponding to Lemma B.2) analyzing the $V$ update changes in its necessary condition.

**Lemma D.2.** *If $\|(\mathbf{I} - U^*(U^*)^\top)U\|_F \leq \min\left(\frac{21}{121}, O\Big(\sqrt{\frac{\lambda_r^*}{\lambda_1^*}\frac{1}{\log(t/K)}}\Big)\right)$ and $m \geq \Omega\Big(\Big(\frac{\sigma}{\sqrt{\lambda_r^*}}\Big)^2 r^2 \log(\frac{t}{K\delta}) + r\log(\frac{1}{\delta})\Big)$, then with a probability of at least $1 - \delta/3$,*

$$\|v^{(i)}\| \leq O\Big(\mu\lambda_r\Big), \quad \lambda_1 \leq 2\lambda_1^*, \text{ and } \lambda_r^*/2 \leq \lambda_r \leq 2\lambda_r^* \tag{318}$$

*and*

$$\sqrt{\frac{rK}{t}} \frac{\|H\|_F}{\sqrt{\lambda_r}} \leq O\left(\sqrt{\frac{\log(\frac{t}{K\delta})}{\log(\frac{1}{\delta})}} \sqrt{\frac{\lambda_1^*}{\lambda_r^*}} \|(\mathbf{I} - U^*(U^*)^\top)U\|_F + \frac{\sigma}{\sqrt{\lambda_r^*}} \sqrt{\frac{r^2 \log(\frac{t}{K\delta})}{m}}\right) \quad (319)$$

$$\sqrt{\frac{rK}{t}} \frac{\|H\|_{\infty,2}}{\sqrt{\lambda_r}} \leq O\left(\sqrt{\frac{\log(\frac{t}{K\delta})}{\log(\frac{1}{\delta})}} \|(\mathbf{I} - U^*(U^*)^\top)U\| \sqrt{\frac{\mu rK}{t}} + \frac{\sigma}{\sqrt{\lambda_r^*}} \sqrt{\frac{r^2 K \log(\frac{t}{K\delta})}{mt}}\right) \quad (320)$$

A proof is in Section D.3.1. We omit the rest of the proof, as it is same as that of Theorem 8.

## D.2 Analysis of task subset selection

*Proof of Lemma D.1 (Subset selection).* Let $\mathcal{P}_k = [1 + (k-1)t/K, tk/K]$ and

$$\mathcal{T}_k = \left\{ i \in [1 + (k-1)t/K, tk/K] \mid \sigma_{\min}(U^\top S^{(i)}U) \geq 1/2 \text{ and } \sigma_{\max}(U^\top S^{(i)}U) \leq 2 \right\}. \quad (321)$$

For all $i \in \mathcal{P}_k$, $X_i = \mathbb{I}(\sigma_{\min}(U^\top S^{(i)}U) \geq 1/2 \text{ and } \sigma_{\max}(U^\top S^{(i)}U) \leq 2)$ be the indicator variable denoting whether index $i$ was select into the subset $\widehat{\mathcal{T}}$.

By Lemma F.7 (by setting $a_j \leftarrow 1$, $x_j \leftarrow U^\top x_j^{(i)}$ for all $j \in [m]$, and $\delta \leftarrow 1/4\mu r$) $X_i$ are i.i.d. Bernoulli random variables with mean $p \geq 1 - \frac{1}{4\mu r}$, if $c \max\left(\sqrt{\frac{r \log(9) + \log(4\mu r)}{m}}, \frac{r \log(9) + \log(4\mu r)}{m}\right) \leq 1/2$, which is satisfied by $m \geq \Omega(r + \log(\mu r))$, for all $i \in \mathcal{P}_k$.

By Hoeffding inequality for Bernoulli random variables, with a probability of at least $1 - \delta/3$

$$||\mathcal{T}_k| - pt/K| = \left|\sum_{i \in \mathcal{P}_k} X_i - (1 - \frac{1}{4\mu r})\frac{t}{K}\right| \leq \frac{t}{K}\sqrt{\frac{K \log(\frac{3}{\delta})}{2t}} \leq \frac{t}{K}O\left(\frac{1}{4\mu r}\right) \quad (322)$$

where we used the fact that $t \geq \Omega(8K\mu^2 r^2 \log(\frac{3}{\delta}))$. Therefore

$$\frac{t}{K} - |\mathcal{T}_k| \leq \frac{t}{K}O\left(\frac{1}{2\mu r}\right), \text{ and } |\mathcal{T}_k| \leq \Theta\left(\frac{t}{K}\right) \quad (323)$$

where we used the fact that $\mu \geq 1$ and $r \geq 1$.

$$\frac{r}{t}\left|z^\top\left(\sum_{i \in \mathcal{T}_k} v^{*(i)}(v^{*(i)})^\top\right)z - z^\top\left(\sum_{i \in \mathcal{P}_k} v^{*(i)}(v^{*(i)})^\top\right)z\right| \leq \frac{r}{t}(t - \widehat{t})\|V^*\|_{\infty,2}^2 \quad (324)$$

$$\leq \frac{r}{t}O\left(\frac{t}{2\mu r}\right) \cdot \|V^*\|_{\infty,2}^2 \leq \frac{\lambda_r}{2}, \quad (325)$$

for all $z \in \mathbb{R}^r$, where $\lambda_r = \lambda_r(\sum_{i \in \mathcal{P}_k} v^{*(i)}(v^{*(i)})^\top)$. Therefore

$$\lambda_r\left(\sum_{i \in \mathcal{T}} v^{*(i)}(v^{*(i)})^\top\right) = \Theta\left(\lambda_r\left(\sum_{i \in \mathcal{P}_k} v^{*(i)}(v^{*(i)})^\top\right)\right), \text{ and} \quad (326)$$

$$\lambda_1\left(\sum_{i \in \mathcal{T}} v^{*(i)}(v^{*(i)})^\top\right) = \Theta\left(\lambda_1\left(\sum_{i \in \mathcal{P}_k} v^{*(i)}(v^{*(i)})^\top\right)\right) \quad (327)$$

Using approximate incoherence of the partition $\mathcal{P}_k$ (Lemma B.1) we get

$$\|V^*\|^2_{\infty,2} \leq O\left(\frac{\mu r K}{t}\right) \lambda_r \left(\sum_{i \in \mathcal{P}_k} v^{*(i)} (v^{*(i)})^\top\right) \tag{328}$$

$$= O\left(\frac{\mu r K}{t}\right) \min_{\|z\|=1} z^\top \left(\sum_{i \in \mathcal{P}_k} v^{*(i)} (v^{*(i)})^\top\right) z \tag{329}$$

$$\leq O\left(\frac{\mu r K}{t}\right) \min_{\|z\|=1} z^\top \left(\sum_{i \in \mathcal{T}_k} v^{*(i)} (v^{*(i)})^\top\right) z + O\left(\frac{\mu r K}{t}\right)\left(\frac{t}{K} - |\mathcal{T}_k|\right)\|V^*\|^2_{\infty,2} \tag{330}$$

$$\leq O\left(\frac{\mu r K}{t}\right) \lambda_r \left(\sum_{i \in \mathcal{T}_k} v^{*(i)} (v^{*(i)})^\top\right) + \frac{1}{2}\|V^*\|^2_{\infty,2} \tag{331}$$

$$\tag{332}$$

This implies that approximate incoherence holds for $\mathcal{T}_k$, $\|V^*\|^2_{\infty,2} \leq O\left(\frac{\mu r K}{t}\right)\lambda_r(\sum_{i \in \mathcal{T}_k} v^{*(i)}(v^{*(i)})^\top) \leq O\left(\frac{\mu r}{|\mathcal{T}_k|}\lambda_r(\sum_{i \in \mathcal{T}_k} v^{(i)}(v^{(i)})^\top)\right).$

$\square$

## D.3 Analysis of update on $V$

### D.3.1 Proof of Lemma D.2

*Proof of Lemma D.2.* The proof is similar to that of Lemma B.2, but instead of using Lemma B.5 to bound some linear operators, we use the definition of selected task subset $\mathcal{T}_k$ and Lemma D.3 to get that $\|(U^\top S^{(i)} U)^\dagger\| \leq 2$ for all $i \in \mathcal{T}_k$ and with a probability of at least $1 - \delta$,

$$\left.\begin{array}{r} \|U^\top S^{(i)} U_\perp U_\perp^\top U^* v^{*(i)}\| \leq \alpha \|U_\perp^\top U^* v^{*(i)}\|, \text{ and} \\ \|U^\top z^{(i)}\| \leq \sigma\alpha, \end{array}\right\} \text{ for all } i \in \mathcal{T}_k \tag{333}$$

where $\alpha = c\sqrt{\frac{r\log(10t/\delta)}{m}}$. We omit the rest of the proof, as it is same as that of Lemma B.2. $\square$

Here we bound the linear operators in the $v^{(i)}$ update.

**Lemma D.3.** *With a probability of at least $1 - \delta$, the following are true for all $i \in [t]$*

$$\|U^\top S^{(i)} U_\perp (U_\perp)^\top U^* v^{*(i)}\| \leq \sqrt{\frac{2cr\log(10t/\delta)}{m}}\|U_\perp U^* v^{*(i)}\|, \text{ and} \tag{334}$$

$$\|U^\top z^{(i)}\| \leq \sigma\sqrt{\frac{2cr\log(10t/\delta)}{m}} \tag{335}$$

*Proof.* Let $i \in [t]$. Let $b = (U_\perp)^\top U^* v^{*(i)} \in \mathbb{R}^r$

Let $\mathcal{S} = \{v \in \mathbb{R}^r \mid \|v\| = 1\}$ be the set of all real vectors of dimension $r$ with unit Euclidean norm. For $\epsilon \leq 1$, there exists an $\epsilon$-net, $N_\epsilon \subset \mathcal{S}$, of size $(1 + 2/\epsilon)^r$ with respect to the Euclidean norm [49, Lemma 5.2]. That is for any $v' \in \mathcal{S}$, there exists some $v \in N_\epsilon$ such that $\|v' - v\|_F \leq \epsilon$.

Consider a $v \in N_\epsilon$, such that $\|v\|_F = 1$. Now we will prove with high-probability that $\langle (U^\top S^{(i)} U_\perp)v, b \rangle$ is small. Consider the the following quadratic form

$$v^\top (U^\top S^{(i)} U_\perp)b = \frac{1}{m}\sum_{j \in [m]} v^\top (U^\top x_j^{(i)}(x_j^{(i)})^\top U_\perp)b \stackrel{d}{=} \|b\|\frac{1}{m}\sum_{j \in [m]} \widetilde{x}_j g_j \tag{336}$$

where $g_j \sim \mathcal{N}(0,1)$) are i.i.d. standard Gaussian random variables and $\widetilde{x}_j = v^\top U^\top x_j^{(i)} \in \mathbb{R}^d$. This follows from the fact that sets of columns of $U$ and $U_\perp$ forms an orthonormal basis.

Note that $g_j$ and $\widetilde{x}_j$ are independent, as $U$ and $U_\perp$ are orthogonal and $U^\top S^{(i)} U$, does not depend on $U_\perp x_j^{(i)}$. We will use the properties of Gaussian random variables to prove that $\|\frac{1}{m} \sum_{j \in [m]} \widetilde{x}_j g_j\|$ concentrates around zero. Note that

$$\frac{1}{m} \sum_{j \in [m]} \widetilde{x}_j g_j \overset{d}{=} \frac{1}{m} \|\widetilde{x}\| g, \text{ where } g \sim \mathcal{N}(0, 1) \tag{337}$$

Then with probability at least $1 - \delta/2t/|N_\epsilon|$, $|g|^2 \le c \log(2t|N_\epsilon|/\delta)$. Additionally, by definition of $\mathcal{T}_k$ we have

$$\frac{1}{m} \|\widetilde{x}\|^2 = \frac{1}{m} \sum_{j \in [m]} \widetilde{x}_j^2 = v^\top U^\top \left(\frac{1}{m} \sum_{j \in [m]} x_j^{(i)} (x_j^{(i)})^\top\right) Uv \le \sigma_{\max}(U^\top S^{(i)} U) \le 2 \tag{338}$$

Therefore

$$v^\top (U^\top S^{(i)} U_\perp) b \le \frac{1}{\sqrt{m}} \|b\| \sqrt{2c} \sqrt{\log(2t|N_\epsilon|/\delta)} \tag{339}$$

For brevity, let $e = (U^\top S^{(i)} U_\perp) b$. Let $v' \in \mathcal{S} \subset \mathbb{R}^r$ be the unit vector parallel to $e$, such that $(v')^\top e = \|e\| = \max_{\|\widetilde{v}\|=1} \widetilde{v}^\top e$. Then there exists some $v \in N_\epsilon$ such that $\|v' - v\| \le \epsilon$.

$$\|e\| = (v')^\top e = v^\top e + (v' - v)^\top e \le v^\top e + \|v' - v\| \|e\| \le v^\top e + \epsilon \|e\| \tag{340}$$

Re-arranging and setting $\epsilon = 1/2$, and $c \leftarrow 2c$, we get

$$\|(U^\top S^{(i)} U_\perp) b\| \le \|b\| \sqrt{\frac{2cr \log(10t/\delta)}{m}}, \text{ with a probability of at least } 1 - \delta/2t \tag{341}$$

Using similar arguments we can also prove that with a probability of at least $1 - \delta$

$$\|U^\top z^{(i)}\| = \|\frac{1}{m} U^\top x_j^{(i)} \varepsilon_j^{(i)}\| \le \sigma \sqrt{\frac{2cr \log(10t/\delta)}{m}}, \text{ with a probability of at least } 1 - \delta/2t \tag{342}$$

Finally taking the union bound over the two bounds over all the tasks in $\mathcal{T}$ gets us the desired result. $\qquad\square$

# E    Corollaries of known results

**Theorem 12** (Theorem 3, Tripuraneni et al. 2020). *Let there be $t$ linear regression tasks, each with $m$ samples satisfying Assumptions 1 and 2, and*

$$mt \ge \widetilde{\Omega}\left(\frac{\lambda_1^*}{\lambda_r^*} \mu dr + \left(\frac{\sigma}{\sqrt{\lambda_r^*}}\right)^4 dr^2\right) \tag{343}$$

*then with a high probability of at least $1 - O((mt)^{-100})$, Method-of-Moments [48, Algorithm 1] outputs an orthonormal matrix $U \in \mathbb{R}^{d \times r}$ such that*

$$\|(\mathbf{I} - U^*(U^*)^\top) U\|_2 \le \tilde{O}\left(\sqrt{\frac{\lambda_1^*}{\lambda_r^*} \frac{\mu dr}{mt}} + \left(\frac{\sigma}{\sqrt{\lambda_r^*}}\right)^2 \sqrt{\frac{dr^2}{mt}}\right) \tag{344}$$

*and*

$$\|(\mathbf{I} - U^*(U^*)^\top) U\|_F \le \tilde{O}\left(\sqrt{\frac{\lambda_1^*}{\lambda_r^*} \frac{\mu dr^2}{mt}} + \left(\frac{\sigma}{\sqrt{\lambda_r^*}}\right)^2 \sqrt{\frac{dr^3}{mt}}\right). \tag{345}$$

*Proof.* From the details of the proof of Theorem 3 in [48] we can derive that, with a high probability of at least $1 - O((mt)^{-100})$,

$$\|(\mathbf{I} - U^*(U^*)^\top)U\|_2 \tag{346}$$

$$\leq \tilde{O}\Big(\sqrt{\frac{dr^2\mathrm{tr}(W^*)\|V^*\|_{\infty,2}^2}{\lambda_r^{*2}\, mt^2}} + \frac{dr\|V^*\|_{\infty,2}^2}{\lambda_r^*\, mt} + \sigma\Big(\sqrt{\frac{dr^2\mathrm{tr}(W^*)}{\lambda_r^{*2}\, mt^2}} + \frac{dr\|V^*\|_{\infty,2}}{\lambda_r^*\, mt}\Big) +$$

$$\sigma^2\Big(\sqrt{\frac{dr^2}{\lambda_r^{*2}\, mt}} + \frac{dr}{\lambda_r^*\, mt}\Big)\Big) \tag{347}$$

$$\leq \tilde{O}\Big(\sqrt{\frac{\lambda_1^*}{\lambda_r^*}\frac{\mu dr}{mt}} + \frac{\mu dr}{mt} + \frac{\sigma}{\sqrt{\lambda_r^*}}\Big(\sqrt{\frac{\lambda_1^*}{\lambda_r^*}\frac{dr}{mt}} + \frac{\sqrt{\mu}\, dr}{mt}\Big) + \Big(\frac{\sigma}{\sqrt{\lambda_r^*}}\Big)^2\Big(\sqrt{\frac{dr^2}{mt}} + \frac{dr}{mt}\Big)\Big) \tag{348}$$

$$\leq \tilde{O}\Big(\sqrt{\frac{\lambda_1^*}{\lambda_r^*}\frac{\mu dr}{mt}} + \frac{\sigma}{\sqrt{\lambda_r^*}}\sqrt{\frac{\lambda_1^*}{\lambda_r^*}\frac{dr}{mt}} + \Big(\frac{\sigma}{\sqrt{\lambda_r^*}}\Big)^2\sqrt{\frac{dr^2}{mt}}\Big) \tag{349}$$

$$\leq \tilde{O}\Big(\sqrt{\frac{\lambda_1^*}{\lambda_r^*}\frac{\mu dr}{mt}} + \Big(\frac{\sigma}{\sqrt{\lambda_r^*}}\Big)^2\sqrt{\frac{dr^2}{mt}}\Big) \tag{350}$$

where $\|V\|_{\infty,2} = \max_{i\in[t]}\|v^{(i)}\|$, and the second-last inequality uses the fact that $mt \geq \tilde{\Omega}(\mu dr)$ and last inequality uses the fact that $\frac{\lambda_1^*}{\lambda_r^*} \leq \mu r$. Additionally we require that

$$mt \geq \tilde{\Omega}\Big(\frac{\lambda_1^*}{\lambda_r^*}\mu dr + \Big(\frac{\sigma}{\sqrt{\lambda_r^*}}\Big)^4 dr^2\Big) \tag{351}$$

$\square$

**Theorem 13.** *[48, Theorem 5] Let $r \leq d/2$ and $mt \geq r(d-r)$, then for all $V^*$, w.p. $\geq 1/2$*

$$\inf_{\widehat{U}} \sup_{U \in \mathrm{Gr}_{r,d}} \frac{\|(\mathbf{I} - U^*(U^*)^\top)\widehat{U}\|_F}{\sqrt{r}} \geq \Omega\Big(\Big(\frac{\lambda_r^*}{\lambda_1^*}\frac{\sigma}{\sqrt{\lambda_r^*}}\Big)\sqrt{\frac{d\,r}{m\,t}}\Big),$$

*where $G_{r,d}$ is the Grassmannian manifold of $r$-dimensional subspaces in $\mathbb{R}^d$, the infimum for $\widehat{U}$ is taken over the set of all measurable functions that takes $mt$ samples in total from the model in Section 2 satisfying Assumption 1 and 2.*

*Proof.* The proof is very similar to that of Theorem 5 of [48]. The main difference is that instead of lower bounding error in spectral norm we have to bound the distance in the Frobenius norm. However, the rest of the details are almost the same, hence we omit a full proof. $\square$

# F Technical Lemmas

This section contains some technical lemmas used in this paper.

**Lemma F.1.** *For a real matrix $A \in \mathbb{R}^{m\times n}$ and a real symmetric positive semi-definite (PSD) matrix $B \in \mathbb{R}^{n\times n}$, the following holds true: $\sigma_{\min}^2(A)\lambda_{\min}(B) \leq \lambda_{\min}(ABA^\top)$, where $\sigma_{\min}(\cdot)$ and $\lambda_{\min}(\cdot)$ represents the minimum singular value and minimum eigenvalue operators respectively.*

*Proof.* The proof directly follows from the definitons of $\sigma_{\min}$ and $\lambda_{\min}$. Since $B$ is a PSD matrix, therefore $ABA^\top$ is also PSD, i.e. $\lambda_{\min}(ABA^\top) \geq 0$. This is because since $B$ is PSD, it has a PSD matrix square root $B^{1/2}$ such that $B = (B^{1/2})^\top B^{1/2}$ and $B^{1/2}$ is PSD. Then

$$z^\top ABA^\top z = z^\top A(B^{1/2})^\top B^{1/2}A^\top z = \|B^{1/2}A^\top z\|^2 \geq 0 \tag{352}$$

First assume that $\sigma_{\min}(A) > 0$, then

$$\lambda_{\min}(ABA^\top) = \min_{\|z\|=1} z^\top ABA^\top z \tag{353}$$

$$= \sigma_{\min}^2(A) \min_{\|z\|=1} \left(\frac{A^\top z}{\sigma_{\min}(A)}\right)^\top B \left(\frac{A^\top z}{\sigma_{\min}(A)}\right) \tag{354}$$

$$\geq \sigma_{\min}^2(A) \min_{1 \leq \|z\| \leq \frac{\sigma_{\max}(A)}{\sigma_{\min}(A)}} z^\top Bz \tag{355}$$

$$\geq \sigma_{\min}^2(A) \min_{\|z\|=1} z^\top Bz \tag{356}$$

$$= \sigma_{\min}^2(A)\lambda_{\min}(B) \tag{357}$$

The second last inequality above follows from the fact that $B$ is a PSD matrix,i.e. $\min_{\|z\|=1} z^\top Bz = \lambda_{\min}(B) \geq 0$. Secondly if $\sigma_{\min}(A) = 0$, then $A$ is rank deficient and hence $ABA^\top$ is also rank deficient, i.e. $\lambda_{\min}(ABA^\top) = 0$. Therefore $\lambda_{\min}(ABA^\top) = 0 = \sigma_{\min}^2(A)\lambda_{\min}(B)$. $\qquad\square$

**Lemma F.2** (Weyl's inequality [1]). *For three real $r$-rank matrices, satisfying $A - B = C$, Weyl's inequality [1, Theorem 3.6], tells that*

$$\sigma_k(A) - \sigma_k(B) \leq \|C\|, \text{ for all } k \in [r] \tag{358}$$

*where $\sigma_k(\cdot)$ is the $k$-th largest singular value operator.*

**Lemma F.3** (a variant of Woodburry matrix identity [26]). *For linear operators $A$ and $B$ such that $A$ and $A + B$ are invertible, then*

$$(A + B)^{-1} - A^{-1} = -A^{-1}B(A + B)^{-1} \tag{359}$$

**Lemma F.4.** *Let $U \in \mathbb{R}^{d \times r}$ and $U^* \in \mathbb{R}^{d \times r}$ be two orthonormal matrices. Let $\{\sin\theta_j(U, U^*)\}_{j=1}^r$ be the singular values of $(U^*)^\top U$. Then following are true.*

$$\|U - U^*(U^*)^\top U\|_F \geq \|\mathbf{I} - (U^*)^\top U\|_F, \tag{360}$$

$$\|U - U^*(U^*)^\top U\|_F \geq r - \|(U^*)^\top U\|_F^2 \geq \sum_{k \in [r]} \sin^2\theta_k(U, U^*), \tag{361}$$

$$\|(\mathbf{I} - U^*(U^*)^\top)U\| = \|(U^*_\perp)^\top U\| = \|U_\perp^\top U^*\| = \|(\mathbf{I} - U(U)^\top)U^*\|, \tag{362}$$

$$\|(\mathbf{I} - U^*(U^*)^\top)U\|_F = \|(U^*_\perp)^\top U\|_F = \|U_\perp^\top U^*\|_F = \|(\mathbf{I} - U(U)^\top)U^*\|_F, \text{ and} \tag{363}$$

$$\sigma_r((U^*)^\top U) \geq \sqrt{1 - \|(\mathbf{I} - U^*(U^*)^\top)U\|} \tag{364}$$

*Proof.*

$$\|U - U^*(U^*)^\top U\|_F^2 = \langle U - U^*(U^*)^\top U, U - U^*(U^*)^\top U \rangle \tag{365}$$

$$= \langle U, U \rangle - 2\langle U^*(U^*)^\top U, U \rangle + \langle U^*(U^*)^\top U, U^*(U^*)^\top U \rangle \tag{366}$$

$$= r - 2\mathrm{tr}(((U^*)^\top U)^\top((U^*)^\top U)) + \mathrm{tr}(((U^*)^\top U)^\top((U^*)^\top U)) \tag{367}$$

$$= r - \mathrm{tr}(((U^*)^\top U)^\top((U^*)^\top U)) \tag{368}$$

$$= r - \sum_{k \in [r]} \cos^2\theta_k(U, U^*) = \sum_{k \in [r]} \sin^2\theta_k(U, U^*) \geq \sin^2\theta_1(U, U^*) \tag{369}$$

$$\geq \sum_{k \in [r]} (1 - \cos^2\theta_k(U, U^*)) \tag{370}$$

$$\geq \sum_{k \in [r]} (1 - \cos\theta_k(U, U^*))^2 \tag{371}$$

$$= \|\mathbf{I} - (U^*)^\top U\|_F^2 \tag{372}$$

$$\|U_\perp^\top U^*\| = \sigma_{\max}(U_\perp^\top U^*) = \sqrt{\lambda_{\max}((U^*)^\top U_\perp U_\perp^\top U^*)} \tag{373}$$

$$= \sqrt{\lambda_{\max}((U^*)^\top U_\perp U_\perp^\top U_\perp U_\perp^\top U^*)} = \|U_\perp U_\perp^\top U^*\| = \|(\mathbf{I} - UU^\top)U^*\| \tag{374}$$

Note that for $\|z\| = 1$

$$1 = z^\top U^\top U z = z^\top U^\top U^*(U^*)^\top U z + z^\top U^\top U^*_\perp (U^*_\perp)^\top U z \tag{375}$$

$$\implies 1 - z^\top U^\top U^*(U^*)^\top U z = z^\top U^\top U^*_\perp (U^*_\perp)^\top U z \tag{376}$$

$$\implies 1 - \min_{\|z\|=1} z^\top U^\top U^*(U^*)^\top U z = \max_{\|z\|=1} z^\top U^\top U^*_\perp (U^*_\perp)^\top U z \tag{377}$$

$$\implies 1 - \sigma^2_{\min}((U^*)^\top U) = \|(U^*_\perp)^\top U\|^2 \tag{378}$$

Therefore

$$\sigma^2_{\min}(U^\top U^*) + \|U^\top_\perp U^*\|^2 = 1 = \sigma^2_{\min}((U^*)^\top U) + \|(U^*_\perp)^\top U\|^2 \implies \|U^\top_\perp U^*\| = \|(U^*_\perp)^\top U\| \tag{379}$$

Rest of the equality can be obtained in a similar fashion using the above two relations.

$$\|U^\top_\perp U^*\|^2_F = \mathrm{tr}((U^*)^\top U_\perp U^\top_\perp U^*) = \mathrm{tr}((U^*)^\top (\mathbf{I} - UU^\top) U^*) \tag{380}$$

$$= \mathrm{tr}((U^*)^\top (\mathbf{I} - UU^\top)^2 U^*) \tag{381}$$

$$= \|(\mathbf{I} - UU^\top) U^*\|^2_F \tag{382}$$

$$= \|(\mathbf{I} - U^*(U^*)^\top) U\|^2_F = \|(U^*_\perp)^\top U\|^2_F \tag{383}$$

Let $E = (\mathbf{I} - U^*(U^*)^\top) U$ and $Q = (U^*)^\top U$. Then $U^\top E = \mathbf{I} - Q^\top Q$. Then by Weyl's inequality (Lemma F.2, by setting $A \leftarrow \mathbf{I}$, $B \leftarrow Q^\top Q$, and $C \leftarrow U^\top E$) we get that

$$1 - \sigma_r(Q)^2 = \sigma_r(\mathbf{I}) - \sigma_r(Q^\top Q) \le \|U^\top E\| \le \|U\|\|E\| \le \|(\mathbf{I} - U^*(U^*)^\top) U\| \tag{384}$$

This implies that $\sigma_r((U^*)^\top U) \ge \sqrt{1 - \|(\mathbf{I} - U^*(U^*)^\top) U\|}$ $\qquad\square$

**Lemma F.5** (Hanson-Wright inequality, Theorem 6.2.1 [50]). *Let $x_1, \ldots, x_m \sim \mathcal{N}(0, \mathbf{I}_{d \times d})$ be $m$ i.i.d. standard isotropic Gaussian random vectors of dimension $d$. Then, for some universal constant $c \ge 0$, the following holds true with a probability of at least $1 - \delta$.*

$$\left| \frac{1}{m} \sum_{j=1}^m x_j^\top A_j x_j - \frac{1}{m} \sum_{j=1}^m \mathrm{tr} A_j \right| \le c \max \left( \sqrt{\sum_{j=1}^m \|A_j\|^2_F \frac{\log(1/\delta)}{m^2}}, \max_{j=1,\ldots,n} \|A_j\|_2 \frac{\log(1/\delta)}{m} \right) \tag{385}$$

**Lemma F.6.** *Let $x_1, \ldots, x_m \sim \mathcal{N}(0, \mathbf{I}_{d \times d})$ be $m$ i.i.d. standard isotropic Gaussian random vectors of dimension $d$. Then, for some universal constant $c \ge 0$, the following holds true with a probability of at least $1 - \delta$.*

$$\left| \frac{1}{m} \sum_{j=1}^m a^\top (x_j x_j^\top) b - a^\top b \right| \le c\|a\|\|b\| \max \left( \sqrt{\frac{\log(1/\delta)}{m}}, \frac{\log(1/\delta)}{m} \right) \tag{386}$$

*Proof.* First notice that $a^\top (x_j x_j^\top) b = \mathrm{tr}(a^\top (x_j x_j^\top) b) = \mathrm{tr}(x_j^\top b a^\top x_j) = x_j^\top b a^\top x_j$ and $a^\top b = \mathrm{tr}(b a^\top)$. Then desired result follows from Lemma F.5, by setting $A_j = b a^\top$. . $\qquad\square$

**Lemma F.7.** *Let $x_1, \ldots, x_m \sim \mathcal{N}(0, \mathbf{I}_{d \times d})$ be $m$ i.i.d. standard isotropic Gaussian random vectors of dimension $d$. Then, for some universal constant $c \ge 0$, the following holds true with a probability of at least $1 - \delta$.*

$$\left\| \frac{1}{m} \sum_{j=1}^m a_j x_j x_j^\top - \frac{1}{m} \sum_{j=1}^m a_j \mathbf{I} \right\| \le c \max \left( \frac{\|a\|_2}{\sqrt{m}} \sqrt{\frac{d\log(9) + \log(1/\delta)}{m}}, \|a\|_\infty \frac{d\log(9) + \log(1/\delta)}{m} \right) \tag{387}$$

*Proof.* For $\epsilon \leq 1$, consider a unit vector $u \in N_\epsilon$ from the $\epsilon$-net of size $|N_\epsilon| = (1 + 2/\epsilon)^d$, of the sphere $\mathbb{S}^{d-1}$ [49, Lemma 5.2]. That is for any $u' \in \mathbb{S}^{d-1}$, there exists some $u \in N_\epsilon$ such that $\|u' - u\| \leq \epsilon$.

Now we will prove a concentration for $\frac{1}{m}\sum_{j=1}^m a_j u^\top x_j x_j^\top u - \frac{1}{m}\sum_{j=1}^m a_j$. Notice that, $a_j u^\top (x_j x_j^\top) u = a\text{tr}(u^\top (x_j x_j^\top) u) = a_j \text{tr}(x_j^\top u u^\top x_j) = x_j^\top (a_j u u^\top) x_j$ and $\text{tr}(a_j u u^\top) = a_j$. Then, by Hanson-Wright inequality (Lemma F.5), for some universal constant $c \geq 0$, the following holds true with a probability of at least $1 - \delta'$.

$$\left| \frac{1}{m}\sum_{j=1}^m a_j u^\top x_j x_j^\top u - \frac{1}{m}\sum_{j=1}^m a_j \right| \leq c \max\left( \frac{\|a\|_2}{\sqrt{m}} \sqrt{\frac{\log(1/\delta')}{m}}, \|a\|_\infty \frac{\log(1/\delta')}{m} \right) \qquad (388)$$

This implies that, through union bound, for the matrix $A' = \frac{1}{m}\sum_{j=1}^m a_j x_j x_j^\top - \frac{1}{m}\sum_{j=1}^m a_j \mathbf{I}$ the following holds true with probability at least $1 - \delta$

$$u^\top A' u \leq c \max\left( \frac{\|a\|_2}{\sqrt{m}} \sqrt{\frac{\log(|N_\epsilon|/\delta)}{m}}, \|a\|_\infty \frac{\log(|N_\epsilon|/\delta)}{m} \right), \quad \text{any } u \in N_\epsilon \qquad (389)$$

Let $u' \in \mathbb{S}^{d-1}$ be the top singular-value of $A'$, then there exists some $u \in N_\epsilon$ such that $\|u' - u\| \leq \epsilon$.

$$\sigma_{\max}(A') = (u')^\top A' u' = (u' - u)^\top A' u' + u^\top A' u + u^\top A' u \qquad (390)$$
$$\leq \|u' - u\|\sigma_{\max}(A')\|u'\| + \|u\|\sigma_{\max}(A')\|u' - u\| + u^\top A' u \qquad (391)$$
$$\qquad (392)$$

Re-arranging and setting $\epsilon = 1/4$ and setting $c \leftarrow 2c$, we get

$$\sigma_{\max}(A') \leq \frac{u^\top A' u}{1 - 2\epsilon} \leq 2c \max\left( \frac{\|a\|_2}{\sqrt{m}} \sqrt{\frac{d\log(9) + \log(1/\delta)}{m}}, \|a\|_\infty \frac{d\log(9) + \log(1/\delta)}{m} \right)$$
$$\qquad (393)$$

$\square$

# G  Sample complexity gain of AltMin over MoM

For the purpose of illustration, suppose that $\sigma = 0$ and $t = 1$, that is we are solving a noiseless single-task regression problem. Here AltMin solves the simple problem

$$\min_u \frac{1}{m}\sum_j (y_j - u^\top x_j)^2 \equiv \min_u (u - u^*)^\top S(u - u^*) \qquad (394)$$

where $S = \frac{1}{m}\sum_j x_j x_j^\top$ is the empirical data covariance matrix. It can exactly recover $u^*$ using just $d$ samples, as $S$ will then be full-rank (with high probability).

However, a 2nd order MoM [35], solving

$$\min_u 1/m \sum_j y_j x_j^\top u \equiv \min_u u^{*\top} S u \qquad (395)$$

achieves $\epsilon$ error only if $S = \frac{1}{m}\sum_j x_j x_j^\top$ is close enough to the true identity covariance matrix $\mathbf{I}_{d \times d}$. This needs at least $\tilde{O}(1/\epsilon^2)$ samples by simple covariance matrix concentration arguments.

This is the intuition behind the fact that, even when $\sigma = 0$ and $t > 1$, our AltMinGD-S achieves $\epsilon$ error using just $\tilde{O}(d)$ samples (Corollary 4), but MoM needs at least $\tilde{O}(d/\epsilon^2)$ samples [48, Theorem 7]. Empirically this is observed in Figure 1a, where the error of AltMin and AltMinGD decreases as $\sigma$ decreases, while the error of MoM and MoM2 remains almost constant. Similar phenomenon is also observed in Figures 1b, and 1c, where there is a constant gap between curves of AltMin and AltMinGD and MoM and MoM2 in these loglog plots, even though the asymptotic convergence rates of AltMin or AltMinGD and MoM or MoM2 in terms of $m$ and $t$ are similar.

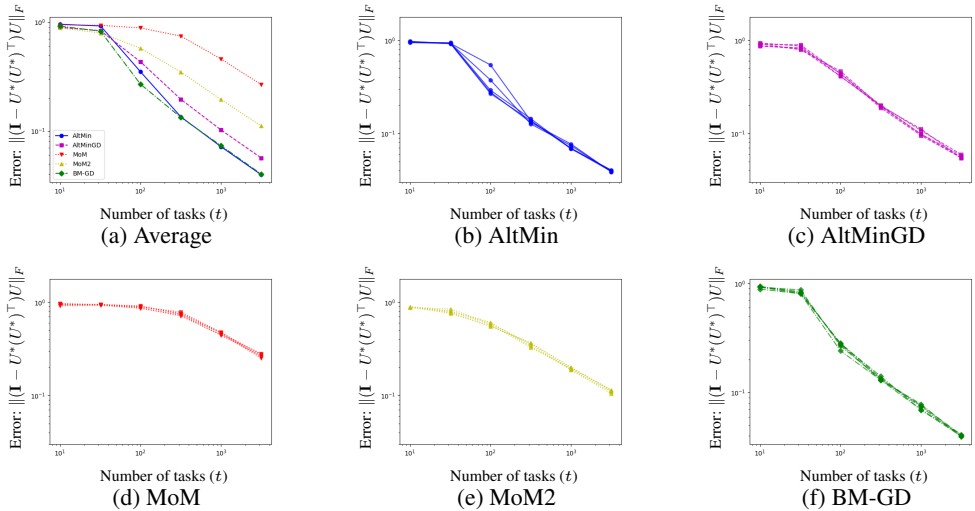

Figure 4: Individual trials of each algorithm when varying number of tasks $t$

## H   Experimental details

We empirically compare the performance of our methods AltMinGD (Algorithm 1) and exact minimization variant AltMin (Algorithm 3 in Appendix), two different versions of Method-of-Moments (MoM [48], MoM2 [35]), and simultaneous gradient descent on $(U, V)$ using the Burer-Monteiro factorized loss (4) (BM-GD [48]). We generate data samples with dimension $d = 100$ and generate random subspace $U^*$ of rank $r = 5$. We sample the task specific true regression parameter from the standard isotropic Gaussian distribution, i.e. $v^{*(i)} \sim \mathcal{N}(0, \mathbf{I})$. In all the figures, the magenta dashed line with square marker represents AltMinGD, the blue straight line with circular marker denotes the AltMin , the red dotted line with downwards pointing triangular marker denotes the MoM, the yellow dotted line with upwards pointing triangular marker represents the MoM2, and the green dashed and dotted line with diamond marker represents the BM-GD. In all the figures we plot the subspace estimation error of the output $U$ of the algorithms. The error is calculated using the rescaled Frobenius norm $\|(\mathbf{I} - U^*(U^*)^\top)U\|_F/\sqrt{r}$, which takes a value in the interval $[0, 1]$.

Figure 1a plots subspace distance against the standard deviation $\sigma$ of the regression noise, $\varepsilon_j^{(i)} \sim \mathcal{N}(0, \sigma^2)$; see (2). We vary $\sigma$ from $10^{-3}$ to $10^2$, while fixing the number of tasks at $t = 200$ and the number of samples per task at $m = 25$. We initialize AltMinGD, AltMin, and BM-GD uniformly at random and run them for $K = 100$, $K = 20$ and $K = 500$ iterations, respectively. AltMinGD and BM-GD use stepsizes $\eta = 1.0$ and $\eta = 0.1$ respectively.

Figure 1b plots the subspace error against the number of tasks $t$. We vary $t$ from 10 to 3163, while the number of samples per task is fixed at $m = 25$ and $\sigma = 1$. We initialize AltMinGD, AltMin, and BM-GD uniformly at random and run them for $K = 200$, $K = 20$ and $K = 500$ iterations, respectively. AltMinGD and BM-GD use stepsizes $\eta = 1.0$ and $\eta = 0.1$ respectively. In Figure 4a, we plot the average over 5 trials for each algorithm. The individual trials for each algorithm are plotted in Figure 4. In Figure 1c, we plot the the error against the number samples per tasks $m$. We vary $m$ from 5 to 78125, while fixing the number of tasks at $t = 20$ and the standard deviation of the regression noise at $\sigma = 1$. We initialize AltMinGD, AltMin, and BM-GD uniformly at random from the set of all orthonormal rank-$r$ matrices and run them for $K = 1000$, $K = 20$ and $K = 500$ iterations, respectively. AltMinGD and BM-GD use stepsizes $\eta = 0.2$ and $\eta = 0.1$ respectively. In Figure 5a, we plot the average over 5 trials for each algorithm. The individual trials for each algorithm are also plotted in Figure 5. We see that BM-GD is very ustable even at a lower or comparable stepsize than AltMinGD.

In Figure 3 we plot the subspace estimation error against the number of iterations of the iterative method AltMinGD, AltMin, and BM-GD for varying levels of task diversity/incoherence (Assumption 2). We vary task diversity while fixing the random noise magnitude at $\sigma = 1$, the number of

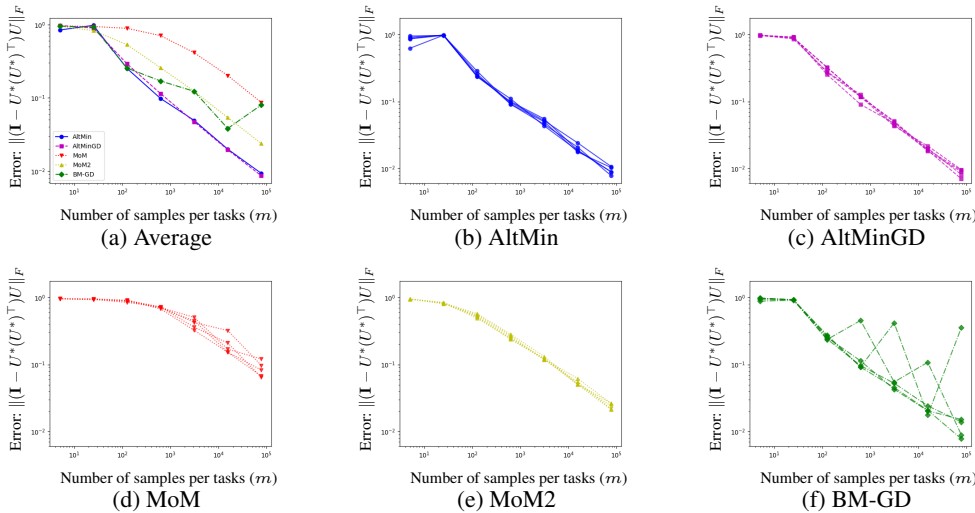

Figure 5: Individual trials of each algorithm when varying number of samples per task $m$

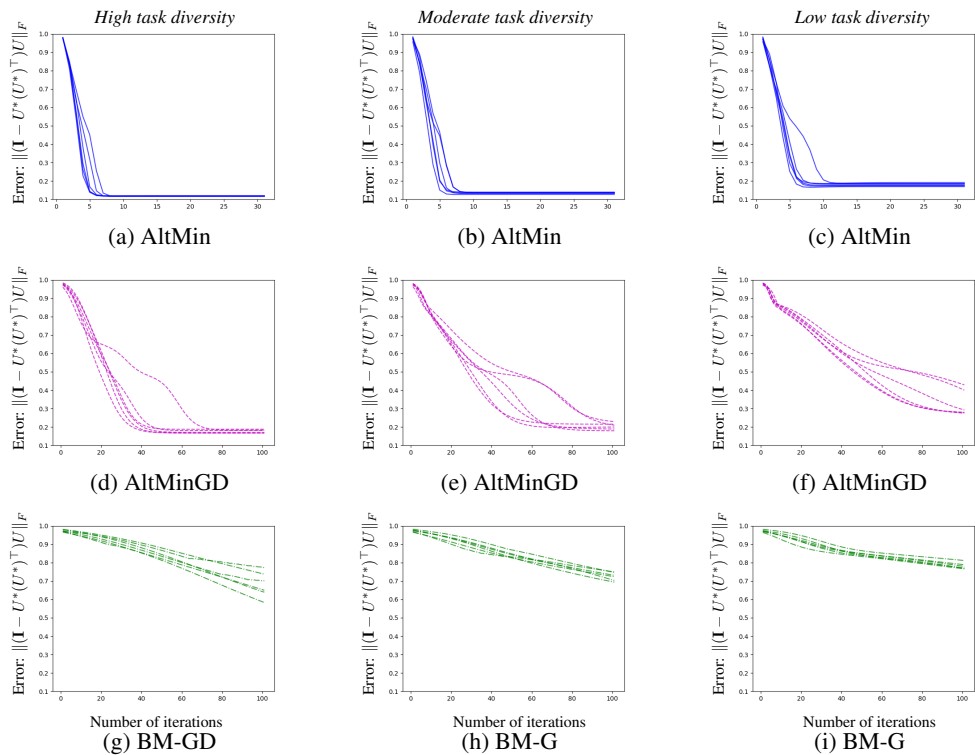

Figure 6: Individual trials of each iterative algorithm when plotting against the number of iterations for different task diversities.

tasks at $t = 200$, and the number of samples per task at $m = 25$. We vary the diversity by generating a fraction of the true task regression parameters from the standard isotropic Gaussian distribution, i.e. $v^{(i)} \sim \mathcal{N}(0, \mathbf{I})$, and setting the rest of them as $v^{(i)} = \sqrt{r}e_1 = [\sqrt{r}, 0, \ldots, 0]^\top$. For high task diversity all task parameters are generated randomly, for moderate task diversity $0.4$ fraction of the task parameters are set as $\sqrt{r}e_1$, and for low task diversity $0.8$ fraction of the task parameters are set as $\sqrt{r}e_1$. In Figure 3, we plot the average over 6 trials for each algorithm. The individual trials for each algorithm for each task diversity are plotted in Figure 6.