# OpenReview forum: "Statistically and Computationally Efficient Linear Meta-representation Learning"
_NeurIPS.cc/2021/Conference — NeurIPS 2021 Poster_

### Official Review · Reviewer_v2WH · 2021-07-13

**Rating:** 7
**Confidence:** 4

**Summary:**

The paper analyses the problem of **linear** meta-representation learning, and more particularly the *alternating minimization and gradient descent* algorithm (AltMinGD) that has become empirically popular for meta-learning methods. The authors prove statistical guarantees for this algorithm.
They also propose an alternate version, AltMinGD-S, which achieves better per-task sample complexity, but is less tractable in practice because it is based on a task subset selection which relies on the linearity of the representations.
Then, they show with experiments than the AltMin algorithms achieve smaller errors than the Method of Moments and similar error than the Burer-Monteiro ERM, while being easier to tune. They also study the number of iterations required, and show that the AltMin algorithms (AltMin, AltMinGD, AltMin-S) are the fastest ones.

**Ethical Concerns:**

I do not see any ethical issues with this paper.

**Limitations And Societal Impact:**

The work presented here is very theoretical. To study the possible negative impacts of this kind of work is to study the negative impacts of the whole field.

**Main Review:**

__**Originality:**__  This paper presents a novel and broad statistical analysis in terms of error and sample complexity of AltMin algorithms. [12] also analyzed the statistical guarantees of AltMinGD in the linear case, but only in the noiseless setting and in the view of federated learning. The authors clearly clarify the differences with [12].

__**Quality:**__  The work presented here is considerable, there is a lot of additional results in the appendix, particularly for the AltMin algorithm. The authors state in the paper that the BM-GD algorithm is hard to tune, but the plots provided in the paper do not help in supporting this. The plots in appendix G are important to see the variability of the results of the BM-GD algorithm. The authors discussed the limitations of their work and compared the different bounds on the sample complexity for the algorithms considered.

__**Clarity:**__  I found the paper dense to read, the writing could improve in clarity.
I think there is an error when referring to related work, the notations and the definition of the AltMin algorithm is associated to [41], but this paper does not use the same notations, and it never refers to AltMin. I assume the authors wanted to refer to [24].
The numbering and the formulation of the theorems is not the same between the main paper and the appendix. I think the theorem 1 in the paper corresponds to theorem 9 in the appendix and same for the proof, but it is hard to confirm because the formulations and the quantities involved in the two theorems are different. It seems like the theorem are repeated multiple times (with different formulations), which messes up the numbering and makes it difficult to find for the proof corresponding to each theorem.

__**Significance:**__ I found the paper interesting as it provides a strong analysis of the AltMin algorithms, though it is restricted to the linear case. These algorithms are widely used in practice for meta-learning methods. Researchers could leverage this work for a non-linear analysis of AltMin or put into practice the insights from these results.

**Time Spent Reviewing:**

I spent between 6 to 7 hours to review this paper.

---

> ### Author Response · Authors · 2021-08-11
> **Moving plots to main text and Clarity**
>
> Thank you for your thorough and constructive review.
>
> 1. __Instability of BM-GD__: As per the reviewer’s comments we will include the plots showing the variability of BM-GD over different trials to the main text.
>
> 2. __Clarity__: We will take the reviewer’s feedback on clarity to heart and improve the presentation in the next revision.
>
> 3. __References for AltMinGD__: We apologize for the confusion here. We refer to [41] in Section 3 to point out a successful instance of AltMinGD-like algorithms in past empirical work in meta-representation learning literature. But we would also like to note that previous empirical works ([41], ANIL [36], MetaOptnNet [29], R2-D2 [7], OML [27]) neither referred to this algorithm as AltMinGD, nor related it to the Alternating Minimization framework. However, we did not intend to use [41] as a reference for the notations and the formal AltMin frame-work. Reviewer is correct that we use the notations and the AltMin framework from [24]. We will make this explicit in the next revision to improve the clarity.
> \
> \
> Additionally, we would also like to highlight that while the high-level proof structure is borrowed from matrix sensing literature [24], none of the technical lemmas can be directly used as we lack classical RIP property. Additionally, due to inherent asymmetry in the problem and our sample regime $m \ll 1/\epsilon^2$, we can recover only $U^*$ and not $V^*$. We get around these by relying instead on the {\em task diversity property} (Assumption 2) that allows us to prove all the necessary concentration bounds from scratch (sketched in Appendix A.3).
>
> 4. __Mis-match of theorem numbering__: We apologize for the difference in numbering of the theoretical results in the main text and the appendix. In the main text, we only give an informal version of the theorems hiding many constants and polylog factors. This was done to improve the readability. However, in the main text we failed to properly point out the full versions of these results in the Appendix. We will mark that the results in the main text as informal versions and then provide references and hyperlinks to the their respective formal theorems in the Appendix.

---

### Official Review · Reviewer_j2Hi · 2021-07-16

**Rating:** 7
**Confidence:** 3

**Summary:**

The paper analyses linear meta-representation learning, deriving formally statistical guarantees for widely adopted and empirically successful alternating minimisation algorithms (AltMinGD). With the gained insight, the authors propose a new variant of AltMinGD, named AltMinGD-S, which introduces a task selection mechanism at each alternating iteration to improve efficiency with respect to number of samples per task in high number of tasks scenarios.

**Limitations And Societal Impact:**

No unstated limitation or societal impact concern.

**Main Review:**

The paper provides valuable theoretical insights and statistical bounds on representation meta-learning. The analysis is thorough and well explained, reaching simple and insightful conclusions. The new proposed method of AltMinGD-S is maybe of incremental novelty, but it is well motivated and rooted in the new insights, although I have some doubts on its practical efficacy (see comments). The experiments are not very comprehensive and not very well explained.

Specific comments:

1) It is unclear from the text in several locations whether AltMinGD is an existing algorithm that is analysed to get statistical bounds, or is a new algorithm proposed in this paper. I think it is the former. If so, some sentences and labels might need to be rephrased to avoid saying "our method" or similar when referring to it, e.g. line 285.

2) The organisation of the paper could be improved to make the above distinction clearer and improve the clarity of the paper in general. As far as I understand, sections 2 and 3 are essentially background, with the exception of subsection 3.1, which introduces the idea of  AltMinGD-S. I think a more organic structure should follow how the contributions are presented in the introduction: i) background on AltMinGD and similar, ii) statistical analysis of AltMinGD (firt half of section 4) and then iii) introduce idea of AltMinGD-S based on these insights (subsection 3.1 + second half of section 4). In the current form, I struggled to understand what is statistical guarantees on existing methods and what is novel method.

3) is the new variant AltMinGD-S named AltMin in the abstract and experiments? I am finding it very confusing to track the experimental proofs of the claims in the experiments.

4)  If I understood well the high level idea of AltMinGD-S, it only selects the tasks that it finds having low error at a particular iteration, leaving out the high error tasks which risk to "throw it off". In this way, is not a bias introduced  in the learning? The model is encouraged to learn some tasks very well and just ignore those that it can not do so well. Maybe the experiments disprove this, but I could not find much evidence in them (as pointed out in point 3).

**Time Spent Reviewing:**

3

---

> ### Author Response · Authors · 2021-08-10
> **Novelty, Organization, and Sub-selection scheme in AltMinGD-S**
>
> We thank the reviewer for their detailed and constructive feedback. We hope we satisfactorily addressed the concerns of the reviewer below and our comments positively influence the reviewer’s evaluation of our paper.
>
>
> 1. __Novelty of AltMinGD__: We regret the confusion in the novelty. The reviewer is correct that AltMinGD is indeed a previously known algorithm [Section 3.1, 41] whose variants appear in many past works (ANIL [36], MetaOptnNet [29], R2-D2 [7], OML [27]) in the context of practical neural meta-representation learning. However, note that previous works neither referred to this algorithm as AltMinGD, nor related it to the Alternating Minimization framework. In this paper, we instantiate the AltMinGD algorithm in the context of solving the canonical linear regression tasks using a linear representation parameterized by a low-rank orthonormal matrix $U^*$. This instantiation is also novel. We will rephrase our sentences to explicitly reflect these.
>
> 2. __Organization__: We thank the reviewer for the detailed instructions for improving the organization of our paper. We will revise our paper according to the reviewer’s suggestions.
> \
> \
> In this paper, we characterize the statistical sample complexity of AltMinGD for learning the representation $U^*$ and show that AltMinGD achieves the optimal statistical complexity of solving this problem. Particularly, we show that the practically popular AltMinGD requires only $\widetilde{\Omega}(d \sigma^2/t \varepsilon^2)$ samples per-task to learn the representation. Which means that, as long we have data from enough $\widetilde{\Omega}(d/\varepsilon^2)$ number of tasks, we only need $\widetilde{\Omega}(1)$ samples per-task to learn the true meta-representation parameter $U^*$ upto $\varepsilon$ error. Thus we theoretically validate the main assumption of meta-representation learning literature: we only need a few samples per-task as long as we have data from a large enough number of tasks $t$. Further we show that this statistical complexity is optimal.
> \
> \
> __Known Guarantees__: As far as we know, this is the first such statistical guarantee for AltMinGD, even though the algorithm was heavily adopted in practice [41, 36, 29, 7, 27]. There have been a few other works which studied the statistical complexity of other algorithms for solving the same meta-representation learning problem (Table 1). However, either these methods are computationally intractable or impractical (Non-convex ERM [14] & Method-of-Moments [42, 28]), or their per-task sample complexity guarantee does not optimally decrease with the number of tasks whose data is available (Burer-Monteiro ERM [42]). Please refer to Section 4.1 and Table 1 for more details.
>
> 3. __AltMin vs AltMinGD-S__: We would like to highlight that AltMin (Algorithm 3, given in Appendix A and referenced in Section 5) and AltMinGD-S (Algorithm 2) are two different algorithms. AltMin is the exact alternating minimization algorithm where the Gradient Descent step for $U$ in AltMinGD is replaced with a full minimization step w.r.t. $U$. Although this has a better statistical guarantee (Theorem 5 in Appendix A) than AltMinGD/AltMinGD-S (Theorem 1/3), AltMin’s per iteration computational complexity $O((dr)^3 + mt \cdot (dr)^2)$ (lines 507-515 in Appendix A) is much higher than that of AltMinGD/AltMinGD-S $O(mt \cdot dr)$ (lines 148-154). Further, for practical neural meta-representation models, AltMin is impractical due the impossibility of the exact minimization of the weights of the shared meta-representation network. We will make this more clear in the next revision.
> \
> \
> __Efficiency of AltMinGD-S in practice__: We would also like to clarify that in the experiments we only compare AltMinGD and AltMin. AltMinGD-S is omitted because we do not observe any gain for it under our settings, as AltMinGD-S’s logarithmic gain ($1/\log(t)$) will only be observed when we have an exponentially large number of tasks ($t$). Exponentially large number of tasks are difficult to simulate using our modest computing hardware. However, for internet scale problems we believe similar kinds of subset select schemes may be useful. We will clarify this in the next revision.
> \
> \
> __Robustness to Adversarial Tasks__: We thank the reviewer for making us realize that we left some of our thoughts about AltMinGD-S unsaid in the manuscript. We want to emphasize that there is potentially significant practical gain of AltMinGD-S when the data is corrupted. We proved that AltMinGD-S is robust to a small number of random natural outlier tasks which might occur when the number of tasks $t$ is large [Theorem 3]. However, we believe that AltMinGD-S could also be robust to a large number of adversarially corrupted tasks.
> \
> \
> Consider the Huber contamination model [CGR] where a fraction of tasks are corrupted and the data input $x$ is drawn from an arbitrary distribution, different from the standard Gaussian distribution we assume in our setting (the paired dependent variable $y$ is still drawn from the linear model conditioned on this corrupted $x$). We conjecture that AltMinGD-S is naturally __robust__ to such contamination (although a rigorous analysis is outside the scope of this paper). We believe AltMinGD-S is an important new solution to _“robust meta-learning”_ with corrupted tasks that we plan on exploring. Although there have been significant breakthroughs recently on robust linear regression, there is little known about practical algorithms and provable guarantees for the linear meta-learning setting with robustness. We will design and add the numerical experiments testing such robustness of AltMinGD-S, under the Huber contamination model.
> \
> \
> [CGR] Chen, Gao, & Ren, (2018). Robust covariance and scatter matrix estimation under Huber’s contamination model. The Annals of Statistics, 46(5), 1932-1960.
>
> 4. __Bias in the Subset Selection scheme of AltMinGD-S__: We agree with the reviewer that discarding some tasks at each iteration, may introduce a bias in the training data at iteration. Our subset selection scheme is essentially trading-off between this bias introduced by the selection scheme, and the reduction in the variance of the projected covariance matrix $\widetilde{U}^\top S^{(i)} \widetilde{U}$ (step 3 of Algorithm 2, AltMinGD-S) through this scheme. The scheme reduces the explicit dependence of the per-task number of samples $m$ to the number of tasks $t$, by adding a new requirement that the number of tasks should be at least $t \geq \widetilde{\Omega}(\mu^2 r^3 \kappa)$ [Theorem 5]. This requirement ensures that our bias is low by ensuring that only a $O(1/\mu r)$ fraction of the tasks are discarded at each step [Lemma D.1 in Appendix D]. As long as $r$, $\mu$ and $\kappa$ are not exponentially large, this is a negligible requirement on the number of tasks $t$ in the regime of our interest where the number of tasks are exponentially large, where we can see a gain in using AltMinGD-S over AltMin.

---

> > ### Comment · Reviewer_j2Hi · 2021-08-11
> > **Rebuttal response**
> >
> > Thank you for the detailed answers. My concerned have been mostly addressed and I will raise the score accordingly. However, I do believe some of the explanations given in this reply should go in some form in the main body. The answers to my points 3 and 4 both have to do with the expected gain of AltMinGD-S to be in very large systems/data sets. I think this should be clearly explained (as you did above) and especially stated as the reason for it being missing from the experiments in the beginning of the experimental section.

---

> > > ### Author Response · Authors · 2021-08-12
> > > **Thank you for your feedback and evaluation**
> > >
> > > Thank you for your quick response and for evaluating our work higher. We will surely add these points and explanations to our next revision. Again, we appreciate your feedback to improve our paper.

---

### Official Review · Reviewer_HZi9 · 2021-07-16

**Rating:** 7
**Confidence:** 3

**Summary:**

This paper provides sample complexity and error bounds for “alternating gradient descent minimization” (AltMinGD), an algorithm used for meta-learning. It shows that AltMinGD applied to linear regression is superior to competitors such as Empirical Risk Minimization (ERM) and Method of Moments (MoM). Furthermore, they show that the bounds are tight and basically cannot be improved upon.


**Ethical Concerns:**

No ethical concerns

**Limitations And Societal Impact:**

No societal impact

**Main Review:**

Even if results apply to linear representations only, I find this work very compelling and useful.
In the last couple of years, increasing evidence shows that linear meta-representation learning is a very useful and tractable framework to understand meta-learning, and to gain some intuition about the phenomena observed in more complicated (non-linear) contexts.
The paper is well written and I think it can be published in its current form.

I only have a couple of clarification questions:
1) By assumption the matrix U is orthogonal. My understanding is that AltMinGD enforces orthogonality of U by a QR decomposition at the end of training. This step is not even mentioned in the text and it seems very brute force. How important is the assumption of orthogonal U? Why doesn’t the QR last step ruin the representation obtained during training?
2) What is the intuition behind the superior performance of AltMinGD with respect to ERM and MoM?

Other comments:

- A number of papers about meta-learning and linear regression have been published recently that the authors have not considered: https://arxiv.org/abs/2102.00940, https://arxiv.org/abs/2010.05843, https://arxiv.org/abs/2010.12916, https://arxiv.org/abs/2010.14672
- In line 124 (Assumption 2), the condition number k is introduced but not used.
- Line 171. How is the number 4 selected as maximum condition number for the Hessian?


**Time Spent Reviewing:**

4

---

> ### Author Response · Authors · 2021-08-10
> **QR decomposition and the intuition behind the gain of AltMin over ERM and MoM**
>
> We thank the reviewer for the positive review, recommendation for publication in its current form, and clarifying questions. We hope our answers positively impact your evaluation and confidence in our work.
>
> 1. **QR decomposition** : We apologize for a typo we made in Algorithm 1 (AltMinGD). We actually analyze a version of AltMinGD which applies the QR decomposition after every $U$ update. That is, the step 6 (QR decomposition) of Algorithm 1 (AltMinGD) should be inside the main “for loop”, not outside. The correct update rule for AltMinGD is present in our analysis in Appendix C [Eqns (210-212)]. QR step does not change the rank-$r$ column space of $U$, but it only finds an $r$-rank orthonormal matrix $U^+$ for the same column space. That is $U = U^+ R$, where $R$ only acts on the row space. Therefore the distance between true column subspace of $U^*$ and the estimated column subspace remains the same before and after the QR decomposition. \
> \
> We can still recover similar results for the algorithm which does not apply QR decomposition after any $U$ update. However, this degrades the sample complexity and statistical error rates in terms of the condition number factors of the task diversity matrix (Assumption 2). With the QR decomposition we ensure that the magnitude of $U$ does not increase or decrease too fast. This ensures that we get a tighter sample complexity and statistical error rates.  \
> \
> For our meta-representation learning problem over the all rank-$r$ subspaces (in $d$-dimensional vector space) parameterized by $U$ ([Grassmanian manifold](https://en.wikipedia.org/wiki/Grassmannian)), one could think of QR decomposition as projection operator to the set of all $r$-rank orthonormal matrices which ensures that the magnitude of $U$ does not grow too fast due to the gradient update.  \
> \
> Notice that for AltMin (Algorithm 3 in Appendix A), even though we apply QR decomposition to $U$ at the end of every iteration, it is not a necessity. All the analysis still follows through even without the QR decomposition due to a simple equivalence argument between the subspaces obtained with or without the QR decomposition. However, this equivalence argument critically uses the fact that we exactly minimize $U$ for the current $V$. Thus the same equivalence cannot be proved for AltMinGD which uses gradient descent to update $U$, because the gradient scaling may change across iterations due to changes in the magnitude of $U$.
>
>
> 2. __AltMin(GD) vs ERM__ : We believe that Alternate Minimization based approaches (AltMin, AltMinGD) performs better than ERM based methods (Burer-Monteiro GD = BM-GD), because the optimization landscape of the ERM problem jointly over $(U,V)$ parameters may contains a lot of bad local minima in the finite sample regime [Theorem 2, 42]. This implies that with high-probability the per-task sample complexity guarantee of BM-GD method $\widetilde{\Omega}(\sigma^2/\varepsilon^2)$ does not asymptotically decrease with the number of tasks [Table 1]. Whereas, AltMinGD has a per-task sample complexity of only $\widetilde{\Omega}(d \sigma^2/t \varepsilon^2)$, which decreases with $t$ [Table 1]. Note that $\widetilde{\Omega}$ suppresses poly-logarithmic factors.
> \
> \
> We empirically observed that BM-GD may converge to bad local minima. We also observed that the BM-GD algorithm is very unstable over multiple trials, even at a lower or comparable step-size than AltMinGD [Figs. 3(f) and 5(f) in Appendix G].
> \
> __AltMin(GD) vs MoM__ : For the purpose of illustration, suppose $\sigma=0$ \& $t=1$, that is we are solving a noiseless single-task regression problem. AltMin solves $\min_{u}  1/m \sum_{j} (y_j - u^\top x_j)^2 = \min_{u} (u - u^*)^\top S(u - u^*)$, where $S = 1/m \sum_{j} x_j x_j^\top$ is the empirical data covariance matrix. It can exactly recover $u^*$ using just $d$ samples, as $S$ will then be full-rank (with high probability).
> \
> However, a 2nd order MoM solving $\min_{u}  1/m \sum_{j}  y_j x_j^\top u = \min_{u} u^{*\top}Su$ achieves $\epsilon$ error only if $S = 1/m \sum_{j} x_j x_j^\top$ is close enough to the true identity covariance matrix $\mathbf{I}_{d \times d}$. This needs at least $\tilde{O}(1/\epsilon^2)$ samples by simple covariance matrix concentration arguments.
> \
> \
> This is the intuition behind the fact that, even for $\sigma=0$ \& $t>1$, our AltMinGD-S achieves $\epsilon$ error using just $\tilde{O}(d)$ samples [Corollary 4], but MoM needs at least $\tilde{O}(d/\epsilon^2)$ samples [Theorem 7, 42].
> \
> Empirically this is observed in Figure 1(a), where the error of AltMin/AltMinGD decreases as $\sigma$ decreases, while the error of MoM/MoM2 remains almost constant. Similar phenomenon is also observed in Figures 1(b) and 1(c), where there is a constant gap between curves of AltMin/AltMinGD and MoM/MoM2 in these loglog plots, even though the asymptotic convergence rates of Alt/AltMinGD and MoM/MoM2 in terms of $m$/$t$ are similar.
>
>
> 3. We thank the reviewer for the references. We will cite these recent works fairly in the next revision. For the subset selection scheme [Section 3.1], choosing $4$ as the maximum Hessian condition number is slightly arbitrary. Similar results can be obtained for any constant $O(1)$ maximum condition number greater than $1$. This is possible because in expectation the projected covariance matrix $E [ U^\top x_j^{(i)} (U^\top x_j^{(i)})^\top]$ is identity $\mathbf{I}_{r \times r}$ with all $1$ eigenvalues.

---

### Official Review · Reviewer_GdPm · 2021-07-18

**Rating:** 7
**Confidence:** 3

**Summary:**

**Summary**:
This paper analyses the statistical and computational properties of one particular type of meta-representation learning approaches, which alternatively optimises a shared feature extractor that produces linear representations, and solves inner task-specific linear models taking in the shared linear representations. Under the canonical settings when the feature extractor is linear, they show that the meta-representation learning algorithm can achieve near optimal error rate.

**Contributions**:
- This paper analyses one family of widely adopted meta-representation learning algorithms they call AltMinGD, and prove a near optimal error rate, following the proof strategy for matrix sensing (which is still significantly different from the linear meta-representation learning problem).
- This paper introduces a new variation of AltMinGD-S, motivated by their analysis. The new algorithm selects a subset of well-behaved tasks at each iteration, and requires a much smaller number of samples per task to estimate representation.
- This paper also analyses a variation called AltMin (in appendix) where the outer loop feature extractor is optimised exactly, and proves slightly improved guarantees.



**Ethical Concerns:**

I have no ethical concerns over this work.

**Limitations And Societal Impact:**

**Limitations**:
This paper is under a simplified setting where the shared representation is linear, and all tasks are considered when updating the feature extractor (non-stochastic optimisation). But I think the conclusions are still useful, and these are sensible simplifications as a first work to analyse this algorithm.

**Societal Impact**:
This is a theoretical work and I have no concerns over its negative societal impact.



**Main Review:**

**Originality**:
Although there are previous theoretical studies trying to analyse the statistical and computational properties of meta-representation learning algorithms, they often focus on algorithms that have little resemblance to those used in practice, or provide a highly suboptimal error rate. This paper focuses on one particular type of algorithm called AltMinGD, which is successful and popular among previous work. They show that in the canonical setting when the learned shared representation is linear, their analysis of this algorithm would give a near-optimal error rate.

Even though the proving strategy is inspired by those from matrix sensing. The two problems are significantly different, and this paper proves all necessary concentration bounds.

They also introduce a new variation called AltMinGD-S that would require much fewer samples per task and analyse this new algorithm.

**Quality**:
This work has clear motivation and I think they are tackling an important problem with practical implications. However, I am not familiar with related work and the theoretical work on meta-representation learning, I am afraid I cannot comment more on the quality of their analysis.

I would like to only highlight one of my concerns below. I would appreciate it if the authors could give an explanation: If I understand correctly, many previous work on meta-representation learning such as MetaOptNet, ANIL, etc, actually backpropagate through the inner loop minimisation when updating their feature extractors. However, in this paper, (if I am not mistaken), the algorithms *alternatively* optimise the feature extractor and solve the inner loop optimisation for each task. When computing the gradient w.r.t the feature extractor parameters $U$, do you consider the fact that the optimal inner parameter $V^{\ast}$ is actually dependent on $U$ , and compute gradients accordingly? If yes, I have no further concerns. If not, could the authors discuss whether this difference might lead to different conclusions in their analysis?

**Clarity**:
The motivation and assumption of this work are clear and well presented. The theoretical analysis part is intense if the reader did not read background references, but I think the authors have already put effort into making it clear.

**Significance**:
I believe this paper is tackling an important problem of meta-representation learning that is of practical relevance. The conclusions and proof strategy presented in this work can also be useful for future work.


**Time Spent Reviewing:**

5

---

> ### Author Response · Authors · 2021-08-10
> **Back-propagating through the inner minimization over $V$**
>
> We thank the reviewer for the positive review and their comprehensive summary of our work.
>
> **Back-prop through optimal $V$** : That is a great observation! Yes, the reviewer is correct that the methods used in practice (MetaOptNet, ANIL, etc.) back-propagate through their respective inner inexact optimization algorithm.
>
> We do use the fact that optimal $V^*(U)$ for the current $U$ is dependent on U [Eqn (210) in Appendix C]. Since we do exact inner minimization with respect to $V$, by a generalization of the [Danskin’s theorem](https://en.wikipedia.org/wiki/Danskin%27s_theorem), back-propagating through the inner exact minimization w.r.t. $V$, is equivalent to computing the gradient $\nabla_U \mathcal{L}(U, V)$ with respect to $U$ and then setting $V=V^*(U)$, where $V^*(U)$ is the minimizer for the current $U$. That is,
>
> $\nabla_U \min_V L(U, V) = \nabla_U L(U, V^*(U))$, where $V^*(U) \in \arg\min_V L(U, V) $
>
> This implies that our AltMinGD algorithm does back-propagate through the inner exact minimization step. We will make this more clear in the next revision.

---

### Decision · Program_Chairs · 2021-09-27

**Decision:**

Accept (Poster)

**Comment:**

All reviewers agree that this is a solid contribution to NeurIPS: it tackles a relevant problem in the context of meta-learning and does so by proposing a novel and interesting strategy inspired by the matrix sensing literature. The authors provide a thorough theoretical investigation of the setting considered, which yields insights also on previous work on the topic of linear meta-representation learning. There are a few issues with clarity that authors have addressed during the discussion period and that will need to be taken care of in the final version of the paper. In particular, including material from the Appendix (e.g. plots concerning the behavior of Burer-Monteiro gradient descent) and adding further insight regarding the distinction between AltMin and AltMinGD-S (also explaining why this was not tested in the experiments).